# Neural silences can be localized rapidly using noninvasive scalp EEG

Alireza Chamanzar [1,2✉], Marlene Behrmann[2,3] & Pulkit Grover [1,2✉]

A rapid and cost-effective noninvasive tool to detect and characterize neural silences can be of important benefit in diagnosing and treating many disorders. We propose an algorithm, SilenceMap, for uncovering the absence of electrophysiological signals, or neural silences, using noninvasive scalp electroencephalography (EEG) signals. By accounting for the contributions of different sources to the power of the recorded signals, and using a hemispheric baseline approach and a convex spectral clustering framework, SilenceMap permits rapid detection and localization of regions of silence in the brain using a relatively small amount of EEG data. SilenceMap substantially outperformed existing source localization algorithms in estimating the center-of-mass of the silence for three pediatric cortical resection patients, using fewer than 3 minutes of EEG recordings (13, 2, and 11 mm vs. 25, 62, and 53 mm), as well for 100 different simulated regions of silence based on a real human head model (12 ± 0.7 mm vs. 54 ± 2.2 mm). SilenceMap paves the way towards accessible early diagnosis and continuous monitoring of altered physiological properties of human cortical function.

[1] Electrical and Computer Engineering Department, Carnegie Mellon University, Pittsburgh, PA, USA. [2] Neuroscience Institute, Carnegie Mellon University, Pittsburgh, PA, USA. [3] Psychology Department, Carnegie Mellon University, Pittsburgh, PA, USA. ✉email: achamanz@andrew.cmu.edu; pgrover@andrew.cmu.edu

An ongoing challenge confronting basic scientists, as well as those at the translational interface, is the ability to access a rapid and cost-effective tool to uncover mechanistic details of neural function and dysfunction. For example, identifying the presence of stroke, establishing altered neural dynamics in traumatic brain damage, and monitoring changes in neural profile in athletes on the sidelines all pose major hurdles. In this paper, using scalp electroencephalography (EEG) signals with relatively little data, we provide theoretical and empirical support for a method for the noninvasive detection of neural silences. We adopt the term silences or regions of silence to refer to the areas of brain tissue with little or no neural activity. These regions reflect ischemic, necrotic, or lesional tissue, resected tissue (e.g., after epilepsy surgery), or tumors[1,2]. Dynamic regions of silence also arise in cortical spreading depolarizations (CSDs), which are slowly spreading waves of silences in the cerebral cortex[3–5].

There has been growing utilization of EEG for diagnosis and monitoring of neurological disorders such as stroke[6], and concussion[7]. Common imaging methods for detecting brain damage, e.g., magnetic resonance imaging (MRI)[8,9], or computed tomography[10], are not portable, are not designed for continuous (or frequent) monitoring, are difficult to use in many emergency situations, and may not even be available at medical facilities in many countries. However, many medical scenarios can benefit from portable, frequent/continuous monitoring of neural silences, e.g., detecting changes in tumor or lesion size/location and CSD propagation. Noninvasive scalp EEG is, however, widely accessible in emergency situations and can even be deployed in the field with only a few limitations. It is easy and fast to setup, portable, and of lower cost compared with other imaging modalities. Additionally, unlike MRI, EEG can be recorded from patients with implanted metallic objects in their body, e.g., pacemaker[11].

**Source vs. silence localization.** An ongoing challenges of EEG is source localization, the process by which the location of the underlying neural activity is determined from the scalp EEG recordings. The challenge arises primarily from three issues: (i) the underdetermined nature of the problem (few sensors, many possible locations of sources); (ii) the spatial low-pass filtering effect of the distance and the layers separating the brain and the scalp; and (iii) noise, including exogenous noise, background brain activity, as well as artifacts, e.g., heart beats, eye movements, and jaw clenching[12,13]. In source localization paradigms applied to neuroscience data[14–16], e.g., in event-related potential paradigms[17,18], scalp EEG signals are aggregated over event-related trials to average out background brain activity and noise, permitting the extraction of the signal activity that is consistent across trials. The localization of a region of silence poses additional challenges, of which the most important is how the background brain activity is treated: while it is usually grouped with noise in source localization (e.g., authors in[16] state: "EEG data are always contaminated by noise, e.g., exogenous noise and background brain activity"), estimating where background activity is present is of direct interest in silence localization where the goal is to separate normal brain activity (including background activity) from abnormal silences. Because source localization ignores this distinction, as we demonstrate in our experimental results below, classical source localization techniques, e.g., multiple signal classification (MUSIC)[19,20], minimum norm estimation (MNE)[15,21–23], and standardized low-resolution brain electromagnetic tomography (sLORETA)[24], even after appropriate modifications, fail to localize silences in the brain ("Methods" details our modifications on these algorithms).

To avoid averaging out the background activity, we estimate the contribution of each source to the recorded EEG across all electrodes. This contribution is measured in an average power sense, instead of the mean, thereby retaining the contributions of the background brain activity. Our silence localization algorithm, referred to as SilenceMap, estimates these contributions, and then uses tools that quantify our assumptions on the region of silence (contiguity, small size of the region of silence, and being located in only one hemisphere) to localize it. Because of this, another difference arises: silence localization can use a larger number of time points (than typical source localization). For example, 160 s of data with the sampling frequency of 512 Hz provides SilenceMap with around 81,920 data points to be used, boosting the signal-to-noise ratio (SNR) over source localization techniques, which typically rely on only a few tens of event-related trials to average over and extract the source activity that is consistent across trials.

Further, we confront two additional difficulties: lack of statistical models of background brain activity, and the choice of the reference electrode. The first is dealt with either by including baseline recordings (in absence of silence; which we did not have for our experimental results) or utilizing a hemispheric baseline, i.e., an approximate equality in power measured at electrodes placed symmetrically with respect to the longitudinal fissure (see Fig. 1b). While the hemispheric baseline used here provides fairly accurate reconstructions, we note that this baseline is only an approximation, and an actual baseline is expected to further improve the accuracy. The second difficulty is related: to retain this approximate hemispheric symmetry in power, it is best to utilize the reference electrode on top of the longitudinal fissure (see Fig. 1a). Using these advances, we propose an iterative algorithm to localize the region of silence in the brain using a relatively small amount of data. In simulations and real data analysis, SilenceMap outperformed existing algorithms in localization accuracy for localizing silences in three participants with surgical resections using only 160 s of EEG signals across 128 electrodes (see "Results" for more details on finding the minimum amount of EEG data for localizing silences using SilenceMap).

## Results

SilenceMap localizes the region of silence in two steps: (1) The first step finds a contiguous region of silence in a low-resolution source grid with the assumption that, at this resolution, the sources are uncorrelated across space. In this low-resolution grid, given that the sources are sufficiently separated, a reasonable approximation is to assume they have independent activity (see "Methods" for more details). We defined a measure for the contribution of brain sources to the recorded scalp signals ($\beta$), i.e., the larger the $\beta$, the greater the contribution of the brain source to the scalp potentials. However, $\beta$ is not a perfect measure of the contribution since it is defined based on an identical distribution assumption on the non-silent sources, which does not hold in the real world. Therefore, using $\beta$ does not reveal the silent sources, i.e., the smallest values of $\beta$ (yellow regions in Fig. 1d) may not be located at the region of silence. However, looking closely at the values of $\beta$ in the inferior surface of the brain (Fig. 1d) reveals a large hemispheric color difference at the region of silence (right occipitotemporal lobe). This motivated us to use a hemispheric baseline, i.e., instead of using $\beta$, we use $\tilde{\beta}$, which is the ratio of $\beta$ values of the mirrored sources, e.g., for source pair of ($A_L$, $A_R$), which are remote from the region of silence, $\tilde{\beta}$ is close to 1 (red-colored sources), while for ($B_L$, $B_R$), where $B_R$ is located in the region of silence (see Fig. 1d), this ratio is close to zero (yellow-colored sources). A contiguous region of silence is localized based on a convex spectral clustering (CSpeC) framework[25–27] in the low-resolution grid. (2) The second step of SilenceMap adopts the above localized region of silence as an initial guess in the high-resolution grid. Then, through iterations, the region of silence is

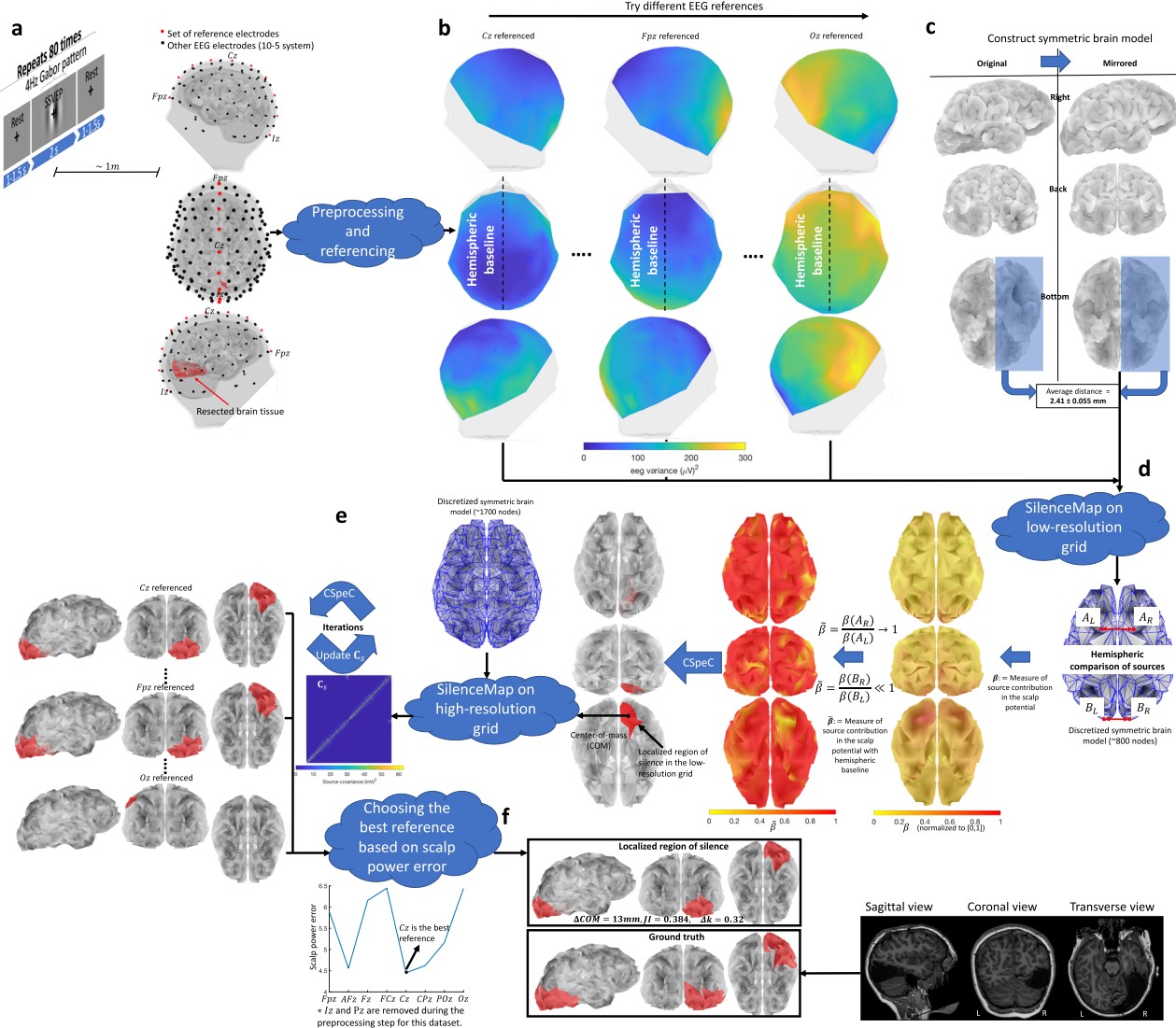

**Fig. 1 SilenceMap with baseline algorithm overview. a** The EEG recording protocol and the locations of scalp electrodes. One of 10 reference electrodes (shown in red) is chosen along the longitudinal fissure for rereferencing against. **b** Average power of scalp potentials for different choices of reference electrodes. **c** Symmetric brain model of a patient (UD) with a right occipitotemporal lobectomy. **d** Steps of SilenceMap in a low-resolution source grid. A measure of the contribution of brain sources in the recorded scalp signals ($\tilde{\beta}$) is calculated relative to a hemispheric baseline. In the brain colormap, yellow indicates no contribution. A contiguous region of silence is localized based on a convex spectral clustering (CSpeC) framework in the low-resolution grid. **e** Steps of SilenceMap in a high-resolution source grid. The source covariance matrix ($\mathbf{C}_s$) is estimated through an iterative method, and the region of silence is localized using the CSpeC framework. **f** Choosing the best reference electrode to reference against (*Cz* in this example), which results in minimum scalp power mismatch (ΔPow). The localized region of silence for this patient (UD) has 13 mm COM distance (ΔCOM) from the original region, with more than 38% overlap (JI = 0.384), and it is 32% smaller (Δk = 0.32).

localized based on the estimated source covariance matrix $\mathbf{C}_s$, until the center-of-mass (COM) of the localized region of silence converged (see "Methods" and Fig. 1e for more details). All steps of SilenceMap, along with the intermediate results for patient UD, are summarized in Fig. 1. We have included similar overview figures for patients SN and OT in Supplementary Fig. 1 and Supplementary Fig. 2, respectively.

We validated the performance of SilenceMap through rigorous experiments, based on simulated and real datasets. We tested the robustness of SilenceMap with and without baseline (see "Methods" for more details), in different scenarios, e.g., different sizes and locations of region of silence, different EEG reference electrodes, and based on both visual and rest EEG datasets (see Fig. 1a). Finally, we used a real dataset to explore the validity of our hemispheric symmetry assumption.

**Localization performance metrics**. For all experiments, we used three performance metrics for determining the accuracy of the silence localization task: (i) center-of-mass (COM) distance (ΔCOM), (ii) Jaccard Index (JI), and (iii) size error (Δk).

(i) COM distance is simply defined as the Euclidean distance between the center-of-mass of the localized and actual region of silence, i.e.,

$$\Delta\text{COM} = \left\| \frac{1}{|\mathcal{S}|}\sum_{i\in\mathcal{S}}\mathbf{f}_i - \frac{1}{|\mathcal{S}^{GT}|}\sum_{i\in\mathcal{S}^{GT}}\mathbf{f}_i \right\|_2, \quad (1)$$

where $\mathbf{f}_i$ is the three-dimensional (3D) location of source $i$ in the brain, $\mathcal{S}$ and $\mathcal{S}^{GT}$ are the set of source indices of the localized region of silence and its ground truth, respectively. ΔCOM

basically measures how far the localized region of silence is from the ground truth.

(ii) JI, first defined by Jaccard in ref. [28], is a widely used performance measure for the 2D image segmentation tasks[29]. In the silence localization task, since we are segmenting the region of silence in 3D space, we can calculate the JI based on the nodes/sources in the discretized brain as follows:

$$\text{JI} = \frac{|\mathcal{S} \cap \mathcal{S}^{GT}|}{|\mathcal{S} \cup \mathcal{S}^{GT}|}, \qquad (2)$$

which measures how well the localized region of silence overlaps with the ground truth region in the brain, and it assumes values between 0 (no overlap) and 1 (perfect overlap). If there is minimal overlap and/or there is a large mismatch between the size of these two regions, JI has a small value.

(iii) Size error measures the error in estimation of the size of the region of silence, and is simply defined as follows:

$$\Delta k = \frac{|k - \hat{k}|}{k}, \qquad (3)$$

where $\hat{k}$ is the estimated number of silent sources in the localized region of silence.

**Simulations**. We simulated the EEG data of regions of neural silence, following the assumptions we made in "Methods: Problem statement", and quantified the performance of SilenceMap.

*Simulation results*. We simulated scalp differential recordings for 100 different regions of silence, with the size of $k = 50$, on a high-resolution source grid with $p = 1744$ sources, and at varying locations on the cortex. The simulated regions of silence lie in only one hemisphere (see the assumptions in "Methods: Problem statement"), and are located no deeper than 3 cm from the surface of scalp, which covers the entire thickness of the gray matter[30–32], while excluding deep sources located in the longitudinal fissure. In the longitudinal fissure, the source dipoles are located deep inside the brain, and mostly oriented tangential to the surface of the scalp, making it difficult for EEG to record their electrical activity[13]. The non-silent sources are assumed to have an identical distribution and correlation across space, and identical distribution over time. To explore the effect of different assumptions for the time-frequency characteristics of neural sources on the silence localization task, we considered two scenarios in the simulations: (i) a flat power spectral density (PSD) profile for the activities of non-silent sources, and (ii) a "Real PSD" profile, which is extracted from an open-source electrocorticography (ECoG) dataset used in[33] and available through the open-source library in ref. [34]. The detailed steps for the simulation are available in "Methods: Simulated Dataset". For SilenceMap, we tried the 10 different reference electrodes located along the longitudinal fissure, i.e., *Fpz, AFz, Fz, FCz, Cz, CPz, Pz, POz, Oz*, and *Iz*, and chose the one with the minimum power mismatch $\Delta$Pow defined in Eq. (39). We reported the performance measures, i.e., $\Delta$COM, JI, and $\Delta k$ under the average signal-to-noise ratio ($\text{SNR}_{avg}$) level of 9 dB (see "Methods" for definition of $\text{SNR}_{avg}$). For SilenceMap, we also reported the convergence rate (CR), which is the ratio of the number of converged cases over the total number of simulated regions of silence in the simulation experiment. The results of the simulations are summarized in Table 1, where $\Delta$COM, JI, and $\Delta k$ are reported in the format of mean ± standard error (SE). Based on the results, SilenceMap outperformed (modifications on) state-of-the-art source localization algorithms: it has 42 mm smaller average COM distance, 41% more average overlap (JI), and 253% smaller size error, compared to the best performing modified source localization algorithms. In addition, SilenceMap

**Table 1 Simulation experiment results ($\text{SNR}_{avg} = 9$ dB, $k = 50$).**

| Algorithms | ΔCOM (mm) | JI | Δk | CR |
|---|---|---|---|---|
| Modified MUSIC | 60 ± 3.5 | 0.09 ± 0.012 | 2.84 ± 0.067 | - |
| Modified MNE | 82 ± 2.2 | 0.01 ± 0.002 | 6.08 ± 0.036 | - |
| Modified sLORETA | 54 ± 2.2 | 0.04 ± 0.002 | 9.62 ± 0.124 | - |
| SilenceMap (flat PSD) | **12** ± 0.7 | 0.50 ± 0.017 | 0.31 ± 0.023 | 0.98 |
| SilenceMap (Real PSD) | 13 ± 0.5 | **0.52** ± 0.015 | **0.28** ± 0.022 | **0.99** |

Bold numbers are the best performance in each column.

showed very similar performance for the flat and Real PSDs, as revealed by examining the effect of changing the (temporal) PSD on the spatial correlation for the simulated brain sources (detailed discussion of this is in "Methods: Simulated dataset"). Our simulations assume identical distribution of brain sources. Noting that SilenceMap with baseline performs substantially better compared to SilenceMap without baseline (see the results in "Results: Real Data"), it appears to us that this assumption is far from accurate. The list of all parameters and their values used in the implementation of SilenceMap is available in Supplementary Note A (see Supplementary Table I).

*Comparison of SilenceMap with source localization algorithms*. We also compared the performance of SilenceMap under different simulated scenarios, as well as real experiments, with state-of-the-art source localization algorithms, modified for the silence localization task (details of modified MNE, MUSIC, and sLORETA algorithms for silence localization are explained in "Methods"). For all modified source localization methods, we chose *Cz* scalp electrode as the reference electrode but, for fair comparison with SilenceMap, also considered the effect of the reference electrode in the modified MNE, MUSIC, and sLORETA (see "Methods"). Based on the simulation results in Table 1, among the modified source localization algorithms, sLORETA shows the minimum average COM distance of 54mm, and MUSIC shows the maximum average overlap of 9% (JI = 0.09), and the minimum average size error of 284% ($\Delta k = 2.84$). This performance is still poor for the silence localization task, while SilenceMap shows good performance based on the simulation results in Table 1 ($\Delta$COM = 12 mm, JI = 0.50, $\Delta k = 0.31$). As evident, source localization algorithms, even after modification, perform poorly in localizing the neural regions of silence.

**Real data**. We also compared the performance of SilenceMap with the modified source localization algorithms, based on a real dataset of patients who have undergone lobectomy surgery, and have a clearly defined resected region in their brain.

*Dataset*. We recorded EEG signals using a BioSemi ActiveTwo system (BioSemi, Amsterdam), with a sampling frequency of 512 Hz, using a 128-electrode cap with electrodes located based on the standard 10-5 system[35]. In addition, we used four electrodes around the eyes, specifically, a pair on the top and bottom of the right eye to detect the vertical eye movements and blinks, and a pair at the outer canthi of each eye to monitor horizontal eye movements. One electrode was placed on the left collar bone to monitor heart beats, and two electrodes were placed on the mastoids. All electrodes were differentially recorded relative to the standard common-mode-sense and driven-right-leg electrodes. We monitored the electrode-gel-scalp contact quality through the

**Table 2 Surgery history of patients[38].**

| Patient | Surgical procedure | Age at resection | Time between resection and MRI scan | Time between resection and EEG recording |
|---------|-------------------|------------------|-------------------------------------|------------------------------------------|
| UD | Right occipital and posterior temporal lobectomy with resection of inferomesial temporal dysembryoplastic neuroepithelial tumor (DNT) | 6 years, 9 months | 4 years, 3 months | 5 years, 10 months |
| SN | Evacuation of left temporal hematoma | 1 day | 12 years, 6 months | 13 years, 1 month |
| OT | Left temporal lobectomy with preservation of mesial structures, gross total resection of left mesial temporal DNT | 13 years, 4 months | 4 years, 3 months | 5 years, 11 months |

data acquisition period using the Electrode Offsets option in the ActiView data acquisition software, which calculates the direct current (DC) potentials generated at the junction of the skin and electrolyte solution (gel) under the electrodes. This DC potential results in a voltage at the amplifier inputs (i.e., DC offset)[36]. Electrodes with larger than 20 mV offset were marked for removal and interpolation in the preprocessing step, and more conductive gel was added to the electrodes with high offset. This is important as the change in electrode impedance can alter the distribution of artifacts (e.g., eye blinks and eye movements) across the scalp and make it harder to detect and remove them using the preprocessing methods[37]. During data acquisition, the participant viewed a screen, located roughly 1 m away. A grating pattern of black and white bars was displayed at the center of the display along with a fixation cross for 2 s, followed by a rest state of 1–1.5 s, where a fixation cross was displayed on a gray-colored background (see Fig. 1a). We repeated this sequence 80 times during the recording session. We used the Rest and Visual sections of the recorded signal separately for the localization and compared the results from these analyses in this section. The steps for data analyses and preprocessing are available in "Methods: Data analysis".

*Participants.* Three male pediatric patients were recruited for this experiment. Two patients (SN and OT) had resections in the left hemisphere and one (UD) had a resection in the right hemisphere. In two of these patients (OT and UD), lobectomy surgery was performed to control pharmacoresistant epilepsy, and, in the third patient (SN), surgery was performed for an emergent evacuation of a cerebral hematoma at day one of life. More information about these patients is included in Table 2.

OT and the parents of SN and UD provided consented for participation. SN and UD provided assent. All procedures were approved by the Carnegie Mellon University Institutional Review Board. The MRI scans of these participants are shown in Fig. 2, where the resected sections can be seen as large asymmetric dark regions[38–40]. The ground truth regions of silence are extracted based on these MRI scans (see "Methods" for more details). The first row in Fig. 3 shows the extracted ground truth regions of silence in the symmetric brain models of the three participants. The intact hemisphere is mirrored across the longitudinal fissure to construct these brain models (see Fig. 1c and "Methods" for more details). These patients have different sizes of regions of silence: UD has a region of silence with $k = 60$ sources/nodes out of total $p = 1740$ sources in the brain, SN has $k = 120$ out of $p = 1758$ sources, and OT has $k = 55$ out of $p = 1744$ total sources. Despite the relatively large sizes of the resected regions in these three pediatric patients, there is rather minimal, if any, observable effect of the resection on performance, indicative of substantial plasticity in the children's brain[41]. This suggests that we cannot characterize the site or size of the resected areas with any precision using standard neuropsychological testing (see Supplementary Table II for neuropsychological test results for these patients, and Supplementary Note B for detailed discussion).

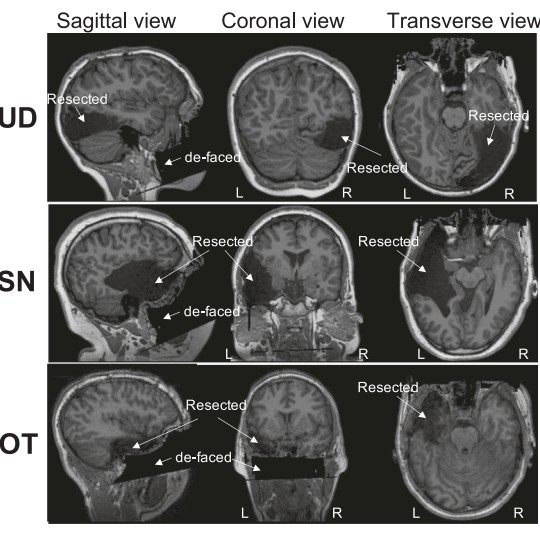

Sagittal view    Coronal view    Transverse view

**Fig. 2 Structural MRI scans of the three participants in the real dataset.** UD with right occipitotemporal lobectomy, SN and OT with left temporal lobectomy (resected regions are highlighted with arrows). We have stripped away facial features to ensure anonymity of participants (de-faced regions are highlighted with arrows).

*Results of real dataset.* We applied SilenceMap, along with the modified source localization algorithms, i.e., MNE, MUSIC, and sLORETA, on the preprocessed EEG recordings of the three participants, and the performance of silence localization is calculated by comparing against the extracted ground truth regions from the post-surgery MRI scans of these patients (see Fig. 2). The visual illustration of localized regions of silence (shown in red on the gray-colored semi-transparent brains), along with their ground truth regions and their corresponding performance measures are all shown in Fig. 3. Based on the Rest dataset, SilenceMap with hemispheric baseline outperforms the modified source localization algorithm: it reduces the COM distance by 12 mm, 46 mm, and 42 mm for UD, SN, and OT respectively, compared to the best performance among the source localization algorithms. It also improves the overlap (JI) by 22%, 49%, and 37%, and the size estimation by 122%, 42%, and 59% for UD, SN, and OT, respectively. SilenceMap with baseline performs well with values of $\Delta COM = 2$ mm, $JI = 0.570$, and $\Delta k = 0.25$ based on the Rest set, and $\Delta COM = 3$ mm, $JI = 0.654$, and $\Delta k = 0.09$ based on the Visual set. Comparing the results of Visual and Rest datasets for SilenceMap with baseline shows that, as expected, the localized regions of silence remain largely the same. This suggests that for each participant, the algorithm can localize the region of silence, regardless of the type of the task performed (Visual or Rest) by the participants during the EEG recording. In SilenceMap with baseline, based on the minimum value of the power mismatch ($\Delta Pow$ defined in Eq. (39) in "Methods"), the best reference electrodes for UD, SN, and OT were $Cz$, $Cz$, and $CPz$,

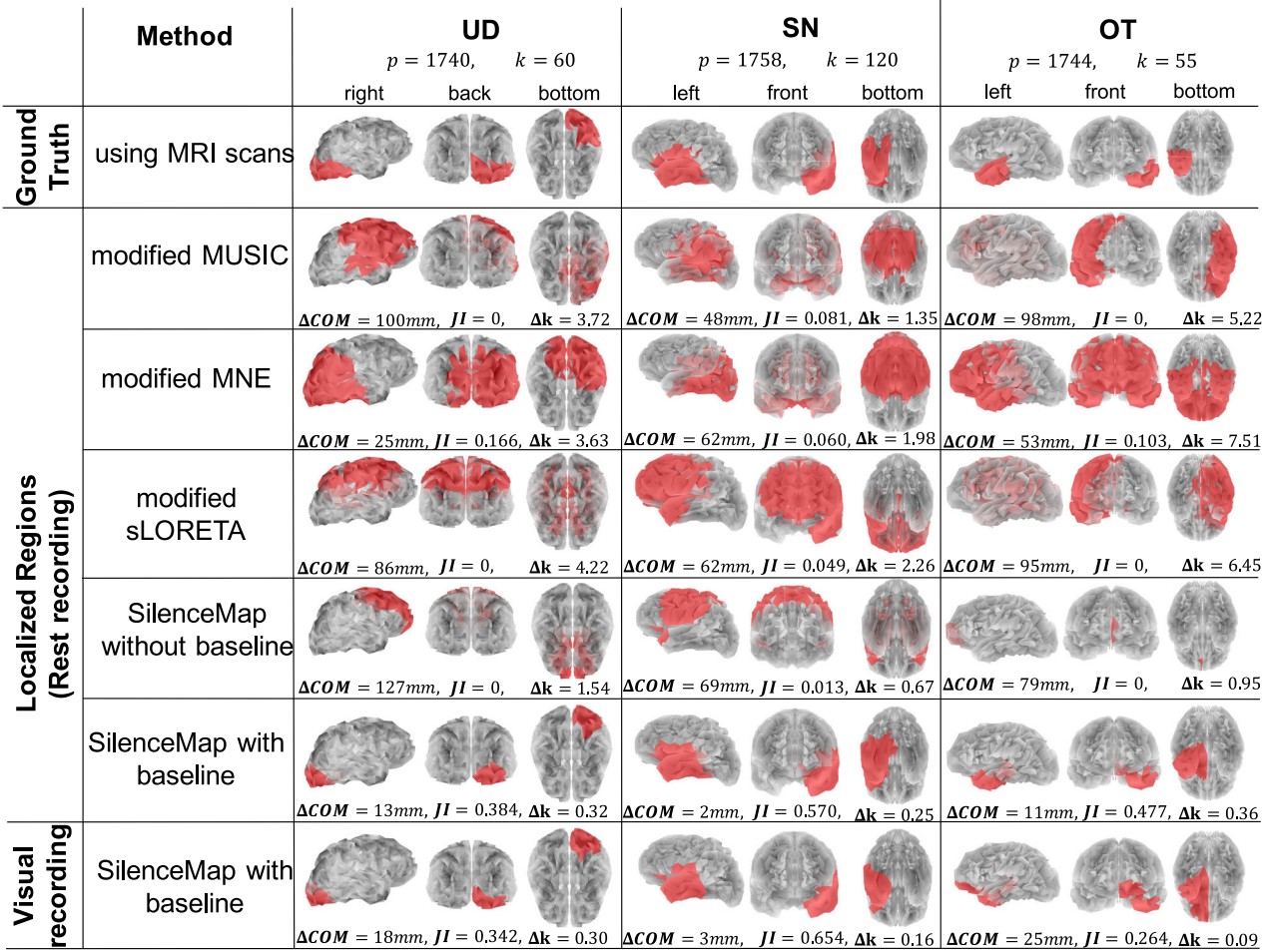

**Fig. 3 Performance of SilenceMap on a real EEG dataset.** The first row shows the extracted ground truth regions of silence (red regions) overlaid on the resected cortical region of three patients based on their symmetric brain models extracted from the structural MRIs (see the MRI scans in Fig. 2); the second, third, and fourth rows show the performance in localization of the silent region using modified source localization algorithms (MNE, MUSIC, and sLORETA), through both visual illustration (red regions) and using performance metrics of center-of-mass (COM) distance (ΔCOM), Jaccard Index (JI), and size error (Δ*k*). The fifth row shows the performance of SilenceMap without baseline, and the last two rows show the localization performance of SilenceMap with baseline, based on the Rest and Visual recordings respectively. *p* is the total number of sources in each brain model, and *k* is the size of ground truth region of silence.

respectively, for the Rest set, and *Cz*, *Pz*, and *CPz* for the Visual set. Based on the results of the Visual set, participant OT shows the poorest localization performance, which might be due to the participant's repetitive jaw clenching during the recording, even after appropriate preprocessing of the data. Jaw clenching is recognized as one of the most severe artifacts adversely impacting the signals of most EEG electrodes[42].

Unlike the simulation results, without baseline, SilenceMap failed to localize the region of silence on the real dataset. One explanation for this is the assumption of the identical distribution of sources in designing the algorithm, which does not hold in the real data. Clearly, using the algorithm with hemispheric baseline is advantageous for better localization.

**Validity of hemispheric symmetry assumption in SilenceMap with baseline.** The hemispheric baseline approach used in SilenceMap is based on an approximate hemispheric symmetry assumption of the brain source activities in the healthy parts of the brain. To further explore the validity of this assumption, we quantified this hemispheric symmetry based on the scalp average power of a neurologically healthy control participant (DH, male, 25 years) whose EEG data were collected using the same protocol used for the patients (see Fig. 1a for the EEG recording protocol).

DH provided informed consent. Excluding the ten electrodes on the longitudinal fissure (red electrodes in Fig. 1a), we calculated the mean absolute difference (MAD) of average power of pairs of scalp electrodes, which are symmetric with respect to the longitudinal fissure, e.g., (*C1*,*C2*), (*T7*,*T8*), and so on, as follows:

$$\text{MAD} = \frac{2}{n-10}\sum_{i=1}^{\frac{n-10}{2}}|\widehat{\text{Var}}(y_i^R) - \widehat{\text{Var}}(y_i^L)| \quad (4)$$

where, $\widehat{\text{Var}}(y_i^R)$ is the estimated variance of the recorded EEG signals referenced to the *Cz* electrode, preprocessed, and denoised signal $y_i^R$ at the electrode *i* in the right hemisphere (see "Methods" for noise removal steps), and $y_i^L$ is the signal of the corresponding electrode in the left hemisphere, and *n* = 128 is the total number of electrodes. Based on Fig. 4, MAD for the control participant is calculated based on the Rest set as 4.1 $(\mu V)^2$, while for the UD, SN, and OT patients with regions of silence MAD is 23.3, 14.2, 16.5 $(\mu V)^2$, respectively. The control participant had a substantially smaller hemispheric difference of scalp power compared to the patients, confirming that using the hemispheric baseline is helpful in localization of regions of silence in either hemisphere. There are two main reasons that the MAD of the healthy control is not perfectly symmetric: (i) brain sources have non-identical

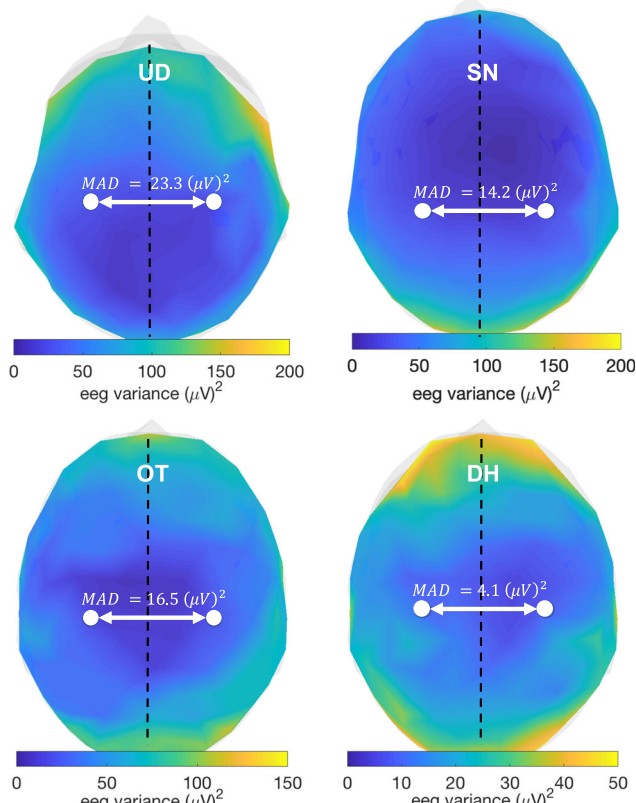

**Fig. 4 Quantification of hemispheric symmetry of scalp average power.** Mean absolute difference of scalp average power (MAD) is reported for a healthy control participant (DH), in comparison to the three patients who have resected brain regions (UD, SN, and OT). The control participant shows substantially smaller MAD compared to the three patients with cortical regions of silence.

brain activities[43,44], and this asymmetry is affected by factors such as age[45], and (ii) the structure of the brain and the head (scalp, skull, cerebrospinal fluid (CSF), and brain) is not perfectly symmetric (see "Discussion" for more discussion), which results in a non-symmetric reflection/transformation of brain activities to the scalp potentials. The latter issue is addressed in SilenceMap, by normalization of the measure of source contribution ($\beta$ in Eq. (35), in "Methods") based on the head structure asymmetry. To improve the performance of SilenceMap, one might take into account the non-identical distribution of sources in the brain (and perhaps use a more realistic model for the source covariance matrix $\mathbf{C}_s$) and normalize the source contribution measure accordingly. Another approach to address this might be to use an asymmetric baseline obtained during the recording of the brain without any region of silence.

**SilenceMap can localize the regions of silence with relatively little EEG data.** As we showed earlier in this section, SilenceMap successfully localized the regions of silence based on only 160s of EEG data. Although this is already quite small, how does SilenceMap perform if we reduce this timespan? To understand this, we did a search for the timespan for [20, 40, 80, 120, 160]s, quantifying the performance for each timespan. For UD, 80 s of data showed almost the same performance as 160s ($\Delta$COM = 17 mm, JI = 0.382, $\Delta k$ = 0.30), while 40 s showed substantial reduction. For SN, the minimum possible amount of data, without compromising the localization performance, is only 40 s ($\Delta$COM = 9 mm, JI = 0.440, $\Delta k$ = 0.20), while for OT, this is

160 s, potentially due to the noisy EEG recording of OT. Nevertheless, the 160 s upper limit is still a relatively short amount of signal acquisition time.

In clinical applications, rapid recording and localization of neural silences might be required. The time-consuming steps for EEG installation—namely, the placement of electrodes and applying conductive gel (~30 min for the high-density EEG we used), electrode impedance monitoring and corrections, and the multistep and offline data preprocessing—may make it difficult to use the system in practice. There has been progress in recent years in developing portable and quick-to-administer EEG systems (e.g., dry, active, low-impedance electrodes, conductive sponge and hydrogel interfaces[46–48]), along with fast and real-time preprocessing and artifact removal techniques (e.g., in ref. [37]). These developments along with SilenceMap (<3 min of EEG recording), and access to sufficient computational power (see "Methods" for computation-complexity analysis of SilenceMap), can enable rapid silence localization.

**Introduced error in silence localization by using symmetric brain models.** Morphological studies of the human brain have shown cortical asymmetry, and how it is affected by factors such as age, sex, and neurological disease[49,50]. Here, we used symmetric brain models of the patients with resection, since the pre-surgery MRI scans of these patients were not available (which may not even have been symmetrical in the first instance). Figure 5 shows the symmetric brain models along with the original models of the three patients. To quantify the introduced error in silence localization by using symmetric models, instead of the original model, we calculated the average distance of sources/nodes of the intact part of the resected hemisphere to the corresponding sources/nodes of the the structurally preserved hemisphere, mirrored across the longitudinal fissure (see Fig. 5). Following the 3D shape matching approach in ref. [51], for a specific source/node in the hemisphere with the region of silence, the corresponding source in the preserved hemisphere is defined as the node with the minimum distance to that specific source. Based on our calculations, the defined average distance between the symmetric brain model and the original brain model is 2.41 ± 0.055 mm, 2.50 ± 0.043 mm, and 2.03 ± 0.044 mm, for UD, SN, and OT, respectively. We excluded the resected parts of the brain in calculating the average distance between the symmetric brain model and the original brain model. To ensure that this average distance is not affected by excluding the resected regions, we also calculated the hemispheric distance in three healthy controls (intact brains) using an open-source MRI database (OASIS-1[52]). The average distance between the symmetric and the original brain model was 2.33 ± 0.012 mm, 2.78 ± 0.016 mm, and 2.35 ± 0.012 mm, for OAS1_0004_MR1 (male, 28 years), OAS1_0005_MR1 (male, 18 years), and OAS1_0034_MR1 (male, 51 years), respectively (see Fig. 5). In fMRI studies, an acceptable motion and voxel displacement, especially in scans of children and adolescents, is typically up to 3 mm[53]. Since the average distance of the symmetric and the original brain models is <3 mm, using the symmetric brain model seems to be a reasonable choice for silence localization.

**Effect of error in the structural segmentation of MRI.** Segmentation of structural MRI scans for patients with brain injuries is complicated[54–56], and standard structural segmentation techniques (e.g., for example, FreeSurfer) may introduce error in silence localization. Standard segmentation methods use anatomical priors extracted from manually or semi-automatically annotated atlases of the healthy brain[56]. However, the anatomy of the damaged brain, especially following severe injury or large

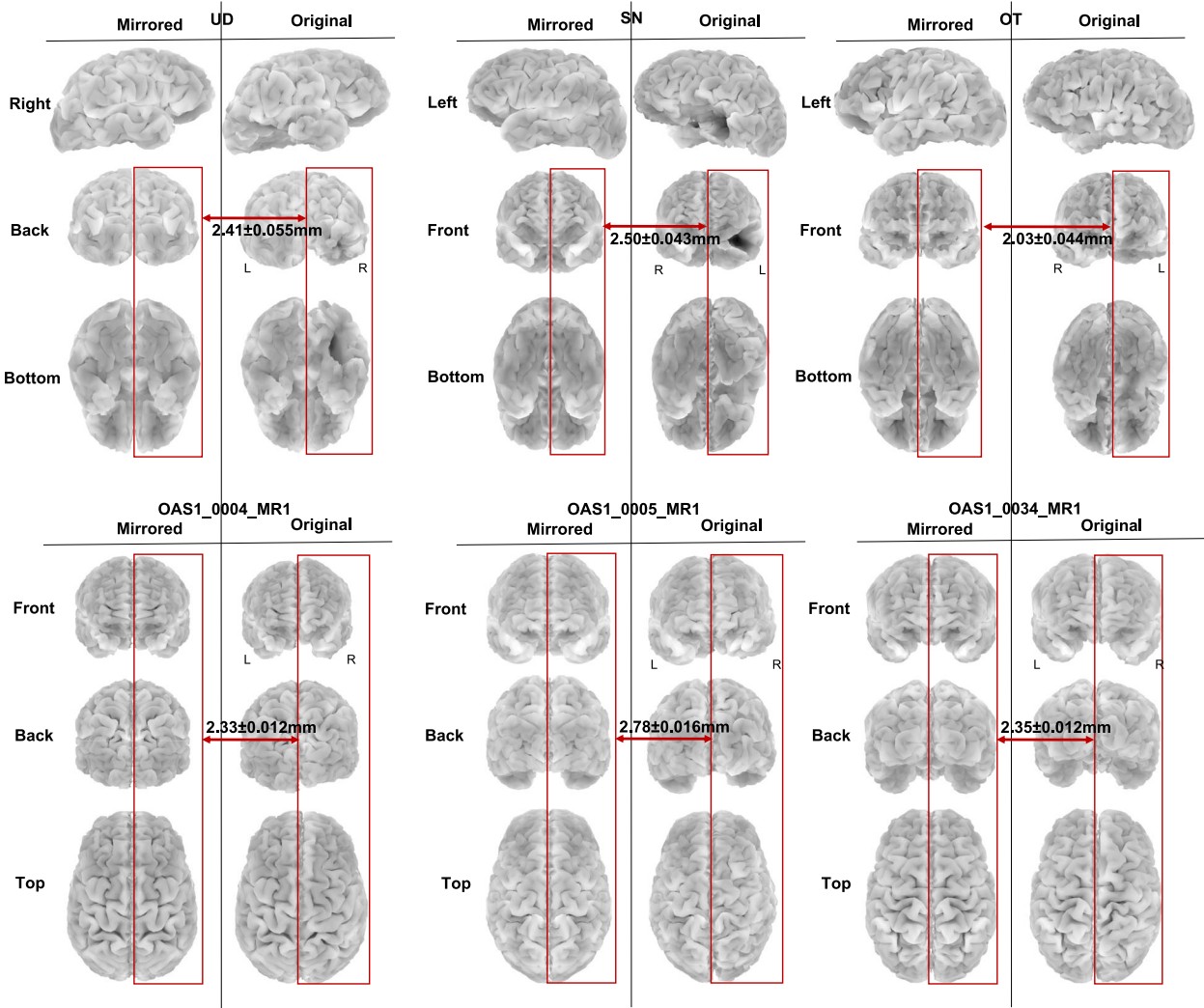

**Fig. 5 Average distance of the symmetric brain model from the original brain model.** The average brain distance is shown in patients with lobectomy, i.e., UD with right occipitotemporal lobectomy, SN with left temporal lobectomy, and OT with left frontotemporal lobectomy, as well as in three healthy controls, namely, OAS1_0004_MR1, OAS1_0005_MR1, and OAS1_0034_MR1. The average distance for all patients and healthy controls are <3 mm, which makes the symmetric brain model a reasonable choice for the silence localization task, as it does not introduce substantial error.

resection, diverges substantially from the anatomy of a healthy brain[54]. To address this, we used an open-source software, AFNI[57,58], which is designed to improve the segmentation of the scans in patients with brain lesions and/or tumors. Additionally, we used specifically designed scripts, provided to us by Dr. Daniel Glen (Computer Engineer, NIH), to segment the structural MRI scans of the patients in our dataset[57,58]. We compared the performance of SilenceMap for the patients in our study with the ground truth regions of silence extracted using AFNI vs. the extracted ground truth using FreeSurfer (Supplementary Fig. 3 contains the results using AFNI). Based on these results, there is a small reduction in the silence localization performance using AFNI in comparison with results from FreeSurfer (Fig. 3 in "Results" and Supplementary Fig. 3 in Supplementary Note C). This suggests that standard techniques (used in FreeSurfer) perform reasonably well in MRI segmentation for our participants, and do not contribute substantially to the silence localization error.

**Effects of brain-to-skull conductivity ratio**. We also explored the effect of different assumptions for the brain-to-skull conductivity ratio on the performance of SilenceMap. Our results (Supplementary Fig. 4 includes results and further discussion is

available in Supplementary Note D) confirm the robustness of our algorithm to imprecise knowledge of brain-to-skull conductivity ratio.

## Discussion

In this paper, we introduced SilenceMap, an algorithm that localizes contiguous regions of silence in the brain based on noninvasive scalp EEG signals. The key technical ideas introduced here include ensuring that background brain activity is separated from silences, using hemispheric baseline, careful referencing, and utilization of a convex optimization framework for clustering. We compared the performance of SilenceMap in stimulations as well as real data comprising structural MRI scans of three patients with cortical resection. SilenceMap substantially outperformed appropriately modified state-of-the-art source localization algorithms, such as MNE, MUSIC, and sLORETA, and reduced the distance error (ΔCOM) by 46 mm, requiring <3 min of EEG signal acquisition time. We also explored potential errors introduced into the algorithmic calculations, e.g., due to our hemispheric baseline assumption, imprecise segmentation of the structural MRI data, or through inaccurate assessment of brain-to-skull conductivity. Our further analyses reveal the

robustness of SilenceMap to these challenges. Altogether, the findings indicate that SilenceMap has considerable potential that permits more general adoption and this EEG-based silence localization method can be used in cases where common imaging modalities such as MRI and computed tomography are not applicable and/or unavailable.

**Limitations and future directions**. As promising as it appears, SilenceMap has its own limitations, which can serve as the focus of future investigations: (i) We used the electrode locations in SilenceMap in multiple steps of the algorithm, including in the estimation of the forward matrix, which is a function of the electrode locations. In placing the EEG cap for each patient, we manually adjust the cap's location so that the electrodes are placed in the standard 10-5 arrangement. This could be improved by using new methods for guided EEG cap placement[59]. (ii) As mentioned in "Methods: Problem statement", SilenceMap assumes that there is one region of silence in the brain, as is the case for the individuals in our real and simulated dataset. One can modify the algorithm to localize multiple regions of silence in the brain, either located in one or both hemispheres. (iii) SilenceMap showed substantial improvement in estimation of size of region of silence, compared to the state-of-the-art modified source localization algorithms. This size estimation in SilenceMap is based on the minimization of scalp average power error, as defined in Eq. (27) in "Methods", which shows an average error of about 30% based on the real dataset in our paper. For applications where more precise estimation of the size of the regions of silence is needed, SilenceMap needs to be improved. (iv) We plan to examine silence localization in individuals with etiologies other than resection. We believe that further improvements and modifications might be needed to use SilenceMap for smaller and deeper regions of silence. (v) We do not consider the effect of the boundary between the intact and the resected brain tissue. In our analysis, the boundary or penumbra is considered to be active and healthy brain tissue. This is a reasonable assumption in surgical removal, and perhaps in other cases where there is a sharp boundary between the affected and the healthy tissue. However, this is not true in diffuse lesions and/or tumors in the brain, and investigation of the boundary effect in these cases requires further consideration. (vi) SilenceMap localizes stationary regions of silence. Extending the algorithm to localize evolving regions of silence, e.g., for CSD propagation, tumor or lesion expansion, will be important. (vii) Finally, there may be changes in the neural functional connectivity because of the region of silence. Estimating these changes are important for, for example, predicting diaschisis (remote effects of a resection) or other, wide-scale changes in signal propagation between regions of cortex.

## Methods

**Notation**. In this paper, we use non-bold letters and symbols (e.g., $a$, $\gamma$, and $\theta$) to denote scalars; lowercase bold letters and symbols (e.g., $\mathbf{a}$, $\boldsymbol{\gamma}$, and $\boldsymbol{\theta}$) to denote vectors; uppercase bold letters and symbols (e.g., $\mathbf{A}$, $\mathbf{E}$, and $\boldsymbol{\Delta}$) to denote matrices, and script fonts (e.g., $\mathcal{S}$) to denote sets.

**Problem statement**. Following the standard approach in the source localization problems, we use the linear approximation of the well-known Poisson's equation to write a linear equation, which relates the neural electrical activities in the brain to the resulting scalp potentials[60,61]. This linear equation is called forward model[62]. In this model, each group of neurons are modeled by a current source or dipole, which is assumed to be oriented normal to the cortical surface[15].

The linear forward model can be written as below:

$$\mathbf{X}_{n \times T} = \mathbf{A}_{n \times p} \mathbf{S}_{p \times T} + \mathbf{E}_{n \times T}, \tag{5}$$

where $\mathbf{A}$ is the forward matrix, $\mathbf{X}$ is the matrix of measurements where each row represents the potentials recorded at one electrode, with reference at infinity, across time. $\mathbf{S}$ is the matrix of source signals, $\mathbf{E}$ is the measurement noise, $T$ is the number of time points, $p$ is the number of sources, and $n$ is the number of scalp sensors.

In practice, we do not have the matrix $\mathbf{X}$, since the reference at infinity cannot be recorded. Only a differential recording of scalp potentials is possible. If we define a $(n-1) \times n$ matrix $\mathbf{M}$ with the last column to be all $-1$ and the first $n-1$ columns compose an identity matrix, the differential scalp signals, with the last electrode's signal as the reference, can be written as follows:

$$\begin{aligned} \mathbf{Y}_{(n-1) \times T} &= \mathbf{M}_{(n-1) \times n} \mathbf{X}_{n \times T} \\ &= \mathbf{M}_{(n-1) \times n} \mathbf{A}_{n \times p} \mathbf{S}_{p \times T} + \mathbf{M}_{(n-1) \times n} \mathbf{E}_{n \times T}, \end{aligned} \tag{6}$$

where $\mathbf{Y}$ is the matrix of differential signals of scalp, $\mathbf{M}$ is a matrix, which transforms the scalp signals with reference at infinity in the matrix $\mathbf{X}$ to the differential signals in $\mathbf{Y}$. Equation (6) can be rewritten as follows:

$$\mathbf{Y}_{(n-1) \times T} = \widetilde{\mathbf{A}}_{(n-1) \times p} \mathbf{S}_{p \times T} + \widetilde{\mathbf{E}}_{(n-1) \times T}, \tag{7}$$

where $\widetilde{\mathbf{A}} = \mathbf{MA}$, and $\widetilde{\mathbf{E}} = \mathbf{ME}$.

*Objective*. Given $\mathbf{M}$, $\mathbf{Y}$, and $\mathbf{A}$, estimate the region of silence in $\mathbf{S}$.

For this objective, we consider two different scenarios: (1) there are no baseline recordings for the region of silence, i.e., no scalp EEG recording is available where there is no region of silence, (2) with baseline recording, i.e., we consider the recording of the hemisphere of the brain, left or right, which does not have any region of silence, as the baseline for the silence localization task. Note that the location of the baseline hemisphere (left or right) is not assumed to be known a priori. Rather, locating the region of silence is the goal of this approach.

We make the following assumptions: (i) $\mathbf{A}$ and $\mathbf{M}$ are known, and $\mathbf{Y}$ has been recorded. (ii) $\widetilde{\mathbf{E}}$ is additive white noise, whose elements are assumed to be independent across space. Thus, at each time point, the covariance matrix is $\mathbf{C}_z$ given by:

$$\begin{aligned} c_{z_{ij}} &= \sigma_{z_i}^2, \quad \text{for all} \quad i = j, \\ c_{z_{ij}} &= 0, \quad\;\; \text{for all} \quad i \neq j, \end{aligned} \tag{8}$$

where $\sigma_{z_i}^2$ is the noise variance at electrode $i$, and it is assumed to be known (to see how this might be estimated see Supplementary Note E and Supplementary Fig. 5). (iii) $k$ rows of $\mathbf{S}$ correspond to the region of silence, which are rows of all zeros. The correlations of source activities reduces as the spatial distance between the sources increases. We assume a spatial exponential decay profile for the source covariance matrix $\mathbf{C}_s$, with identical variances ($\sigma_s^2$) for all non-silent sources, whose signals model the background brain activities:

$$\begin{aligned} c_{s_{ij}} &= \sigma_s^2 e^{-\gamma \|\mathbf{f}_i - \mathbf{f}_j\|_2^2}, \quad \text{for all} \quad i, j \notin \mathcal{S}, \\ c_{s_{ij}} &= 0, \quad\qquad\qquad\;\;\; \text{for all} \quad i, j \in \mathcal{S}. \end{aligned} \tag{9}$$

where $\mathbf{f}_i$ is the 3D location of source $i$ in the brain, $\gamma$ is the exponential decay coefficient, and $\mathcal{S}$ is the set of indices of silent sources ($\mathcal{S} := \{i | s_{it} = 0$ for all $t \in \{1, 2, \cdots T\}\}$). We assume that the elements of $\mathbf{S}$ have zero mean, and follow a wide-sense stationary (WSS) process. (iv) $\mathbf{M}$ is a $(n-1) \times n$ matrix where the last column is $-\mathbb{1}_{n-1 \times 1}$ and the first $n-1$ columns form an identity matrix ($\mathbf{I}_{(n-1) \times (n-1)}$). (v) We assume $p - k \gg k$, where $p - k$ is the number of active, i.e., non-silent sources, and $k$ is the number of silent sources. (vi) Silent sources are contiguous. We define contiguity based on a $z$-nearest neighbor graph, where the nodes are the brain sources (i.e., vertices in the discretized brain model). In this $z$-nearest neighbor graph, two nodes are connected with an edge, if either or both of these nodes is among the $z$-nearest neighbors of the other node, where $z$ is a known parameter (see "Methods" to learn what values of $z$ can be used). A contiguous region is defined as any connected subgraph of the defined nearest neighbor graph, i.e., between each two nodes in the contiguous region, there is at least one connecting path. (vii) For simplicity, we assume that silence lies in only one hemisphere (as is the case for the three individuals examined in the Results). However, the location of this hemisphere is not assumed to be known (see "Methods: SilenceMap with baseline recordings" for the details of SilenceMap with baseline).

*With baseline recordings*. In the absence of baseline recordings, estimating the region of silence proves difficult. In order to exploit prior knowledge about neural activity, we use the approximate symmetry of power of neural activity in the two hemispheres of a healthy brain (see the "Results" for more details on the hemispheric symmetry of scalp potentials, along with examples from the real dataset). (viii) As an additional simplification, we assume that even when there is a region of silence, if the electrode is located far away from the region of silence, then the symmetry still holds. For example, if the silence is in the occipital region, then the power of the signal at the electrodes in the frontal region (after subtracting noise power) is assumed to be symmetric in the two hemispheres (mirror imaged along the longitudinal fissure). This is only an approximation because (a) the brain activity is not completely symmetric, and (b) a silent source affects the signal everywhere, even far from the silent source (see Fig. 4 in the "Results"). Nevertheless, as we will see, this assumption enables more accurate inferences about the location of the silence region in real data using SilenceMap with baseline, in comparison to SilenceMap without baseline. The simplification assumptions in this

**Table 3 List of simplification assumptions and their effects on silence localization.**

| Assumption number in "Methods: Problem statement" | Assumption | Effect | Possible ways to relax these assumptions |
|---|---|---|---|
| (ii) | Spatio-temporal independence of additive noise $\widetilde{\mathbf{E}}$ | It affects the noise variance estimation (see Supplementary Note E) | Using more realistic assumptions on the general shape of noise PSD (non-flat PSD), and the spatial correlation profile (non-diagonal $\mathbf{C}_z$), noise variance estimation can be improved. |
| (iii) | Spatial exponential decay profile for the source covariance matrix $\mathbf{C}_s$, with identical variances ($\sigma_s^2$) for all non-silent sources | It affects the source covariance estimation in SilenceMap (see Eq. (34)) | Using more realistic and data-driven assumptions on the spatial correlation profile of brain sources, as well as estimation of non-identical source variances based on baseline recordings of silences. |
| (vi) | Contiguity of silent sources as a single region of silence | It affects the design of the CSpeC framework proposed in SilenceMap (see Eqs. (36) and (18)) | With the assumption of multiple regions of silence, with different sizes, using methods such as the extension of CSpeC method for multiple clusters in a graph can be used in SilenceMap[64]. |
| (vii) | Silence lies in only one hemisphere | Based on this assumption, we use the hemispheric baseline for silence localization. | This assumption can be relaxed if we have a baseline recording for the regions of silence (e.g., recording of the brain without any silence). |
| (vii) | Hemispheric symmetry of scalp potentials for regions far from silence | Based on this assumption, we use hemispheric baseline and select a subset of scalp electrodes to estimate the source covariance matrix (see Eqs. (32) and (33)). | This assumption can be relaxed if we have a baseline recording for the regions of silence (e.g., recording of the brain without any silence), and use a non-identical distribution model for the non-silent source activities (see assumption (iii) and its relaxation). |

section are summarized in Table 3, where we discuss the effect of each assumption, along with possible ways to relax them.

We first provide the details of this two-step algorithm under the condition where we do not have any baseline, and then with a hemispheric baseline.

**SilenceMap without baseline recordings.** If we do not have any baseline recording, we design the two-step silence localization algorithm as follows:

*Low-resolution grid and uncorrelated sources.* For the iterative method in the second step, we need an initial estimate of the region of silence to select the electrodes whose powers are affected the least by the region of silence. We coarsely discretize the cortex to create a very low-resolution source grid with sources that are located far enough from each other, so that the elements of $\mathbf{S}$ can be assumed to be uncorrelated across space:

$$c_{s_{ij}} = \sigma_s^2, \quad \text{for all} \quad i = j \ \& \ i, j \notin \mathcal{S},$$
$$c_{s_{ij}} = 0, \quad \text{for all} \quad i \neq j \ or \ i, j \in \mathcal{S}. \quad (10)$$

Under this assumption of uncorrelatedness and identical distribution of brain sources in this low-resolution grid, we find a contiguous region of silence through the following steps:

(i) Cross-correlation: Eq. (7) can be written in the form of linear combination of columns of matrix $\widetilde{\mathbf{A}}$ as follows:

$$\mathbf{y}_t = \sum_{i=1}^{p} \widetilde{\mathbf{a}}_i s_{it} + \widetilde{\boldsymbol{\epsilon}}_t, \quad \text{for} \ t = \{1, 2, \cdots T\}, \quad (11)$$

where $s_{it}$ is the $i^{th}$ element of the $t^{th}$ column in $\mathbf{S}$, $\mathbf{Y} = [\mathbf{y}_1, \cdots, \mathbf{y}_T] \in \mathbb{R}^{(n-1) \times T}$, $\mathbf{S} = [\mathbf{s}_1, \cdots, \mathbf{s}_T] \in \mathbb{R}^{p \times T}$, $\widetilde{\mathbf{A}} = [\widetilde{\mathbf{a}}_1, \cdots, \widetilde{\mathbf{a}}_p] \in \mathbb{R}^{(n-1) \times p}$, and $\widetilde{\mathbf{E}} = [\widetilde{\boldsymbol{\epsilon}}_1, \cdots, \widetilde{\boldsymbol{\epsilon}}_T] \in \mathbb{R}^{(n-1) \times T}$.

Based on Eq. (11), each column vector of differential signals, i.e., $\mathbf{y}_t$, is a weighted linear combination of columns of matrix $\widetilde{\mathbf{A}}$, with weights equal to the corresponding source values. However, in the presence of silences, the columns of $\widetilde{\mathbf{A}}$ corresponding to the silent sources do not contribute to this linear combination. Therefore, we calculate the cross-correlation coefficient $\mu_{qt}$, which is a measure of the contribution of the $q^{th}$ brain source to the measurement vector $\mathbf{y}_t$ (across all electrodes) at the $t^{th}$ time-instant, defined as follows:

$$\mu_{qt} = \widetilde{\mathbf{a}}_q^T \mathbf{y}_t = \sum_{i=1}^{p} \widetilde{\mathbf{a}}_q^T \widetilde{\mathbf{a}}_i s_{it} + \widetilde{\mathbf{a}}_q^T \widetilde{\boldsymbol{\epsilon}}_t, \quad \text{for all} \ q = \{1, 2, \cdots p\},$$
$$\text{for all} \ t = \{1, 2, \cdots T\}. \quad (12)$$

This measure is imperfect because the columns of the $\widetilde{\mathbf{A}}$ matrix are not orthogonal. The goal here is to attempt to quantify relative contributions of all sources to the recorded signals, and use that to arrive at a decision on which sources are silent because their contribution is zero.

(ii) Estimation of variance of $\mu_{qt}$: In this step, we estimate the variances of the correlation coefficients calculated in the step (i). Based on Eq. (12) we have:

$$\text{Var}(\mu_{qt}) = \text{Var}\left(\sum_{i=1}^{p} \widetilde{\mathbf{a}}_q^T \widetilde{\mathbf{a}}_i s_{it} + \widetilde{\mathbf{a}}_q^T \widetilde{\boldsymbol{\epsilon}}_t\right)$$
$$\overset{(a)}{=} \text{Var}\left(\sum_{i=1}^{p} \widetilde{\mathbf{a}}_q^T \widetilde{\mathbf{a}}_i s_{it}\right) + \text{Var}(\widetilde{\mathbf{a}}_q^T \widetilde{\boldsymbol{\epsilon}}_t)$$
$$\overset{(b)}{=} \sum_{i=1}^{p} \text{Var}(\widetilde{\mathbf{a}}_q^T \widetilde{\mathbf{a}}_i s_{it}) + \text{Var}(\widetilde{\mathbf{a}}_q^T \widetilde{\boldsymbol{\epsilon}}_t)$$
$$\overset{(c)}{=} \sum_{\substack{i=1 \\ i \notin \mathcal{S}}}^{p} (\widetilde{\mathbf{a}}_q^T \widetilde{\mathbf{a}}_i)^2 \sigma_s^2 + \widetilde{\mathbf{a}}_q^T \mathbf{C}_z \widetilde{\mathbf{a}}_q, \quad (13)$$

where $\mathcal{S}$ is the indices of silent sources. In (13), the equality (a) holds because of independence of noise and sources, and the assumption that they have zero mean, (b) holds because the elements of $\mathbf{S}$, i.e., $s_{it}$'s, are assumed to be uncorrelated and have zero mean in the low-resolution grid, and (c) holds because $s_{it}$'s are assumed to be identically distributed. It is important to note that $\sigma_s^2$ in (13) is a function of source grid discretization and it does not have the same value in the low-resolution and high-resolution grids. We estimate the variance of $\mu_{qt}$ using its power spectral density, as is explained in detail in the Supplementary Note F.

Based on Eq. (13), the variance of $\mu_{qt}$, excluding the noise variance, can be written as follows:

$$\widetilde{\text{Var}}(\mu_{qt}) = \text{Var}(\mu_{qt}) - \widetilde{\mathbf{a}}_q^T \mathbf{C}_z \widetilde{\mathbf{a}}_q = \sum_{\substack{i=1 \\ i \notin \mathcal{S}}}^{p} (\widetilde{\mathbf{a}}_q^T \widetilde{\mathbf{a}}_i)^2 \sigma_s^2 \quad (14)$$

where $\widetilde{\text{Var}}(\mu_{qt})$ is the variance of $\mu_{qt}$ without the measurement noise term, which is a function of the size and location of region of silence through the indices in $\mathcal{S}$.

However, this variance term, as is, cannot be used to detect the silent sources, since some sources may be deep, and/or oriented in a way that they have weaker representation in the recorded signal $\mathbf{y}_i$, and consequently have smaller $\mathrm{Var}(\mu_{qt})$ and $\widetilde{\mathrm{Var}}(\mu_{qt})$.

(iii) Source contribution measure ($\beta$): To be able to detect the silent sources and distinguish them from sources, which inherently have different values of $\widetilde{\mathrm{Var}}(\mu_{qt})$, we need to normalize this variance term for each source by its maximum possible value, i.e., when there is no silent source ($\widetilde{\mathrm{Var}}^{\max}(\mu_{qt}) = \sum_{i=1}^{p} (\widetilde{\mathbf{a}}_q^T \widetilde{\mathbf{a}}_i)^2 \sigma_s^2$):

$$\overline{\mathrm{Var}(\mu_{qt})} = \frac{\widetilde{\mathrm{Var}}(\mu_{qt})}{\widetilde{\mathrm{Var}}^{\max}(\mu_{qt})} = \frac{\widetilde{\mathrm{Var}}(\mu_{qt})}{\sum_{i=1}^{p} (\widetilde{\mathbf{a}}_q^T \widetilde{\mathbf{a}}_i)^2 \sigma_s^2}, \quad (15)$$

where $\overline{\mathrm{Var}(\mu_{qt})}$ is the normalized variance of $\mu_{qt}$, without noise, and it takes values between 0 (all sources silent) and 1 (no silent source). Note that it does not only depend on whether $q \in \mathcal{S}$, where $\mathcal{S}$ is the set of indices of silent sources. In general, this normalization requires estimation of source variance $\sigma_s^2$, but under the assumption that sources have identical distribution, they all have identical variances. Therefore, $\sigma_s^2$ in the denominator of Eq. (15) is the same for all sources. We multiply both sides of Eq. (15) by $\sigma_s^2$ and obtain:

$$\sigma_s^2 \overline{\mathrm{Var}(\mu_{qt})} = \frac{\widetilde{\mathrm{Var}}(\mu_{qt})}{\sum_{i=1}^{p} (\widetilde{\mathbf{a}}_q^T \widetilde{\mathbf{a}}_i)^2} = \frac{\mathrm{Var}(\mu_{qt}) - \widetilde{\mathbf{a}}_q^T \mathbf{C}_z \widetilde{\mathbf{a}}_q}{\sum_{i=1}^{p} (\widetilde{\mathbf{a}}_q^T \widetilde{\mathbf{a}}_i)^2}, \quad (16)$$

Therefore,

$$\begin{aligned} \beta_q &:= \sigma_s^2 \overline{\mathrm{Var}(\mu_{qt})} = \frac{\mathrm{Var}(\mu_{qt}) - \widetilde{\mathbf{a}}_q^T \mathbf{C}_z \widetilde{\mathbf{a}}_q}{\sum_{j=1}^{p} (\mathbf{a}_q^T \mathbf{a}_j)^2} \\ &\approx \frac{\widehat{\mathrm{Var}}(\mu_{qt}) - \widetilde{\mathbf{a}}_q^T \widehat{\mathbf{C}}_z \widetilde{\mathbf{a}}_q}{\sum_{j=1}^{p} (\widetilde{\mathbf{a}}_q^T \widetilde{\mathbf{a}}_j)^2}, \end{aligned} \quad (17)$$

where $\beta_q$ is called the contribution of the $q^{th}$ source in the differential scalp signals in $\mathbf{Y}$, which takes values between 0 (all sources silent) and $\sigma_s^2$ (no silent sources). In Eq. (17), $\widehat{\mathrm{Var}}(\mu_{qt})$ is an estimate of variance of $\mu_{qt}$, and $\widehat{\mathbf{C}}_z$ is an estimate of noise covariance matrix (see Supplementary Note E and Supplementary Note F to see how these estimates might be obtained).

(iv) Localization of silent sources in the low-resolution grid: In this step, we find the silent sources based on the $\beta_q$ values defined in the previous step, through a convex spectral clustering (CSpeC) framework as follows:

$$\begin{aligned} \mathbf{g}^\star(\lambda, k) = \arg\min_{\mathbf{g}} \quad & \boldsymbol{\beta}^T (\mathbb{1} - \mathbf{g}) + \lambda (\mathbb{1} - \mathbf{g})^T \mathbf{L} (\mathbb{1} - \mathbf{g}), \\ s.t. \quad & g_i \in [0, 1], \quad \text{for all } i \in \{1, 2, \cdots p\} \\ & \|\mathbf{g}\|_1 \leq p - k. \end{aligned} \quad (18)$$

where $\boldsymbol{\beta}^T = [\beta_1, \cdots, \beta_p]$ is the vector of source contribution measures, $\mathbf{g} = [g_1, \cdots, g_p]^T$ is a relaxed indicator vector with values between 0 (for silent sources) and 1 (for active sources), $k$ is the number of silent sources, i.e., the size of the region of silence, $\lambda$ is a regularization parameter, and $\mathbf{L}$ is a graph Laplacian matrix defined in Eq. (23) below. Based on the linear term in the cost function of Eq. (18), the optimizer finds the solution $\mathbf{g}^\star$ that (ideally) has small values for the silent sources, and large values for the non-silent sources. The $\ell 1$ norm convex constraint controls the size of region of silence in the solution. To make the localized region of silence contiguous, we have to penalize the sources, which are located far from each other. This is done using the quadratic term in the cost function in Eq. (18) and through a graph spectral clustering approach, namely, relaxed RatioCut partitioning[25–27]. We define a z-nearest neighbor undirected graph with the nodes to be the locations of the brain sources (i.e., vertices in the discretized brain model), and a weight matrix $\mathbf{W}$ defined as follows:

$$\begin{aligned} w_{ij} &= e^{-\frac{\|\mathbf{f}_i - \mathbf{f}_j\|_2^2}{2\theta^2}}, \text{ for all } i \in z\text{-nearest neighbor of } j \\ & \qquad \qquad OR \; j \in z\text{-nearest neighbor of } i, \\ w_{ij} &= 0, \qquad \text{for all } i \notin z\text{-nearest neighbor of } j \\ & \qquad \qquad AND \; j \notin z\text{-nearest neighbor of } i, \end{aligned} \quad (19)$$

where the link weight is zero (no link) between node $i$ and $j$, if node $i$ is not among the z-nearest neighbors of $j$, and node $j$ is not among the z-nearest neighbors of $i$. In Eq. (19), we choose z to be equal to the number of silent sources, i.e., $k$, and $\theta$ is an exponential decay constant, which normalizes the distances of sources from each other in a discretized brain model, by their variance as follows:

$$\theta^2 = \mathrm{Var}(\|\mathbf{f}_i - \mathbf{f}_j\|_2) \approx \frac{1}{N-1} \sum_{i=1}^{P} \sum_{j=i+1}^{P} (\|\mathbf{f}_i - \mathbf{f}_j\|_2 - \overline{\delta \mathbf{f}})^2, \quad (20)$$

where $N = \frac{p(p-1)}{2}$ is the total number of inter-source distances, and $\overline{\delta \mathbf{f}}$ is an estimated average of these inter-source distances, given by:

$$\overline{\delta \mathbf{f}} = \frac{1}{N-1} \sum_{i=1}^{P} \sum_{j=i+1}^{P} \|\mathbf{f}_i - \mathbf{f}_j\|_2, \quad (21)$$

The degree matrix of the graph ($\mathbf{D}$) is given by:

$$\mathbf{D} = \left\{ [d_{ij}] | d_{ij} = \sum_{l=1}^{p} w_{il}, \qquad \text{for all } i = j, \text{ and} \right.$$
$$\left. d_{ij} = 0, \text{ for all } i \neq j \right\} \quad (22)$$

Using the degree and weight matrices defined in Eqs. (19) and (22), the graph Laplacian matrix, $\mathbf{L}$ in Eq. (18), is defined as follows:

$$\mathbf{L} = \mathbf{D} - \mathbf{W} \quad (23)$$

Based on one of the properties of the graph Laplacian matrix[63], we can write the quadratic term in the objective function of Eq. (18) as follows:

$$(\mathbb{1} - \mathbf{g})^T \mathbf{L} (\mathbb{1} - \mathbf{g}) = \frac{1}{2} \sum_{i,j=1}^{p} w_{ij} (g_i - g_j)^2. \quad (24)$$

where $\mathbf{g} \in \mathbb{R}^p$. This quadratic term promotes the contiguity in the localized region of silence, e.g., an isolated node in the region of silence, which is surrounded by a number of active sources in the nearest neighbor graph, causes a large value in the quadratic term in Eq. (24), since $w_{ij}$ has large value due to the contiguity, and the difference $(g_i - g_j)$ has large value, since it is evaluated between pairs of silent (small $g_i$)-active (large $g_j$) sources.

For a given $k$, the regularization parameter $\lambda$ in Eq. (18), is found through a grid-search and the optimal value ($\lambda^\star$) is found as the one which minimizes the total normalized error of source contribution and the contiguity term as follows:

$$\begin{aligned} \lambda^\star(k) = \arg\min_{\lambda} \quad & \frac{(\boldsymbol{\beta}^T (\mathbb{1} - \mathbf{g}^\star(\lambda, k)))^2}{\max_{\lambda_1} \; (\boldsymbol{\beta}^T (\mathbb{1} - \mathbf{g}^\star(\lambda_1, k)))^2} \\ & + \frac{((\mathbb{1} - \mathbf{g}^\star(\lambda, k))^T \mathbf{L} (\mathbb{1} - \mathbf{g}^\star(\lambda, k)))^2}{\max_{\lambda_2} \; ((\mathbb{1} - \mathbf{g}^\star(\lambda_2, k))^T \mathbf{L} (\mathbb{1} - \mathbf{g}^\star(\lambda_2, k)))^2}. \end{aligned} \quad (25)$$

In addition, the size of region of silence, i.e., $k$, is estimated through a grid-search as follows:

$$\hat{k} = \arg\min_{k} \sum_{i=1}^{n-1} \left\| (\widetilde{\mathbf{A}} \mathbf{C}_s(k) \widetilde{\mathbf{A}}^T)_{ii} + \widehat{\sigma}_{z_i}^2 - \widehat{\mathrm{Var}}(y_i) \right\|_2^2, \quad (26)$$

where $(.)_{ii}$ indicates the element of a matrix at the intersection of the $i^{th}$ row and the $i^{th}$ column, $\widehat{\mathrm{Var}}(y_i)$ is the estimated variance of the $i^{th}$ differential signal in $\mathbf{Y}$, and $\widehat{\sigma}_{z_i}^2$ is the estimated noise variance at the $i^{th}$ electrode location (see Supplementary Note E and Supplementary Note G to see how these might be estimated). In Eq. (26), $\mathbf{C}_s(k)$ is the source covariance matrix, when there are $k$ silent sources in the brain. The estimate $\hat{k}$ minimizes the cost function in Eq. (26), which is the squared error of difference between the powers of scalp differential signals, resulting from the region of silence with size $k$, and the estimated scalp powers based on the recorded data, with the measurement noise power removed. Under the assumption of identical distribution of sources, and lack of spatial correlation in the low-resolution source grid, and based on Eq. (10), we can rewrite Eq. (26) as follows:

$$\begin{aligned} \hat{k} &= \arg\min_{k} \sum_{i=1}^{n-1} \left\| \sum_{\substack{j=1 \\ j \notin \mathcal{S}}}^{p} \widetilde{a}_{ij}^2 \sigma_s^2 + \widehat{\sigma}_{z_i}^2 - \widehat{\mathrm{Var}}(y_i) \right\|_2^2 \\ &= \arg\min_{k} \sum_{i=1}^{n-1} \left\| \frac{\sum_{\substack{j=1 \\ j \notin \mathcal{S}}}^{p} \widetilde{a}_{ij}^2}{\max_l \; \sum_{\substack{j=1 \\ j \notin \mathcal{S}}}^{p} \widetilde{a}_{ij}^2} - \frac{\widehat{\mathrm{Var}}(y_i) - \widehat{\sigma}_{z_i}^2}{\max_m \; (\widehat{\mathrm{Var}}(y_m) - \widehat{\sigma}_{z_m}^2)} \right\|_2^2 \end{aligned} \quad (27)$$

where $\widetilde{a}_{ij}$ is the element of matrix $\widetilde{\mathbf{A}}$ at the intersection of the $i^{th}$ row and the $j^{th}$ column, and $\mathcal{S}$ is the set of indices of $k$ silent sources, i.e., indices of sources corresponding to the $k$ smallest values in $\mathbf{g}^\star(\lambda^\star, k)$, which is the solution of Eq. (18). The second equality in Eq. (27) normalizes the power of electrode $i$ using the maximum power of scalp signals for each $i$. This step eliminates the need to estimate $\sigma_s$ in the low-resolution.

Finally, the region of silence is estimated as the sources corresponding to the $\hat{k}$ smallest values in $\mathbf{g}^\star(\lambda^\star, k)$. The 3D coordinates of the center-of-mass (COM) of the estimated contiguous region of silence in the low-resolution grid, i.e., $\mathbf{f}_{\mathrm{COM}}^{\mathrm{low}}$, is used as an initial guess for the silence localization in the high-resolution grid, as explained in the next step.

*Iterative algorithm based on a high-resolution grid and correlated sources.* In this step, we use a high-resolution source grid, where the sources are not uncorrelated anymore. We try to estimate the source covariance matrix $\mathbf{C}_s$ based on the spatial

exponential decay assumption in (9). In each iteration, based on the estimated source covariance matrix, the region of silence is localized using a CSpeC framework.

(i) Initialization: In this step, we initialize the source variance $\sigma_s^2$, the exponential decay coefficient in the source covariance matrix $\gamma$, and the set of indices of silent sources $\mathcal{S}$ as follows:

$$\gamma^{(0)} = 1, \ \sigma_s^{2^{(0)}} = 1,$$
$$\mathcal{S}^{(0)} = \left\{ i \middle| \ \| \mathbf{f}_i - \mathbf{f}_{COM}^{low} \|_2^2 \leq \| \mathbf{f}_j - \mathbf{f}_{COM}^{low} \|_2^2, \right.$$
$$\left. \text{for all} \ \ j = 1, 2, \cdots p \right\}. \tag{28}$$

where $\mathcal{S}^{(0)}$ is simply the index of nearest source in the high-resolution grid to the COM of the localized region of silence in the low-resolution grid, i.e., $\mathbf{f}_{COM}^{low}$.

For $r = 1, 2, \cdots R$, we repeat the following steps until either the maximum number of iterations ($R$) is reached, or COM of the estimated region of silence $\mathbf{f}_{COM}^{high^{(r)}}$ has converged, where $\mathbf{f}_{COM}^{high^{(0)}}$ is the location of the source with index $\mathcal{S}^{(0)}$ in the high-resolution grid. The convergence criterion is defined as below:

$$\| \mathbf{f}_{i,COM}^{high^{(j)}} - \mathbf{f}_{i,COM}^{high^{(j-1)}} \|_2 \leq \delta, \ \ \text{for} \ j \in \{r-1, r\}, \ r \geq 2. \tag{29}$$

where $\delta$ is a convergence parameter for COM displacement through iterations, and $\mathbf{f}_{COM}^{high} \in \mathbb{R}^{3 \times 1}$.

(ii) Estimation of $\sigma_s^2$ and $\gamma$: In this step, we estimate the source variance $\sigma_s^2$, and the exponential decay coefficient of source covariance matrix $\gamma$, based on their values in the previous iteration and the indices of silent sources in $\mathcal{S}^{(r-1)}$. We define $\mathbf{C}_s^{full}$ as the source covariance matrix when there are no silent sources in the brain, and use it to measure the effect of region of silence on the power of each electrode. The source covariance matrix in the previous iteration ($r-1$) is calculated as follows:

$$\mathbf{C}_s^{(r-1)} = \left\{ [c_{s_{ij}}] | c_{s_{ij}} = \sigma_s^{2^{(r-1)}} e^{-\gamma^{(r-1)} \| \mathbf{f}_i - \mathbf{f}_j \|^2}, \right.$$
$$\left. \text{for all} \ i, j \notin \mathcal{S}^{(r-1)}, \ and \ c_{s_{ij}} = 0, \ \text{if} \ i \ or \ j \ \in \mathcal{S}^{(r-1)} \right\}, \tag{30}$$

and $\mathbf{C}_s^{full}$ is given by:

$$\mathbf{C}_s^{full^{(r-1)}} := \left\{ [c_{s_{ij}}] | c_{s_{ij}} = \sigma_s^{2^{(r-1)}} e^{-\gamma^{(r-1)} \| \mathbf{f}_i - \mathbf{f}_j \|^2}, \right.$$
$$\left. \text{for all} \ i, j = 1, 2, \cdots p \right\}, \tag{31}$$

where there is no zero row and/or column, i.e., there is no silence. To be able to estimate $\sigma_s^2$ and $\gamma$ based on the differentially recorded signals in $\mathbf{Y}$, we need to find the electrodes, which are the least affected by the region of silence. Based on the assumption (v) in "Methods: Problem statement", the region of silence is much smaller than the non-silent brain region and some electrodes can be found on scalp, which are not substantially affected by the region of silence. We find these electrodes by calculating a power-ratio for each electrode, i.e., the power of electrode when there is a silent region, divided by the power of electrode when there is not any region of silent in the brain, as follows:

$$\mathbf{h}^{(r)} = \left\{ [h_i] | h_i = \frac{(\widetilde{\mathbf{A}} \mathbf{C}_s^{(r-1)} \widetilde{\mathbf{A}}^T)_{ii}}{(\widetilde{\mathbf{A}} \mathbf{C}_s^{full^{(r-1)}} \widetilde{\mathbf{A}}^T)_{ii}}, \ \ \text{for all} \ i = 1, 2, \cdots n-1 \right\}, \tag{32}$$

where $\mathbf{h}$ is a vector with values between 0 (all sources silent) and 1 (no silent source). Using this power-ratio, we select the electrodes as follows:

$$\mathcal{S}_{elec}^{(r)} = \{ i | \text{indices of the} \ \phi \ \text{maximum values in} \ \mathbf{h}^{(r)} \}, \tag{33}$$

where $\mathcal{S}_{elec}$ is the indices of the top $\phi$ electrodes, which have the least power reduction due to the silent sources in $\mathcal{S}$. Based on the differential signals of the selected $\phi$ electrodes in Eq. (33), $\gamma^{(r)}$ and $\sigma_s^{(r)}$ are estimated as the least-square solutions in the following equation:

$$(\gamma^{(r)}, \sigma_s^{(r)}) = \arg \min_{\gamma, \sigma_s} \sum_{i \in \mathcal{S}_{elec}^{(r)}} \left\| (\widetilde{\mathbf{A}} \mathbf{C}_s^{full}(\gamma, \sigma_s) \widetilde{\mathbf{A}}^T)_{ii} \right.$$
$$\left. + \widehat{\sigma}_{z_i}^2 - \widehat{Var}(y_{it}) \right\|_2^2. \tag{34}$$

(iii) Localization of silent sources in the high-resolution grid: Based on the correlatedness assumption of sources in the high-resolution grid, we modify the source contribution measure definition (from Eq. (17)) as follows:

$$\beta_q^{high^{(r)}} := \frac{Var(\mu_{qt}) - \widetilde{\mathbf{a}}_q^T \mathbf{C}_z \widetilde{\mathbf{a}}_q}{\widetilde{\mathbf{a}}_q^T (\widetilde{\mathbf{A}} \mathbf{C}_s^{full} \widetilde{\mathbf{A}}^T) \widetilde{\mathbf{a}}_q} \approx \frac{\widetilde{Var}(\mu_{qt}) - \widetilde{\mathbf{a}}_q^T \widetilde{\mathbf{C}_z} \widetilde{\mathbf{a}}_q}{\widetilde{\mathbf{a}}_q^T (\widetilde{\mathbf{A}} \mathbf{C}_s^{full^{(r)}} \widetilde{\mathbf{A}}^T) \widetilde{\mathbf{a}}_q}, \tag{35}$$

where $\beta_q^{high^{(r)}}$ takes values between 0 (all sources silent), and 1 (no silent source in the brain). The only difference between $\beta_q^{high^{(r)}}$ in the high-resolution grid and $\beta_q$ in the low-resolution grid is in their denominators, which are essentially the variance

terms in the absence of any silent source ($\widetilde{Var}^{max}(\mu_{qt})$ in Eq. (15)). In $\beta_q$, the denominator is divided by the source variance $\sigma_s^2$, to be able to calculate $\beta_q$ without estimation of $\sigma_s^2$. However, in the high-resolution grid, the denominator of $\beta_q^{high^{(r)}}$ is simply $\widetilde{Var}^{max}(\mu_{qt})$, which is calculated under the source correlatedness assumption and using the estimated $\mathbf{C}_s^{full^{(r)}}$. Using the definition of source contribution measure $\beta_q^{high^{(r)}}$ in the high-resolution grid, at iteration $r$, the contiguous region of silence is localized through a CSpeC framework, similar to the one defined in Eq. (18). However, we use the estimated source covariance matrix in each iteration to introduce a new set of constraints on the powers of the electrodes, which are less affected by the region of silence, i.e., the electrodes in $\mathcal{S}_{elec}^{(r)}$, as defined in Eq. (33). Based on these power constraints, we obtain a convex optimization framework to localize the region of silence in the high-resolution brain model as follows:

$$\mathbf{g}^{\star(r)}(\lambda, k, \zeta) = \arg \min_{\mathbf{g}} \ \ \boldsymbol{\beta}^{high^{(r)^T}} (\mathbb{1} - \mathbf{g}) + \lambda(\mathbb{1} - \mathbf{g})^T \mathbf{L}(\mathbb{1} - \mathbf{g}),$$
$$s.t. \ \ g_i \in [0, 1], \ \ \text{for all} \ \ i \in \{1, 2, \cdots p\}$$
$$\| \mathbf{g} \|_1 \leq p - k, \tag{36}$$
$$(\mathbb{1}^T (\bar{\mathbf{A}}_i \mathbf{C}_s^{full^{(r)}} \bar{\mathbf{A}}_i^T) \mathbf{g} + \widehat{\sigma}_{z_i}^2 - \widehat{Var}(y_i))^2 \leq \zeta_i,$$
$$\text{for all} \ i \in \mathcal{S}_{elec}^{(r)}.$$

where $\boldsymbol{\beta}^{high^{(r)^T}} = [\beta_1^{high^{(r)}}, \cdots, \beta_p^{high^{(r)}}]$, $\mathbf{g} = [g_1, \cdots, g_p]^T$, $\boldsymbol{\zeta} = [\zeta_1, \cdots, \zeta_\phi]^T$, $\lambda$ and $\zeta_i$ are regularization parameters, and $\bar{\mathbf{A}}_i$ is a diagonal matrix, with the elements of $i^{th}$ row of $\widetilde{\mathbf{A}}$ on its main diagonal, defined as below:

$$\bar{\mathbf{A}}_i = \{ \bar{a}_{qv} | \bar{a}_{qv} = \widetilde{a}_{iq} \ \text{for all} \ q = v, \ \bar{a}_{qv} = 0 \ \text{for all} \ q \neq v \}. \tag{37}$$

In Eq. (36), $\zeta_i$ is chosen to be equal to the square of the residual error in (34), for each $i \in \mathcal{S}_{elec}$, i.e.,

$$\zeta_i = ((\widetilde{\mathbf{A}} \mathbf{C}_s^{full^{(r)}}(\gamma^{(r)}, \sigma_s^{(r)}) \widetilde{\mathbf{A}}^T)_{ii} + \widehat{\sigma}_{z_i}^2 - \widehat{Var}(y_i))^2. \tag{38}$$

In each iteration $r$, values of $\lambda$ and $k$ are found in a similar way as they are found in the low-resolution grid (see Eqs. (25) and (26)). However, to estimate $k$ based on Eq. (26), in the high-resolution grid we use $\mathbf{C}_s(k) = \mathbf{C}_s^{(r)}$, as is defined in Eq. (30). After each iteration, the set of silent indices in $\mathcal{S}^{(r)}$ is updated with the indices of the $\hat{k}$ smallest values in the solution of Eq. (36), i.e., $\mathbf{g}^{\star(r)}(\lambda^\star, \hat{k}, \zeta)$.

After convergence, i.e., when the convergence criterion is met (see Eq. (29)), the final estimate of region of silence is the set of source indices in $\mathcal{S}^{(r_{final})}$.

*Choosing the best reference electrode.* the final solution $\mathcal{S}^{(r_{final})}$ may change as we choose different EEG reference electrodes, which changes the matrix of differential signals of scalp $\mathbf{Y}$ and the forward matrix $\widetilde{\mathbf{A}}$ in Eq. (7). The question is how to choose a reference electrode, which gives us the best estimation of region of silence? To address this question, we use an approach similar to the estimation of $\hat{k}$, i.e., we choose the reference electrode, which gives us the minimum scalp power mismatch. We define the power mismatch $\Delta Pow$ as follows:

$$\Delta Pow = \sum_{i=1}^{n-1} \left\| \frac{(\widetilde{\mathbf{A}} \mathbf{C}_s(\hat{k}) \widetilde{\mathbf{A}}^T)_{ii}}{\max_i \ (\widetilde{\mathbf{A}} \mathbf{C}_s(\hat{k}) \widetilde{\mathbf{A}}^T)_{ii}} - \frac{\widehat{Var}(y_i) - \widehat{\sigma}_{z_i}^2}{\max_i \ (\widehat{Var}(y_i) - \widehat{\sigma}_{z_i}^2)} \right\|_2^2, \tag{39}$$

where both $\widetilde{\mathbf{A}}$ and $y_i$ are calculated based on a specific reference electrode. $\Delta Pow$ is the total squared error between the normalized powers of scalp differential signals, resulting from the region of silence with size $\hat{k}$, and the estimated scalp powers based on the recorded data with a specific reference.

**SilenceMap with baseline recordings.** If we consider a hemispheric baseline or, more generally, have a baseline recording, the 2-step SilenceMap algorithm remains largely the same. In an ideal case where we have a baseline recording of scalp potentials, we simply compare the contribution of each source in the recorded scalp signals when there is a region of silence in the brain, with its contribution to the baseline recording. This results in a minor modification of SilenceMap. The definitions of source contribution measures in Eqs. (17) and (35), need to be changed as follows:

$$\tilde{\beta}_q = \min \left\{ \frac{\beta_q}{\beta_q^{base}}, 1 \right\}, \ \ \text{for all} \ q \in \{1, 2, \cdots, p\} \tag{40}$$

where $\beta_q$ is defined in Eq. (17) for the low-resolution grid, and in Eq. (35) for the high-resolution grid, and $\beta_q^{base}$ is the corresponding contribution measure of source $q$ in the baseline recording. However, if the baseline recording is not available for the silence localization (as it was not available in our dataset used in the Results), based on the assumption of hemispheric symmetry in "Methods: Problem statement" (see assumption (viii)), one can use a hemispheric baseline. The source contribution measure is defined in a relative way, i.e., each source's contribution

measure is calculated in comparison with the corresponding source in the other hemisphere, as follows:

$$\tilde{\beta}_q = \begin{cases} \min\left\{\frac{\beta_q}{\beta_{q_m}}, 1\right\}, & \text{for all } q \in \mathcal{S}^{\text{LH}} \cup \mathcal{S}^{\text{RH}} \\ 1, & \text{for all } q \notin \mathcal{S}^{\text{LH}} \cup \mathcal{S}^{\text{RH}} \end{cases} \quad (41)$$

where $\mathcal{S}^{\text{LH}}$ is the set of indices of sources in the left hemisphere and $\mathcal{S}^{\text{RH}}$ is the set of indices of sources in the right hemisphere, and source indices, which are not in $\mathcal{S}^{\text{LH}} \cup \mathcal{S}^{\text{RH}}$, are located across the longitudinal fissure, which is defined as a strip of sources on the cortex, with a specific width $z^{\text{gap}}$. The index $q_m$ in Eq. (41) is the index of the mirror source for source $q$, i.e., source $q$'s corresponding source in the other hemisphere.

Equation (41) reveals the advantage of having a baseline for the silence localization task, i.e., we can relax the identical distribution assumption of sources in the source contribution measure, which makes $\tilde{\beta}$ robust against the violation of the identical distribution assumption of sources in the real world. The rest of the algorithm remains the same, as is explained in "Methods: SilenceMap without baseline recordings."

To find the solution of the CSpeC optimization in Eqs. (18) and (36), CVX, a MATLAB package for specifying and solving convex programs[64,65], is used. In addition, MATLAB nonlinear least-square solver is used to find the solution of Eq. (34).

**Time complexity of SilenceMap.** The bottleneck of time complexity among the steps in our algorithm is the high-resolution convex optimization (see Eq. (36)). This is classified as a convex quadratically constrained quadratic program. However, the quadratic constraints in Eq. (36) are all scalar and each can be rewritten in forms of two linear constraints. This reduces the problem to a convex quadratic program, which can be solved either using semidefinite programming[66] or second-order cone programming (SOCP)[67]. However, it is much more efficient to solve the QPs using SOCP rather than using the general solutions for SDPs[68,69]. Following the steps in[69], we can write our problem in Eq. (36) as a SOCP with $\nu = 2p + 2\phi + 1$ constraint of dimension one, and one constraint of dimension $p + 1$, where $p$ is the number of sources in the brain and $\phi$ is the number of selected electrodes in Eq. (33). Using the interior-point methods, the time complexity of each iteration is $\mathcal{O}(p^2(\nu + p + 1)) \approx \mathcal{O}(p^3)$, where the number of iterations for the optimizer is upper bounded by $\mathcal{O}(\sqrt{\nu + 1}) = \mathcal{O}(\sqrt{2p + 2\phi + 2})$[69]. Therefore, the CSpeC framework for high resolution (see Eq. (36)) has the worst case time complexity of $\mathcal{O}(p^{3.5})$. Similarly, the low-resolution CSpeC framework (see Eq. (18)), has the same time complexity of $\mathcal{O}(p^{3.5})$, since it only has $2\phi$ less linear constraints, in comparison with the quadratic program version of Eq. (36). It is important to mention that this time complexity is calculated without considering the sparsity of the graph Laplacian matrix ($\mathbf{L}$) defined in Eq. (23). Exploiting such sparsity may reduce the computational complexity of solving the equivalent SOCP for our CSpeC framework[70]. The other steps of SilenceMap have lower degrees of polynomial time complexity (e.g., the least-square solution in Eq. (34) with time complexity of $\mathcal{O}(2^2\phi)$, where $\phi \ll p$). Therefore, the general time complexity of SilenceMap is $\mathcal{O}(\text{itr}_{\text{ref}}(p^{3.5} + \text{itr}_{\text{conv}}.\text{itr}_k.\text{itr}_\lambda(p^{3.5}))) \approx \mathcal{O}(\text{itr}_{\text{ref}}.\text{itr}_{\text{conv}}.\text{itr}_k.\text{itr}_\lambda(p^{3.5}))$, including the number of iterations for finding the optimal regularization parameters ($\text{itr}_\lambda$ iterations for finding $\lambda^\star$ in Eq. (25), and $\text{itr}_k$ iterations for finding $k^\star$ in Eq. (26)), the required number of iterations for convergence of SilenceMap to a region of silence in the high-resolution step ($\text{itr}_{\text{conv}}$), and the number of iterations to find the best reference electrode ($\text{itr}_{\text{ref}}$). It is worth mentioning that time required to run SilenceMap depends on the resolution of the search grids for the parameters used in the algorithm (see Supplementary Table I in Supplementary Note A), the resolution of the cortical models used, and the convergence criterion defined (see Eq. (29)). We acknowledge that there is room for improvement of the implementation and the algorithm itself to obtain a faster silence localization tool, e.g., by exploiting the sparsity of the graph Laplacian matrix in solving Eq. (36), parallelizing the iterations of the algorithm, and exploring lower-cost clustering methods, and this is left for the future work.

**Modification of source localization algorithms for comparison with SilenceMap.** To compare the performance of SilenceMap with the state-of-the-art source localization algorithms, namely, MNE, MUSIC, and sLORETA, we modified them for the silence localization task. These modifications largely consist of adding additional steps to select the silent sources based on the estimated source localization in each algorithm. These modifications only make for a fairer analysis and answer the question of whether small modifications on existing source localization algorithms can localize silences. The details of these modifications are explained in this part.

*Modified minimum norm estimation (MNE).* MNE is one of the most commonly used source localization algorithms[15,21,22]. In this algorithm, the brain source activities are estimated based on a minimal power assumption, and through the following regularization method:

$$\widehat{\mathbf{S}} = \arg\min_{\mathbf{S}} \ \|\mathbf{Y} - \widetilde{\mathbf{A}}\mathbf{S}\|_F^2 + \lambda \|\mathbf{S}\|_F^2, \quad (42)$$

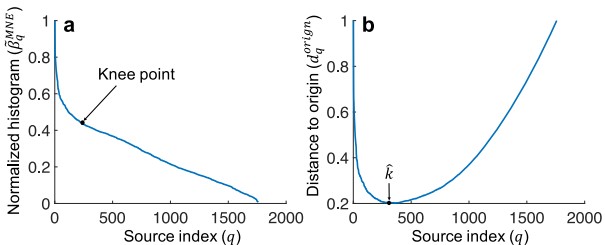

**Fig. 6 Estimation of the size of the region of silence ($k$) in the modified MNE algorithm based on the knee point detection. a** $\tilde{\beta}_q^{\text{MNE}}$ is the normalized and sorted histogram of sources in a descending order, which captures the frequency of a source to being among the $\hat{k}$ sources with the minimum power over time. In this curve, a Knee point is defined as the point with maximum curvature, i.e., the point where the curve is substantially different from a straight line; **b** Distances of points on the curve in **a** from the origin ($q = 0, \tilde{\beta}_q^{\text{MNE}} = 0$). The index of the point with the minimum distance from the origin ($\hat{k}$) is chosen as an estimation of $k$.

where $\widehat{\mathbf{S}}$ is the estimated matrix of source signals, $\widetilde{\mathbf{A}} = \mathbf{MA}$, $\mathbf{Y}$ is the matrix of scalp differential signals defined in Eq. (6), $\lambda$ is the regularization parameter, and $\|.\|_F\|.\|_F$ denotes the Frobenius norm of a matrix. Equation (42) has the following closed form solution:

$$\widehat{\mathbf{S}} = \widetilde{\mathbf{A}}^T\left(\widetilde{\mathbf{A}}\widetilde{\mathbf{A}}^T + \lambda \mathbf{I}_{(n-1)\times(n-1)}\right)^{-1}\mathbf{Y}, \quad (43)$$

where $\mathbf{I}$ is the identity matrix, and $\lambda$ is obtained using a grid-search and based on the $L - \text{curve}$ method[71]. The MNE algorithm is kept unchanged until this point. $\widehat{\mathbf{S}}$, the estimated localization across time, is used to localize silences. For a fair comparison, we do so by using the two-step approach used in SilenceMap, i.e., we start from a low-resolution source grid and localize the region of silence, which is used as an initial guess for source localization in a high-resolution grid.

Low-resolution grid: In a low-resolution source grid, we localize the region of silence through the following steps: (i) We initialize the number of silent sources as $\hat{k} = k_0$; (ii) The squares of the elements in $\widehat{\mathbf{S}}$ ($\hat{s}_{ij}^2, \forall i = \{1, 2, \cdots p\}, \forall j = \{1, 2, \cdots t\}$) are calculated for source power comparison; (iii) For each time point $j$, sort the estimated source powers $\hat{s}_{ij}^2$ in the ascending order and choose the first $\hat{k}$ corresponding sources, which are the sources with the minimum power at time $j$. We name the set of indices of these sources as $\mathcal{S}_{\text{MNE}}^j$; (iv) Based on the repetition of sources in $\mathcal{S}_{\text{MNE}}^j$, we calculate a histogram. Then this histogram is normalized and sorted in the descending order (the source with the largest population of 1 has the first index). The normalized population of source $q$ is shown as $\tilde{\beta}_q^{\text{MNE}}$; (v) In this step, we find an estimate of the size of region of silence ($\hat{k}$). This is done by finding the knee point in the curve of $\tilde{\beta}_q^{\text{MNE}}$ vs. $q$ (see Fig. 6a). In a curve, the knee point is defined as the point where the curve has maximum curvature, i.e., the point where the curve is substantially different from a straight line[72–74]. To find the knee point in the curve of $\tilde{\beta}_q^{\text{MNE}}$ vs. $q$, we define a measure of distance to the origin ($q = 0, \tilde{\beta}_q^{\text{MNE}} = 0$) as follows:

$$d_q^{\text{origin}} = \left(\tilde{\beta}_q^{\text{MNE}}\right)^2 + \left(\frac{q}{p}\right)^2, \quad (44)$$

where $d_q^{\text{origin}}$ is the defined distance of point ($q = 0, \tilde{\beta}_q^{\text{MNE}} = 0$) on the curve to the origin, and $p$ is the total number of sources in the descritized brain model. Fig. 6b shows the calculated $d_q^{\text{origin}}$ for the curve in Fig. 6a. We choose the closest point to the origin as the knee point ($q = \hat{k}$), where the index of this knee point $\hat{k}$ is an estimation for $k$ (see Fig. 6b).

(vi) In this step, we exploit the knowledge of contiguity of the region of silence and estimate the region based on the estimated number of silent sources $\hat{k}$. First we choose the $2\hat{k}$ sources with the minimum power over time, i.e., the $2\hat{k}$ sources, which have maximum $\tilde{\beta}_q^{\text{MNE}}$. Then we calculate the COM of the $2\hat{k}$ selected sources in the low-resolution grid ($\mathbf{f}_{\text{MNE}}^{\text{low}}$), and choose the $\hat{k}$-nearest neighbors of $\mathbf{f}_{\text{MNE}}^{\text{low}}$ as the estimated region of silence in the low-resolution grid.

High-resolution grid: We use the COM of the estimated region of silence in the low-resolution grid ($\mathbf{f}_{\text{MNE}}^{\text{low}}$), as an initial guess and try to improve the localization performance in a high-resolution source grid. The steps are mainly the same as the steps used in the low-resolution grid, except in the last step (vi), where we use the COM of the estimated region of silence in the low-resolution grid, and choose the $\hat{k}$-nearest neighbors of $\mathbf{f}_{\text{MNE}}^{\text{low}}$ as the estimated region of silence in the high-resolution grid, where $\hat{k}$ is the estimated size of region of silence in the high-resolution grid based on the knee point detection method in step (v).

*Modified multiple signal classification (MUSIC)*. MUSIC is a source localization algorithm, which is based on a sequential search of sources, rather than finding all sources at the same time[19,20]. In MUSIC, the singular value decomposition (SVD) of the matrix of scalp recording signals $\mathbf{Y}_{(n-1) \times T} \left( = \mathbf{U}_{(n-1) \times (n-1)} \mathbf{\Sigma}_{(n-1) \times T} \mathbf{V}_{T \times T}^T \right)$ is used to reconstruct an orthogonal projection to the noise space of $\mathbf{Y}$ to quantify the contribution of each source in the recorded signal $\mathbf{Y}$[15]. The MUSIC algorithm follows these steps for source localization: (i) We select the left singular vectors (columns of $\mathbf{U}$), which correspond to the large singular values up to $\rho$% of the total energy of the matrix $\left( \sum_{i=1}^{(n-1)} \Sigma_{ii}^2 \right)$, where $\rho$ is a constant. These selected singular vectors ($\mathbf{U}_s$) form a basis for the observation data; (ii) We construct an orthogonal projection matrix to the noise space of $\mathbf{Y}$ as $\mathbf{P}^\perp = \mathbf{I}_{(n-1) \times (n-1)} - \mathbf{U}_s \mathbf{U}_s^T$. Using this matrix the MUSIC cost function is written as[15]:

$$\beta_q^{\text{MUSIC}} = \frac{\| \mathbf{P}^\perp \widetilde{\mathbf{a}}_q \|_2^2}{\| \widetilde{\mathbf{a}}_q \|_2^2}, \tag{45}$$

where $\widetilde{\mathbf{a}}_q$ is the $q^{th}$ columns in $\widetilde{\mathbf{A}}$, and $\beta_q^{\text{MUSIC}}$ is a measure of contribution of source $q$ in the noise space of the recorded scalp potentials in $\mathbf{Y}$. The MUSIC algorithm is kept unchanged until this point. The next steps of the modified MUSIC algorithm, in both low-resolution and high-resolution grids closely follow the last two steps in the Modified MNE algorithm, and we use the measure of source contribution in MUSIC ($\beta_q^{\text{MUSIC}}$) instead of $\widetilde{\beta}_q^{\text{MNE}}$, where the source with no contribution in the differential measured signal $\mathbf{Y}$ has $\widetilde{\beta}_q^{\text{MNE}} = 1$. Therefore, the main difference between the MUSIC algorithm and the modified MUSIC, is that in the MUSIC, the measure of contribution of source $\beta_q^{\text{MUSIC}}$ is used to fined the active sources, i.e., the sources with small $\beta_q^{\text{MUSIC}}$ values, while for the silence localization the sources with large $\beta_q^{\text{MUSIC}}$ values are selected based on the contiguity assumption of the region of silence and using the knee point thresholding mechanism (step (v) in the modified MNE).

*Modified standardized low-resolution brain electromagnetic tomography (sLORETA)*. We modify the source localization sLORETA algorithm, introduced in ref. [24], in the same way that we modified the MNE algorithms for the silence localization. However, the minimum-norm solution requires an additional step of normalization by the estimated source variances. Since we assume that the orientations of dipoles in the brain are known, i.e., they are normal to the surface of the brain, following equation (22) in ref. [24], the estimated power of source activities in the brain based on the sLORETA algorithm is given by the following equation:

$$\hat{s}_{it}^2 = \frac{\widetilde{s}_{it}^2}{(\mathbf{C}_s^{\text{sLORETA}})_{ii}}, \text{ for all } i = \{1, 2, \cdots p\}, \tag{46}$$
$$\text{for all } t = \{1, 2, \cdots T\}.$$

where $\widetilde{s}_{it}$ is the $i^{th}$ element of the $t^{th}$ column in the minimum-norm solution $\widetilde{\mathbf{S}}$, which is given by Eq. (43), and $\mathbf{C}_s^{\text{sLORETA}}$ is defined as[24]:

$$\mathbf{C}_s^{\text{sLORETA}} = \widetilde{\mathbf{A}}^T \left( \widetilde{\mathbf{A}} \widetilde{\mathbf{A}}^T + \lambda \mathbf{I}_{(n-1) \times (n-1)} \right)^{-1} \widetilde{\mathbf{A}}, \tag{47}$$

$\widehat{\mathbf{S}}$ in Eq. (46) is used for the silence localization task, following the steps mentioned for the modified MNE algorithm. The minimum-norm solution in the sLORETA algorithm, as is defined in ref. [24], is based on an average reference for the scalp potentials. However, we rewrite the minimum-norm solution as $\widetilde{\mathbf{S}}$ in Eq. (43) based on a specific reference electrode, rather than the average reference electrode. The parameters used in the implementation of these modified source localization algorithms are available in the Supplementary Table I.

### Data analysis

*Prepossessing steps*. We preprocess the recorded EEG signals using EEGLAB[75] toolbox in MATLAB. First, we bandpass filter the EEG data in the frequency range of [1,100]Hz using a Hamming windowed sinc finite impulse response (FIR) filter. Then, we visually inspect the noisy channels, remove and spatially interpolate them. In the next step, we calculate two differential channels based on the pairs of eye electrodes, one for vertical and one for the horizontal eye movements, and along with the heart channel and all scalp electrodes, an independent component analysis is applied to remove the eye artifacts and heart beats from the EEG signals. After removing the artifact components from the EEG signals, we examine the channels one more time using the channel statistics, where a normal distribution is fitted to the data of each channel and based on the standard deviation, skewness, and kurtosis, channels with substantially different statistics are removed and interpolated. Finally, the signals are epoched into 2-s intervals and epochs with abnormal trends, values, and/or abnormal power spectral densities are removed, using the EEGLAB toolbox.

*Ground truth regions and MRI scans*. The ground truth regions of silence are extracted based on the MRI scans of patients (see Fig. 2) following these steps: (i) 3D models of descritized cortex are extracted by processing the MRI scans using the FreeSurfer software[76–82], and removing the layers of the head, namely, CSF,

skull, and scalp using the MNE open-source software[83], (ii) the sources/nodes of the intact hemisphere, i.e., the hemisphere without any missing part, are mirrored along the longitudinal fissure, (iii) the smallest distance of the mirrored sources are calculated from the sources in the hemisphere with the resected part, (iv) $N$ sources with the largest distance are selected as the region of silence, where $N$ is determined by visual comparison of the extracted ground truth in the 3D model, and its corresponding MRI scan.

Displayed figures in this paper are generated using MATLAB, Microsoft PowerPoint, and FreeSurfer software.

### Simulated dataset

*Flat PSD simulations*. We simulate EEG signals at 128 electrodes, located at the 10-5 standard system of scalp locations[35], as follows: (i) First, we use a high-density source grid, extracted by discretizing a real brain model, and randomly choose a node along with its $k$ nearest neighbors, as the region of silence with size $k$, (ii) then we simulate the source signals using a multivariate Gaussian random process with independent time points (flat PSD), and a zero-lag covariance matrix $\mathbf{C}_s$ defined in Eq. (9), where $\mathcal{S}$ is the set of indices of silent sources specified in the first step, the source variance is $\sigma_s = 1$ *a.u.*, where *a.u.* is an arbitrary unit for amplitude, and the exponential decay coefficient is $\gamma = 0.12$ (mm)$^{-2}$, (iii) the measurement noise in Eq. (7), i.e., $\widetilde{\mathbf{E}}$, is simulated using a multivariate gaussian random process with a covariance matrix $\mathbf{C}_z$ defined in Eq. (8), where $\sigma_{z_i}$ is chosen randomly from a uniform distribution in the range of $[0, \sigma_z^{\max}]$, and $\sigma_z^{\max}$ is chosen so that the baseline EEG signals, i.e., without any region of silence, have a specific average SNR, defined as below[84–86]:

$$\text{SNR}_{\text{avg}} = 10\log_{10} \left( \frac{1}{n-1} \sum_{i=1}^{n-1} \frac{(\widetilde{\mathbf{A}} \mathbf{C}_s^{\text{full}}(\gamma, \sigma_s) \widetilde{\mathbf{A}}^T)_{ii}}{\sigma_z^{\max}} \right) \tag{48}$$

(iv) In the next step, the forward matrix $\mathbf{A}$ in Eq. (5) is calculated based on a real head model, which is obtained from MRI scan of patient OT in the real database (Fig. 2). In this paper, we use the FreeSurfer software[76–82] to process the MRI images, and use MNE open-source software[83] to extract different layers of the head, i.e., CSF, skull, and scalp. Then having these layers of head, and using the boundary element method, the forward matrix $\mathbf{A}$ is computed, where a free MATLAB toolbox, FieldTrip[87], is used. In the boundary element method model, volume conductivity ratios of [1,0.067,5,1] are used for scalp, skull[88], CSF, and brain, respectively. We take the intact hemisphere (the one without any missing parts) and mirror it across the longitudinal fissure to form a symmetric brain model, which is used as the source grid in our algorithm. In "Results" we discuss and quantify the localization error we introduce by using this symmetric brain model. (v) Finally, the calculated forward matrix $\mathbf{A}$ is used to simulate the scalp EEG signals following the Eq. (7).

*Real PSD simulations*. To explore the effect of PSD profile of the non-silent neural sources on the performance of SilenceMap, we simulate source activities with Real PSD following these steps: (i) First, we extract a general shape of PSD for the normal brain activities based on a real recorded electrocorticography (ECoG) dataset used in ref. [33] and available through the open-source library in ref. [34]. This general shape of PSD results from averaging over the PSDs of the ECoG recordings of an epileptic patient in ref. [34] (see Supplementary Fig. 6 and more details on the reprocessing and the average PSD extraction in Supplementary Note H). (ii) In the next step, we design a linear phase FIR filter (Supplementary Fig. 7 in Supplementary Note H), with the magnitude equal to the square root of the noiseless average PSD shown in Supplementary Fig. 6. (iii) Following the steps (i) and (ii) in the previous section (Flat PSD simulations), we simulate non-silent source activities with a flat PSD (instead of the 1/f behavior observed in practice, see Supplementary Fig. 8), and then apply the designed filter on them. This results in simulated signals, which have PSDs similar to the PSD extracted from the real recorded ECoG signals in the brain, called Real PSD hereinafter (see Supplementary Fig. 9). We assume the signals have identical distribution over the cortical space, but with spatial and temporal dependency profiles extracted from the real recordings of the brain. Following the steps (iii) to (v) in the previous section (Flat PSD simulations), we obtain the electrical signals of scalp EEG electrodes based on the Real PSD simulated signals in the brain. The general shape of PSD for the simulated EEG signals based on the Real PSD (see Supplementary Fig. 10) is close to the PSD of a real recorded EEG signal from a patient with a region of silence in the brain (see Supplementary Fig. 11, showing the PSD of Rest recordings for patient OT in this study, bandpass filtered in the [1,100]Hz interval).

*The effect of different PSD profiles on the silence localization performance*. Based on these results presented in Table 1, SilenceMap shows almost the same performance for the flat and Real PSDs. Why do the results not change substantially? To understand this, we looked at the effect of changing the (temporal) PSD on the spatial correlation for the simulated brain sources. We observe that it is indeed expected that changing this PSD does not affect the spatial correlations (as we next discuss), and hence the localization, which only depends on spatial correlations (with sufficient data) is unaffected.

We assume an exponential decay profile for the source covariance matrix (see $\mathbf{C}_s$ defined in Eq. (9)), which is consistent with our assumption in the flat PSD

simulations. In section "Real PSD simulations", the cross-correlation coefficients of the neural source activities are being changed through the filtering step (see step (iii) in section "Real PSD simulations"). This change can be explored by looking at the filtering process in the time domain. Let's assume $S_i(t)$ is the simulated signal, using the flat PSD, at the $i^{th}$ source and time point $t$, and $h(t)$ is an FIR filter:

$$S'_i(t) = (h * S_i)(t) = \sum_{q=1}^{N_h} h(q)S_i(t-q),\qquad(49)$$

where $S'_i(t)$ is the filtered signal with the Real PSD at the $i^{th}$ source, and $N_h$ is the length of the FIR filter $h$. The cross-correlation function of $S'_i(t)$ at lag $l$ is as follows:

$$
\begin{aligned}
R_{S'_iS'_j}(l) &= E[S'_i(t)\overline{S'_j(t+l)}]\\
&= E\left[\sum_{q=1}^{N_h}h(q)S_i(t-q)\overline{\sum_{r=1}^{N_h}h(r)S_j(t+l-r)}\right]\\
&= \sum_{q=1}^{N_h}\sum_{r=1}^{N_h}h(q)\overline{h(r)}E\left[S_i(t-q)\overline{S_j(t+l-r)}\right]\\
&= \sum_{q=1}^{N_h}\sum_{r=1}^{N_h}h(q)\overline{h(r)}R_{S_iS_j}(q+l-r),
\end{aligned}
\qquad(50)
$$

where $R_{S_iS_j}(q+l-r)$ is the cross-correlation of simulated signals at the $i^{th}$ and $j^{th}$ sources with flat PSDs, which implies independence over time points:

$$R_{S_iS_j}(l) = c_{s_{ij}}\delta(l),\qquad(51)$$

where $\delta(l)$ is the unit sample function with value 1 at $l = 0$ and value 0 elsewhere, and $c_{s_{ij}}$ is the zero-lag cross-correlation of brain activities at the $i^{th}$ and $j^{th}$ sources defined in Eq. (9). Based on the equality in Eq. (51), we can rewrite the Eq. (50) as follows:

$$R_{S'_iS'_j}(l) = \sum_{q=1}^{N_h}h(q)\overline{h(l+q)}R_{S_iS_j}(0) = \rho_h(l)R_{S_iS_j}(0),\qquad(52)$$

where, $\rho_h(l)$ is the autocorrelation of the FIR filter $h(t)$ with lag $l$. Based on the Eq. (52), after filtering the flat PSD signals $\mathbf{S} \in \mathbb{R}^{p \times T}$ with the covariance matrix of $\mathbf{C}_s \in \mathbb{R}^{p \times p}$, the resulting signals $\mathbf{S}' \in \mathbb{R}^{p \times T}$ have a covariance matrix at zero lag, which is only a scaled version of $\mathbf{C}_s$ by a constant $\rho_h(0)$. In addition, some non-zero-lag ($l \neq 0$) correlations show up in $\mathbf{S}'$, which means that unlike the flat PSD signals, the simulated signals using the Real PSD are not independent over time. In designing SilenceMap, we do not assume temporal independence for the brain source signals, which explains why there is no silence localization performance reduction in the Real PSD experiments. However, due to the temporal correlations in section "Real PSD simulations", a larger number of time points are required for the variance and covariance estimations in our algorithm to achieve the same performance as the flat PSD results. In Table 1, we have used $T = 100{,}000$ time points for all of the simulations.

**Statistics and reproducibility**. We recruited three participants with different resections against whose data we could run the various analytic comparisons. A spatial resolution of 128 scalp EEG electrodes was used for each participant. We recorded 320 s of EEG data (160 s for Rest state and 160 s for the Visual task), with a sampling frequency of 512 Hz, which results in a total number of 81,920 data points over time (for each task), which is considered to be a large enough sample for the statistical estimations in this study (i.e., mean, covariance, power spectral density (PSD), and noise). In addition to the recordings from the participants in this study, we simulated 100 regions of silence at 100 different random locations on a real brain model extracted from the MRI scans. This sample size of 100 regions of silence was large enough to keep the reported standard errors (SE) small. Last, we included a single control individual for further evaluation of the hemispheric assumptions.

The silence localization was repeated based on the Visual dataset for each participant, and the performance was compared to the silence localization based on the Rest dataset. The results remained largely the same, which verifies the reproducibility of the experimental findings. In addition, we repeated the silence localization for different temporal lengths of the EEG data, for each participant, and the results are compared in the Result section.

**Ethics oversight**. All procedures were approved by the Carnegie Mellon University Institutional Review Board (IRB HS13-666 for the controls, and IRB HS14-607 for the patients, "Visual recovery after severe brain injury or visual pathway disturbance").

**Reporting summary**. Further information on research design is available in the Nature Research Reporting Summary linked to this article.

## Data availability
The anonymized raw EEG dataset and MRI scans (shown in Fig. 2) of the participants in this research are made available online on KiltHub, Carnegie Mellon University's online data repository (https://doi.org/10.1184/R1/12402416[89]).

## Code availability
SilenceMap was developed in MATLAB, using standard toolboxes, and the CVX MATLAB package[64,65]. All MATLAB code is made available online on GitHub (https://doi.org/10.5281/zenodo.3892185[90]).

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

## Acknowledgements

This work was supported, in part, by grants from the Chuck Noll Foundation for Brain Injury Research, CMU BrainHub, the Pennsylvania Infrastructure Technology Alliance, the NSF STTR program, the Center for Machine Learning and Health at CMU, under the Pittsburgh Health Data Alliance, and a CMU Swartz Center Innovation Fellowship. M.B. was supported by a grant from the NIH (NEI GrantRO1EY027018). We thank Chaitanya Goswami, Praveen Venkatesh, Ashwati Krishnan, Marge Maallo, Sarah Haigh, Animesh Kumar, and Maysamreza Chamanzar for helpful discussions, Ashwati Krishnan and Patricia Brosseau for help in data collection, and Daniel Glen (Computer Engineer, NIH) for sharing the script for segmenting structural MRI scans in AFNI.

## Author contributions

A.C., M.B., and P.G. initiated, designed, and executed the research. A.C. and M.B. acquired and interpreted the data. M.B. and P.G. supervised the data collection. A.C. and P.G. developed the algorithms. A.C. developed the software tools necessary for conducting the experiments and analyzing the data. A.C., M.B., and P.G. wrote the manuscript.

## Competing interests

The authors have applied for a provisional patent on the technology, assigned to Carnegie Mellon University. P.G. and M.B. are co-founders of a medical device company that intends to license the resulting patent from Carnegie Mellon University, and A.C. has equity in this company.
