## [Peer Review File · Communications Biology]

Reviewers' comments:

Reviewer #1 (Remarks to the Author):

The major claim of the authors is that their proposed 'SilenceMap' algorithm can be applied to noninvasive scalp EEG to localize neural 'silences' in patients with brain injury such as lobectomy, tumors or stroke. Although structural MRI can provide anatomical information about the location of such brain lesions, it is also important to monitor and understand the effects of such lesions on possibly evolving (abnormal) brain activity in the underlying areas, e.g., after a lesion (in the acute phase), during development or aging. Moreover, the neural silence information may also be helpful to decode cognitive-motor-affective states for brain-computer interface application, including neuromodulation. Thus, the potential impact of this work on neural engineering and clinical neurophysiology is high.

I commend the authors for their spirit of data and algorithm sharing from their research. This will allow the scientific community to further validate their proposal and perhaps improve upon it. I also commend the authors for their upfront review of assumptions and limitations of their algorithm. Please see below for additional issues.

Nevertheless, there are however several concerns with the work, which I present below in no particular order:

1) It has been shown that changes in electrode impedance during EEG recordings can significantly alter the distribution of artifacts (e.g., eye blinks, etc) across the scalp (see for example, DOI: 10.1088/1741-2560/13/2/026013), as well as the patterns of ERPs and even rest activity. This would affect the assumptions of the algorithm and thus the accuracy of the estimates of neural silences. Authors should indicate whether electrode impedance was measured/monitored and indicate the values/threshold.

2) Source analysis: The authors do not mention whether the electrode locations were used during the source analyses as they would likely affect the solutions. Likewise, segmentation of MRIs of patients with brain injury is complicated, and in the developing brain is very difficult, particularly in the first 2 years of age (subjects here were older children). This seems to have been done by visual inspection with the help of the MNE software. This can be an important source of errors.

3) Volume conductivity. Volume conductivity ratios can greatly affect the source localization. What was the basis for selecting [1,0.067,5,1] as the estimates in the source analysis (forward matrix) and the simulated datasets? Do the authors neglect the boundaries between intact and resected brain tissues? How do they treat the forward matrices? how would a tumor or diffuse cortical lesion affect their algorithm?

4) Cross-validation: The authors present findings from 3 children with lobectomy and eeg simulations. The authors tested their algorithm with patients with brain resections only. It would have been useful to show the method in a patients with cortical or subcortical stroke, or tumor.

5) The authors claim that their proposed method permit 'rapid detection and localization of regions of silence ...' ; however, it seems that this multi-steps method would depend on computational

resources and may not be particularly fast. Can you please provide some computational metrics for the algorithm (computers used, % of FP operations, processing time, memory, etc)

6) Functional connectivity: Even at rest, but particularly during ERP or willed tasks, functional connectivity is an important metric of brain-to-brain communication, which would be expected to be affected by regions of 'silence'. I think they could provide some additional validation when applied to both real and simulated EEG data.

7) The main goal of the paper, according to the authors, is the localization of the region of silence of patients with lobectomies. Can the authors comment on the time, frequency, time-frequency characteristics of the EEG that define this region in the real datasets of the patients? can you please compare them to the simulated EEG?

8) The model assumes symmetry across hemispheres. There are many studies that have shown cortical asymmetry and how it is affected by age, disease, etc. The authors should discuss this.

Overall, given the points above, I am not convinced that the proposed method substantially outperforms existing modified source locations methods. This needs to be revisited.

9) Could the authors comment on the relatively small amount of EEG data used? Did they compute localization accuracy as a function of amount of EEG data (say 3 min vs tens of minutes)?

10) Abstract: The abstract appears to claim more than it is offered. The authors talk about 'a rapid and cost-effective tool to detect and characterize abnormal neural function...' however, the patient group was limited to a small group of subjects with lobectomy only, and it does not yet characterize abnormal neural function.

11) please provide the data (as in Figure 1) for the other 2 patients in the appendix/ supplementary material section.

12) Please include a table with list of assumptions and potential effect on the neural silence estimate, and possible way to mitigate the assumption.

Reviewer #2 and #3 (Remarks to the Author):

The authors have described a novel and interesting algorithm for purposes of localizing regions of brain damage, or regions cortical silence. The paper is very thorough and technically sound. The algorithms and equations are straight-forward and intriguing to read about. I applaud the authors for developing this technique. However, there are a few main issues that need to be addressed before the manuscript is suitable for publication:

1) The papers claim that the current algorithm is superior to others, and make comparisons to support this idea. The descriptions of these other algorithms is lacking and a discussion specifically on why they don't function as well as the current should be included.

2) Being able to detect a lack of a signal, rather than the presence unique signal, poses a unique

challenge in terms of background noise. The researchers need to overcome background to a great degree to be ensure that a negative signal is being recorded. There was not much detail about this in the manuscript.

3) The brain physiology background component was unclear as well. It appears that the success of the algorithm is dependent on finding a cortical baseline for comparison. This can be done if you know where the damage is but may be challenging otherwise. How do the researchers expect to overcome a case where the damage is diffuse or unknown so that a reliable baseline cannot be established?

4) Although the algorithm is interesting and the authors provide proof of concept on real brain recordings. The extent of damage in the patients used here is so great that any standard neuropsychological exam could easily pick this up and localize lesion with good accuracy. Therefore, no machine would be needed. Thus, there is a question of sensitivity here, if the researchers hope to use this in a medical setting as suggested in the introduction, they need to show it can be reliably used in smaller lesions. At this stage they might not be able to answer this question, but they should at least think forward on how to adjust the algorithm to increase localization sensitivity.

Neural silences can be localized rapidly using noninvasive scalp EEG

Alireza Chamanzar, Marlene Behrmann, Pulkit Grover

Response letter cover:

We thank the reviewers and the editor of our paper for the positive feedback and constructive comments they provided through the peer review process. We are pleased that they have found the paper “interesting with high potential impact on neural engineering and clinical neurophysiology”. In this revised version of manuscript, we believe we have fully addressed the reviewers’ comments. We have also performed additional analyses based on their suggestions and to provide additional evidence for our claims. We are grateful for these suggestions, which has, in our view, resulted in substantial improvement in our paper. We apologize for the length of this response letter. As will be evident, we have excerpted the relevant sections of the paper which have been revised (and have added new analyses and references) and this has resulted in an unusually long response. Following is an outline of the major changes in this revision and, in the point-by-point response letter attached, we provide a list of detailed changes to the reviewers’ comments:

- **New pipeline for the segmentation of the MRI scans based on the open source AFNI software:** The results of the silence localization method based on the anatomical segmentation of MRI scans using *AFNI* for the three pediatric patients in our study are included in Section F of the Supplementary Materials, and briefly discussed in Section III. Please see our response to Reviewer#1’s comment 2-part2.
- **Exploring the effect of brain-to-skull conductivity ratio (BSCR) on the silence localization performance:** In addition to the results for a BSCR value of 15 that were included in the initial submission, the results of the silence localization based on a BSCR value of 80 are discussed in Section III and included in the Supplementary Materials, Section G. Please see our response to Reviewer#1’s comment 3-part1.
- **Exploring the effect of time-frequency characteristics of the brain activities on the silence localization performance:** Simulation experiments were conducted based on a real power spectral density (PSD), extracted from an open source electrocorticography (ECoG) dataset, and the results are included and discussed in Section II (Table I), with more details available in Section IV G and Supplementary Materials, Section D. Please see our response to Reviewer#1’s comment 7.
- **Neuropsychological exam results for UD and OT patients:** The performance of standard neuropsychological exams in detection and localization of the resections in the two of the three pediatric patients in our study is now discussed in response to Reviewer#2 and 3’s comment 4-part1. These details are included in Supplementary Materials, Section H, and briefly discussed in Section II.

In addition, some minor modifications have been done in the revised paper to clarify the claims, and to add further discussion to the sections on the limitations and the future directions of the paper. We have also included additional figures and tables following the reviewers’

suggestions and comments, which makes the content of the paper easier to understand and follow.

Following is our point-by-point responses to the reviewers' comments:

Color codes:

- Reviewers' comments/suggestions/feedbacks
- Authors' responses
- Modified/added texts in the revised manuscript

Reviewers' comments:

Reviewer #1 (Remarks to the Author):

The major claim of the authors is that their proposed 'SilenceMap' algorithm can be applied to noninvasive scalp EEG to localize neural 'silences' in patients with brain injury such as lobectomy, tumors or stroke. Although structural MRI can provide anatomical information about the location of such brain lesions, it is also important to monitor and understand the effects of such lesions on possibly evolving (abnormal) brain activity in the underlying areas, e.g., after a lesion (in the acute phase), during development or aging. Moreover, the neural silence information may also be helpful to decode cognitive-motor-affective states for brain-computer interface application, including neuromodulation. Thus, the potential impact of this work on neural engineering and clinical neurophysiology is high.

I commend the authors for their spirit of data and algorithm sharing from their research. This will allow the scientific community to further validate their proposal and perhaps improve upon it. I also commend the authors for their upfront review of assumptions and limitations of their algorithm. Please see below for additional issues.

We thank the reviewer for the positive assessment of the manuscript and the approach we adopt.

Nevertheless, there are however several concerns with the work, which I present below in no particular order:

1) It has been shown that changes in electrode impedance during EEG recordings can significantly alter the distribution of artifacts (e.g., eye blinks, etc) across the scalp (see for example, DOI: 10.1088/1741-2560/13/2/026013), as well as the patterns of ERPs and even rest activity. This would affect the assumptions of the algorithm and thus the accuracy of the estimates of neural silences. Authors should indicate whether electrode impedance was measured/monitored and indicate the values/threshold.

Response to 1) We thank the reviewer for this important comment. As mentioned in the paper, Section II, “Real Data” subsection, we used the Biosemi ActiveTwo system with active electrodes which can tolerate high electrode impedances and thus the system can be used without the usual skin preparation [1,2]. In addition, we monitored the electrode-gel-scalp contact quality through the data acquisition period using the “Electrode Offsets” option in the *ActiView* data acquisition software, which calculates the DC potentials generated at the junction of the skin and electrolyte solution (gel) under the electrodes. This DC potential results in a voltage at the amplifier inputs (i.e., DC offset) [3]. Electrodes with larger than 20mV offset were marked for removal and their signal was interpolated in the preprocessing step. Before starting the next recording session, we added more conductive gel to the electrodes with high offset. This is all reported in Section II.

In addition, we recorded eye blinks, eye movements, and heart beats using EOG and ECG electrodes, and applied ICA to remove these artifacts in the preprocessing step. After removing these artifact components from the EEG signals, we examined the EEG electrodes using the statistics of the scalp signals, where a normal distribution is fitted to the data of each electrode and, based on the standard deviation, skewness, and kurtosis, electrodes with significantly different statistics are removed and interpolated. Finally, the signals were epoched into 2-second intervals and epochs with abnormal trends, values, and/or abnormal power spectral densities were removed, using the *EEGLAB* toolbox. During the epoching, trials which were contaminated with the artifacts missed in the ICA step will be removed. This mostly compensates for the shortcoming of ICA in removing artifacts in case of changes in the electrode impedance [4].

We are aware that in clinical applications and situations where rapid recording and localization of neural silences are required, there are several time-consuming steps for EEG installation namely, placement of electrodes and application of conductive gel, which may take up to 30 minutes for the high-density EEG cap we used in our experiments, electrode contact quality monitoring and corrections, and multistep and offline data preprocessing. This overhead can limit the use of silence localization in practice. Fortunately, there has been great progress over the past few years in designing portable and easy-to-administer EEG systems with dry and active electrodes, where the electrode impedances are kept low without requiring significant additional care (monitoring the offsets and/or impedances, adding conductive gel, etc.) during the recording and acquisition period [5,6,7]. In addition, fast and real-time preprocessing and artifact removal techniques for EEG signals have been developed (e.g., in [4]). Using the fast-to-setup EEG systems, real-time preprocessing techniques, along with SilenceMap, which only requires less than 3 minutes of EEG recording, and having access to enough computational power (see Section IV D, subtitle “Time Complexity of SilenceMap”), can pave the way towards rapid silence localization.

To address this comment, we included this discussion in Section II, under the “Real Data” subtitle:

Lines 309-322:

two electrodes were placed on the mastoids. All electrodes
were differentially recorded relative to the standard common-
mode-sense (CMS) and driven-right-leg (DRL) electrodes.
We monitored the electrode-gel-scalp contact quality through
the data acquisition period using the “Electrode Offsets” op-
tion in the *ActiView* data acquisition software, which calcu-
lates the DC potentials generated at the junction of the skin
and electrolyte solution (gel) under the electrodes. This DC
potential results in a voltage at the amplifier inputs (i.e., DC
offset) [36]. Electrodes with larger than 20 *mV* offset were
marked for removal and interpolation in the preprocessing
step, and more conductive gels were added to the electrodes
with high offset. This is important as the change in electrode
impedance can significantly alter the distribution of artifacts
(e.g., eye blinks and eye movements) across the scalp and
make it harder for the preprocessing methods to detect and
remove them [37]. During the acquisition of EEG data, the

And in Section III (Discussion), under the “SilenceMap can localize the regions of silence with relatively little EEG data.” subtitle:

Lines 497-513:

[20, 40, 80, 120, 160]s, quantifying the performance for each
timespan. For UD, 80s of data showed almost the same per-
formance as 160s ($\Delta COM = 17mm$, $JI = 0.382$, $\Delta k = 0.30$),
while 40s showed significant reduction. For SN, the minimum
possible amount of data, without compromising the localiza-
tion performance, is only 40s ($\Delta COM = 9mm$, $JI = 0.440$,
$\Delta k = 0.20$), while for OT, this is 160 sec, potentially due to the
noisy EEG recording of OT, as discussed in Section II. Nev-
ertheless, the 160s upper limit is still a relatively short amount
of signal acquisition time.

In clinical applications, rapid recording and localization of
neural silences might be required. The time-consuming steps
for EEG installation – namely, the placement of electrodes and
applying conductive gel (~30 minutes for the high-density
EEG we used), electrode impedance monitoring and correc-

tions, and the multistep and offline data preprocessing – may
make it difficult to use the system in practice. There has
been great progress in recent years on portable and quick-
to-administer EEG systems (e.g. dry, active, low-impedance
electrodes, conductive sponge and hydrogel interfaces [45–
47]). Additionally, fast and real-time preprocessing and arti-
fact removal techniques for EEG have been developed (e.g.,
in [37]). Using fast-to-setup EEG systems, real-time prepro-
cessing techniques, along with SilenceMap (< 3 minutes of
EEG recording), and access to sufficient computational power
(see Section IV D for computation-complexity analysis of Si-
lenceMap), can enable rapid silence localization.

**Introduced error in silence localization by using sym-**
**metric brain models.** Morphological studies of the human
brain have shown cortical asymmetry, and how it is affected

[1] https://www.biosemi.com/faq/skin_preparation.htm

[2] Van Rijn, A.M., Peper, A. and Grimbergen, C.A., 1990. High-quality recording of bioelectric events. *Medical and Biological Engineering and Computing*, 28(5), pp.389-397.

[3] Kamp, A, Pfurtscheller, G, Edlinger, G & Lopes da Silva, F 2005, Technological Basis of EEG Recording. in *Electroencephalography, Basic principles, Clinical applications and related fields*. vol. 5, Lippincott Williams & Wilkins, Philadelphia, pp. 127-138.

[4] Kilicarslan, A., Grossman, R.G. and Contreras-Vidal, J.L., 2016. A robust adaptive denoising framework for real-time artifact removal in scalp EEG measurements. *Journal of neural engineering*, 13(2), p.026013.

[5] Krishnan, A., Kumar, R., Venkatesh, P., Kelly, S. and Grover, P., 2018, July. Low-cost Carbon Fiber-based Conductive Silicone Sponge EEG Electrodes. In *2018 40th Annual International Conference of the IEEE Engineering in Medicine and Biology Society (EMBC)* (pp. 1287-1290). IEEE.

[6] <https://zeto-inc.com>

[7] <http://biosignalgroup.com/product-service/microeeg/>

2-part1) Source analysis: The authors do not mention whether the electrode locations were used during the source analyses as they would likely affect the solutions.

Response to 2-part1) The authors thank the reviewer for raising this important issue. Indeed, electrode locations are used at multiple steps in our SilenceMap algorithm, including in estimation of the leadfield matrix, which is a function of the potential locations of sources (current dipoles in the brain), the locations of sensors (scalp electrodes), and the conductivity ratio of different head layers (i.e., scalp, skull, cerebrospinal fluid (CSF), and brain). While placing the EEG cap for each participant/patient, we did our best to align the cap in a way so that the electrodes are located at the 10-5 standard locations. However, human error in placement of the EEG cap is an important potential source of error. In our future work, we are

planning to address this issue using new methods for guided and repeatable EEG cap placement, e.g., the proposed method in [8].

To address this comment, we have listed the accuracy of electrode locations as an important aspect a potential source of error in Section III, under the “**Electrode locations, a potential source of error**” subtitle:

Lines 584-591:

**Electrode locations, a potential source of error.** We used
the electrode locations in SilenceMap in multiple steps of the
algorithm, including in the estimation of the forward matrix,
which is a function of the electrode locations. In placing the
EEG cap for each patient, we manually adjust the cap’s loca-
tion so that the electrodes are placed in the standard “10-5”
arrangement. This could be improved by using new methods
for guided EEG cap placement [58].

[8] Song, C., Jeon, S., Lee, S., Ha, H.G., Kim, J. and Hong, J., 2018. Augmented reality-based electrode guidance system for reliable electroencephalography. *BioMedical Engineering OnLine*, 17(1), pp.1-10.

2-part2) Likewise, segmentation of MRIs of patients with brain injury is complicated, and in the developing brain is very difficult, particularly in the first 2 years of age (subjects here were older children). This seems to have done by visual inspection with the help of the MNE software. This can be an important source of errors.

Response to 2-part2) To address this comment, we used a new pipeline for the segmentation of the MRI scans in our study. We used *AFNI*, an open source software developed by National Institute of Mental Health (NIMH) and specifically designed scripts, provided to us by Dr Daniel Glen (Computer Engineer, NIH), to segment the structural MRI scans of the patients in our dataset [9,10]. This function is designed to improve the segmentation of the scans in patients with lesions and/or tumors in their brain. It takes into account which hemisphere is intact in the brain, and along with a brain atlas (MNI152_T1_2009c+tlrc was used in our analysis), it strips the skull and segments the MRI scan. After segmentation of the structural MRI scans of the participants in our study using *AFNI*, we extracted the ground truth regions of silence following the steps in Section IV F of our paper. We performed the silence localization for the three pediatric patients in our study using SilenceMap with baseline for both the rest and the visual stimulation condition and compared the performance of the silence localization algorithm based on the ground truth regions of silence extracted using *AFNI* vs. the extracted ground truth using the *FreeSurfer* software. The results are included in the following Figure:

Fig. 14 has been added to the paper in Supplementary Materials, Section F:

		Method	UD $p = 1748, k = 60$ right back bottom			SN $p = 1760, k = 120$ left front bottom			OT $p = 1742, k = 53$ left front bottom		
Localized Regions	Ground Truth	using MRI scans									
	Rest recording	SilenceMap with baseline	 $\Delta COM = 18mm, JI = 0.321, \Delta k = 0.47$	 $\Delta COM = 5mm, JI = 0.490, \Delta k = 0.20$	 $\Delta COM = 18mm, JI = 0.361, \Delta k = 0.13$						
	Visual recording		 $\Delta COM = 22mm, JI = 0.353, \Delta k = 0.47$	 $\Delta COM = 8mm, JI = 0.407, \Delta k = 0.30$	 $\Delta COM = 17mm, JI = 0.346, \Delta k = 0.06$						

FIG. 14. Performance of SilenceMap with baseline, for both the rest and the visual stimulation condition, based on the ground truth regions of silence extracted using the open source *AFNI* software: the first row shows the extracted ground truth regions of silence (red regions) overlaid on the resected cortical region of three patients based on their symmetric brain models extracted from the structural MRIs (see the MRI scans in Fig. 2); the second and third rows show the performance in localization of SilenceMap with baseline, based on the Rest and Visual recordings respectively, through both visual illustration (red regions) and using performance metrics of center-of-mass (COM) distance (ΔCOM), Jaccard Index (JI), and size error (Δk). p is the total number of sources in each brain model, and k is the size of ground truth region of silence. There is a slight reduction in the silence localization performance using the *AFNI* software in comparison with the results using the *FreeSurfer* software (see Fig. 3 in our paper)

Based on these results, there is a slight change (a small reduction) in the silence localization performance using the *AFNI* software in comparison with the results using the *FreeSurfer* software (see Fig. 3 in our paper). This confirms that the reported pipeline for segmentation of the MRI scans in our paper is performing reasonably well and is not a significant source of error in silence localization using the scalp EEG.

To better address this comment, we now briefly mention the results in Section III, under the subtitle “The effect of error in the structural segmentation of MRI”:

Lines 556-583:

**Effect of error in the structural segmentation of MRI.**
Segmentation of structural MRI scans for patients with brain
injuries is a complicated task [53–55], and using standard
structural segmentation techniques (e.g., methods available in
*FreeSurfer*) can be a potential source of error in silence local-
ization. Standard segmentation methods use anatomical priors
extracted from manually or semi-automatically annotated at-
lases of the healthy brain [55]. However, the anatomy of the
damaged brain, especially following severe injury or large re-
section, diverges significantly from the anatomy of a healthy
brain [53]. To address this, we used an open-source software,
*AFNI* [56, 57], which is designed to improve the segmentation
of the scans in patients with brain lesions and/or tumors. Ad-
ditionally, we used specifically designed scripts, provided to
570 us by Dr. Daniel Glen (Computer Engineer, NIH), to segment
the structural MRI scans of the patients in our dataset [56, 57].

We compared the performance of the SilenceMap for the pa-
tients in our study with the ground truth regions of silence
extracted using *AFNI* vs. the extracted ground truth using
*FreeSurfer* (Supplementary Materials, Section F, contains the
results using *AFNI*). Based on these results, there is a slight
change (a small reduction) in the silence localization per-
formance using *AFNI* in comparison with the results using
*FreeSurfer* (Fig. 3 in Section II and Fig. 14 in Supplementary
Materials, Section F). This suggests that standard techniques
(used in *FreeSurfer*) perform reasonably well in MRI segmen-
tation for our participants, and do not contribute significantly
to the silence localization error.

We have also included these results in the Supplementary Materials, Section F of our paper, with the subtitle "Silence localization based on the structural segmentation of MRI using *AFNI* software":

Lines 1940-1961:

**F. Silence localization based on the structural segmentation of**
**MRI using *AFNI* software**

As is explained in Section III, we used the open source *AFNI* soft-
ware [56, 57] to explore the effect of error in the structural segmen-
tation of MRI on the silence localization. *AFNI* takes into account
which hemisphere is intact in the brain, and along with a brain atlas
(MNI152_T1_2009c+tlrc was used in our analysis [57]), it strips
the skull and segments the MRI scan. Following the steps in Sec-
tion IV F, the ground truth regions of silence are extracted from the
processed MRI scans using *AFNI*. We conducted the silence local-
ization for the three pediatric patients in our study using SilenceMap
with baseline, for both the rest and the visual stimulation condition,
based on the ground truth regions of silence extracted using *AFNI*.
The results are included in Fig. 14. Based on these results, there
is a slight change (small reduction) in the silence localization per-
formance (5mm, 3mm, and 7mm higher distance error (ΔCOM) for
UD, SN, and OT respectively) using the *AFNI* software in compari-
son with the results using the *FreeSurfer* software (see Fig. 3 in our
paper). This confirms that the reported pipeline for segmentation of
the MRI scans in our paper using the *FreeSurfer* software is perform-
ing reasonably well and is not a significant source of error in silence
localization using the scalp EEG.

[9] Joachims, T., 1998. *Making large-scale SVM learning practical* (No. 1998, 28). Technical Report.

[10] Cox, R.W., 1996. AFNI: software for analysis and visualization of functional magnetic resonance neuroimages. *Computers and Biomedical research*, 29(3), pp.162-173.

3-part1) Volume conductivity. Volume conductivity ratios can greatly affect the source localization. What was the basis for selecting [1,.067,5,1] as the estimates in the source analysis (forward matrix) and the simulated datasets?

Response to 3-part1) The authors thank the reviewer for this important and critical comment. Regarding the assumptions on the brain-to-skull conductivity ratio (BSCR), there is no consensus in the literature [11]. The BSCR was first estimated to be 80 in [12,13]. This BSCR was widely accepted and used by researchers. However, in the past twenty years the value of BSCR has been the subject of much debate. In [14], Oostendorp et al. estimated a BSCR of 15 using both in vivo and in vitro experiments. In later work, Lai et al. [15] used a spherical head model to estimate the human BSCR as 24.8 from simultaneously recorded intra- and extracranial potentials in 5 epilepsy patients. However, in 2006, Zhang et al. [16] further suggested the BSCR to be 18.7 by using simultaneous intra- and extracranial recordings in two epilepsy patients. To address this comment and to further explore the effect of BSCR on the silence localization performance, we repeated the silence localization experiments for data from both the visual and rest conditions using the leadfield matrices calculated based on the largest, and widely used BSCR of 80. We compared the results with the silence localization performances reported in Fig. 3 of our paper, where we have used a BSCR of 15 (the smallest BSCR reported in the literature supported by in vitro, as well as in vivo experiments [14]). The following figure summarizes the performance of our SilenceMap algorithm using BSCR of 80:

Fig. 15 was added to the paper in Supplementary Materials, Section G:

		Method	UD $p = 1740, k = 60$			SN $p = 1758, k = 120$			OT $p = 1744, k = 55$		
			right	back	bottom	left	front	bottom	left	front	bottom
Localized Regions	Ground Truth	using MRI scans										Rest recording	SilenceMap with baseline										Visual recording		$\Delta COM = 6mm, JI = 0.403, \Delta k = 0.32$	$\Delta COM = 6mm, JI = 0.508, \Delta k = 0.37$	$\Delta COM = 12mm, JI = 0.494, \Delta k = 0.09$						
														$\Delta COM = 23mm, JI = 0.338, \Delta k = 0.42$	$\Delta COM = 5mm, JI = 0.500, \Delta k = 0.38$	$\Delta COM = 27mm, JI = 0.223, \Delta k = 0.09$						

FIG. 15. Performance of SilenceMap with baseline, for both the rest and the visual stimulation condition, based on a large brain-to-skull conductivity ratio (BSCR) of 80: the first row shows the extracted ground truth regions of silence (red regions) overlaid on the resected cortical region of three patients based on their symmetric brain models extracted from the structural MRIs (see the MRI scans in Fig. 2); the second and third rows show the performance in localization of SilenceMap with baseline, based on the Rest and Visual recordings respectively, through both visual illustration (red regions) and using performance metrics of center-of-mass (COM) distance (ΔCOM), Jaccard Index (JI), and size error (Δk). p is the total number of sources in each brain model, and k is the size of ground truth region of silence. The results show only small difference in the silence localization performance using a large BSCR of 80 in comparison with the results using a small BSCR of 15 (see Fig. 3 in our paper)

Based on these results, there is only a small difference between the performance of the SilenceMap with baseline using BSCR of 15 vs. 80, in both the visual and rest recordings, which confirms the robustness of our algorithm to the potential error in the estimated forward matrix due to the assumptions on the conductivity ratios.

To better address this comment, we have briefly mentioned the results in Section III, under the subtitle “Effects of brain-to-skull conductivity ratio”:

Lines 592-597:

**Effects of brain-to-skull conductivity ratio.** We also ex-
plored the effect of different assumptions for the brain-to-skull
conductivity ratio (BSCR) on the performance of SilenceMap.
Our results (Supplementary Materials, Section G includes re-
sults and further discussion) confirm the robustness of our al-
gorithm to imprecise knowledge of BSCR.

And we have included these results in the **Supplementary Materials, Section G** of our paper, under subtitle “**The effect of brain-to-skull conductivity ratio on the localization of the silence**”:

Lines 1963-1990:

**G. The effect of brain-to-skull conductivity ratio on the**
**localization of the silence**
Regarding the assumptions on the brain-to-skull conductivity ratio
(BSCR), there is no consensus in the literature [98]. The BSCR was
first estimated to be 80 in [99, 100]. This BSCR was widely accepted
and used by researchers. However, in the past twenty years the value
of BSCR has been the subject of much debate. In [87], Oostendorp
et al. estimated a BSCR of 15 using both in vivo and in vitro exper-
iments. In later work, Lai et al. [101] used a spherical head model
to estimate the human BSCR as 24.8 from simultaneously recorded

intra- and extracranial potentials in 5 epilepsy patients. However, in
2006, Zhang et al. [102] further suggested the BSCR to be 18.7 by
using simultaneous intra- and extracranial recordings in two epilepsy
patients. To address this comment and to further explore the effect
of BSCR on the silence localization performance, we conducted the
silence localization for the three pediatric patients in our study using
SilenceMap with baseline, for both the rest and the visual stimulation
condition, with the forward matrices (\mathbf{A} in equation (5)) calculated
based on the largest, and widely used BSCR of 80. The results are
included in Fig. 15. We compared the results with the silence local-
ization performances reported in Fig. 3 of our paper, where we have
used a BSCR of 15 (the smallest BSCR reported in the literature sup-
ported by in vitro, as well as in vivo experiments [87]). Based on the
results, there is only a small difference between the performance of
the SilenceMap with baseline using BSCR of 15 vs. 80, in both the
visual and rest recordings, which confirms the robustness of our al-
gorithm to the potential error in the estimated forward matrix due to
the assumptions on the conductivity ratios.

[11] Wang, G. and Ren, D., 2013. Effect of brain-to-skull conductivity ratio on EEG source localization accuracy. *BioMed research international*, 2013.

[12] Rush, S. and Driscoll, D.A., 1968. Current distribution in the brain from surface electrodes. *Anesthesia & Analgesia*, 47(6), pp.717-723.

[13] Cohen, D. and Cuffin, B.N., 1983. Demonstration of useful differences between magnetoencephalogram and electroencephalogram. *Electroencephalography and clinical neurophysiology*, 56(1), pp.38-51.

[14] Oostendorp, T.F., Delbeke, J. and Stegeman, D.F., 2000. The conductivity of the human skull: results of in vivo and in vitro measurements. *IEEE transactions on biomedical engineering*, 47(11), pp.1487-1492.

[15] Lai, Y., Van Drongelen, W., Ding, L., Hecox, K.E., Towle, V.L., Frim, D.M. and He, B., 2005. Estimation of in vivo human brain-to-skull conductivity ratio from simultaneous extra-and intra-cranial electrical potential recordings. *Clinical neurophysiology*, 116(2), pp.456-465.

[16] Zhang, Y., Van Drongelen, W. and He, B., 2006. Estimation of in vivo brain-to-skull conductivity ratio in humans. *Applied physics letters*, 89(22), p.223903.

3-part2) Do the author neglect the boundaries between intact and resected brain tissues? How do they treat the forward matrices? how would a tumor or diffuse cortical lesion affect their algorithm?

Response to 3-part2) This is a very important point in silence localization. Indeed, we neglect the effect of boundaries between the intact and the resected brain tissue. In our analysis, the boundary is considered to be active and healthy brain tissue. This appears to be a reasonable assumption in resections, and maybe also in cases where there is a sharp boundary between the dead and the healthy tissue, e.g. a “really dead” lesion on the brain. However, this is not true in diffuse lesions and/or tumors in the brain, where the boundary effect plays an important role in the silence localization. We are planning to investigate the boundary effect in tumors and diffuse cortical lesions for silence localization in our future works.

To better address this comment, we have included this as a limitation and future direction of our algorithm in Section III, under the subtitle “Limitations and future directions.”:

Lines 620-631:

lenceMap for smaller and deeper regions of silence. (iv) We
do not consider the effect of the boundary between the intact
and the resected brain tissue. In our analysis, the boundary is
considered to be active and healthy brain tissue. This appears
to us to be a reasonable assumption in resections, and maybe
in other cases where there is a sharp boundary between the
dead and the healthy tissue. However, this is not true in dif-
fuse lesions and/or tumors in the brain, where the boundary
effect plays an important role in the silence localization. In-
vestigation of the boundary effect in tumors and diffuse corti-
630 cal lesions for silence localization is another future direction
for our work. (v) The proposed algorithm is designed to lo-

4) Cross-validation: The authors present findings from 3 children with lobectomy and eeg simulations. The authors tested their algorithm with patients with brain resections only. It would have been useful to show the method in a patients with cortical or subcortical stroke, or tumor.

Response to 4) The authors thank the reviewer for this comment and suggestion. We are very interested in testing the performance of SilenceMap on patients with smaller and deeper regions of silence, e.g., cortical or subcortical stroke, cortical laser ablations, and tumors. Unfortunately due to the COVID-19 situation, we have not been able to recruit more patients for our study at this time. We leave this part for future work when we are able to recruit more patients.

We have mentioned this future direction in Section III, under the subtitle “Limitations and future directions”:

Lines 616-620:

silence, SilenceMap needs to be improved. (iii) We plan to
extend our work to examine silence localization in individuals
with etiologies other than resection. We believe that further
improvements and modifications might be needed to use Si-
lenceMap for smaller and deeper regions of silence. (iv) We

Lines 631-637:

for our work. (v) The proposed algorithm is designed to lo-
calize stationary regions of silence. Designing an algorithm
to track and localize the evolving regions of silence, e.g. for
CSD propagation, tumor or lesion expansion, is an impact-
ful future direction. In these applications, previous recordings,
(*with* silence) may be used as a baseline for localization and,
tracking the evolution. (vi) Finally, there might be changes in,

5) The authors claim that their proposed method permit 'rapid detection and localization of regions of silence ...' ; however, it seems that this multi-steps method would depend on computational resources and may not be particularly fast. Can you please some computational metrics for the algorithm (computers used, % of FP operations, processing time, memory, etc)

Response to 5) In this study, we have used a cluster with two Intel(R) Xeon(R) CPU E5-2640 v4 @ 2.40GHz processors, which have the total number of physical cores of 20, and the computational capability of ~73.8 Gflops/sec. Using this cluster, each iteration of the algorithm, including the steps for low-resolution (882 sources) and high-resolution (1744 sources) cortical model, takes ~13.4 minutes to complete, where we have used standard toolboxes in MATLAB (R2019a), and the CVX MATLAB package (v2.1). According to the Real PSD simulation results, for a region of silence with 50 silent sources (out of 1744 total brain sources), the algorithm converges in only 2.93 ± 0.05 iterations (averaged over 90 regions of silence), which takes around ~38 minutes to complete. Throughout the process, the memory usage did not exceed 2.9 GBs. The SilenceMap algorithm is not optimized for the real-time silence localization, and we performed all of the data analyses offline.

To better address this comment, we have also conducted a computation-complexity analysis for SilenceMap (i.e., how does the number of operations scale with the resolution of the cortical models used and the value of other parameters used in the algorithm). The bottleneck of time complexity among the steps in our algorithm is the high-resolution convex optimization (see equation 36 in our paper). This is classified as a convex quadratically constrained quadratic program (QCQP). However, the quadratic constraints in (36) are all scalar and each can be rewritten in forms of two linear constraints. This reduces the problem to a convex quadratic program (QP), which can be solved either using semidefinite programming (SDP) [14] or second-order cone programming (SOCP) [15]. However, it is much more efficient to solve the QPs using SOCP rather than using the general solutions for SDPs [16,17]. Following the steps in [17], we can write our QP in (36) as a SOCP with $\nu = 2p + 2\phi + 1$ constraint of dimension

one, and one constraint of dimension $p + 1$. We follow the notations in the paper, where p is the number of sources in the brain and ϕ is the number of selected electrodes in (33). Using the interior-point methods, the time complexity of each iteration is $O(p^2 (v + p + 1)) \sim O(p^3)$, where the number of iterations for the optimizer is upper bounded by $O(\sqrt{v + 1}) = O(\sqrt{2p + 2\phi + 2})$ [17]. Therefore, the convex spectral clustering (CSpeC) framework for high resolution (see 36 in the paper) has the worst case time complexity of $\sim O(p^{3.5})$. Similarly, the low-resolution CSpeC framework (see equation (18) in our paper), has the same time complexity of $\sim O(p^{3.5})$, since it only has 2ϕ less linear constraints, in comparison with the QP version of (36). It is important to mention that this time complexity is calculated without considering the sparsity of the graph Laplacian matrix (L). Exploiting such sparsity may reduce the computational complexity of solving the equivalent SOCP for our CSpeC framework [18]. The other steps of the SilenceMap algorithm have lower degrees of polynomial time complexity (e.g., the least square solution in equation (34) with time complexity of $\sim O(2^2 \cdot \phi)$, where $\phi \ll p$). Therefore, the general time complexity of the SilenceMap algorithm is $O(itr_{ref} \cdot (p^{3.5} + itr_{conv} \cdot itr_k \cdot itr_\lambda(p^{3.5}))) \sim O(itr_{ref} \cdot itr_{conv} \cdot itr_k \cdot itr_\lambda(p^{3.5}))$, including the number of iterations for finding the optimal regularization parameters (itr_λ iterations for finding λ^* and itr_k iterations for finding k^*), the required number of iterations for convergence of the SilenceMap to a region of silence in the high-resolution step (itr_{conv}), and the number of iterations to find the best reference electrode (itr_{ref}).

It is worth mentioning that the time required to run the SilenceMap algorithm depends on the resolution of the search grids for the parameters used in the algorithm (see Table IV in Supplementary Materials, Section E of our paper), the resolution of the cortical models used, and the convergence criterion defined (see equation (29) in our paper). We acknowledge that there is room for improvement of the implementation and the algorithm itself to obtain a faster silence localization tool, e.g., by exploiting the sparsity of the graph Laplacian matrix in the CSpeC framework, parallelizing the iterations of the algorithm and distributing the computations across multiple processing units, and exploring lower-cost clustering methods. These are beyond the scope of the current work and we leave them for the future work.

We have included the information discussed above as a separate subsection D in Section III, with the title “**Time complexity of SilenceMap**”.

Lines 1066-1117:

D. Time complexity of SilenceMap

The bottleneck of time complexity among the steps in our algorithm is the high-resolution convex optimization (see equation (36) in Section IV B). This is classified as a convex quadratically constrained quadratic program (QCQP). However, the quadratic constraints in (36) are all scalar and each can be rewritten in forms of two linear constraints. This reduces the problem to a convex quadratic program (QP),

which can be solved either using semidefinite programming
 (SDP) [65] or second-order cone programming (SOCP) [66].
 However, it is much more efficient to solve the QPs using
 SOCP rather than using the general solutions for SDPs [67,
 68]. Following the steps in [68], we can write our problem
 in (36) as a SOCP with $v = 2p + 2\phi + 1$ constraint of dimension
 one, and one constraint of dimension $p + 1$, where p is the
 number of sources in the brain and ϕ is the number of selected
 electrodes in (33). Using the interior-point methods, the time
 complexity of each iteration is $\mathcal{O}(p^2(v + p + 1)) \approx \mathcal{O}(p^3)$,
 where the number of iterations for the optimizer is upper
 bounded by $\mathcal{O}(\sqrt{v+1}) = \mathcal{O}(\sqrt{2p+2\phi+2})$ [68]. There-
 fore, the convex spectral clustering (CSpeC) framework for
 high resolution (see equation (36)) has the worst case time
 complexity of $\mathcal{O}(p^{3.5})$. Similarly, the low-resolution CSpeC
 framework (see equation (18)), has the same time complex-
 ity of $\mathcal{O}(p^{3.5})$, since it only has 2ϕ less linear constraints, in
 comparison with the QP version of (36). It is important to
 mention that this time complexity is calculated without consid-
 ering the sparsity of the graph Laplacian matrix (L) de-
 fined in (23). Exploiting such sparsity may reduce the com-
 putational complexity of solving the equivalent SOCP for our
 CSpeC framework [69]. The other steps of the SilenceMap
 algorithm have lower degrees of polynomial time complexity
 (e.g., the least square solution in equation (34) with time com-
 plexity of $\mathcal{O}(2^2\phi)$, where $\phi \ll p$). Therefore, the general time
 complexity of the SilenceMap algorithm is $\mathcal{O}(itr_{ref}(p^{3.5} +$
 $itr_{conv}.itr_k.itr_\lambda(p^{3.5}))) \approx \mathcal{O}(itr_{ref}.itr_{conv}.itr_k.itr_\lambda(p^{3.5}))$, in-
 cluding the number of iterations for finding the optimal reg-
 ularization parameters (itr_λ iterations for finding λ^* in (25),
 and itr_k iterations for finding k^* in (26)), the required number
 of iterations for convergence of the SilenceMap to a region
 of silence in the high-resolution step (itr_{conv}), and the num-
 ber of iterations to find the best reference electrode (itr_{ref}). It
 is worth mentioning that time required to run the SilenceMap
 algorithm depends on the resolution of the search grids for
 the parameters used in the algorithm (see Table IV in Supple-
 mentary Materials, Section E), the resolution of the cortical
 models used, and the convergence criterion defined (see equa-
 tion (29)). We acknowledge that there is room for improve-
 ment of the implementation and the algorithm itself to obtain
 a faster silence localization tool, e.g., by exploiting the spar-
 sity of the graph Laplacian matrix in solving (36), paralleliz-
 ing the iterations of the algorithm, and exploring lower-cost
 clustering methods, and this is left for the future work.

- [14] Diamond, S. and Boyd, S., 2015. Convex optimization with abstract linear operators. In *Proceedings of the IEEE International Conference on Computer Vision* (pp. 675-683).
- [15] Alizadeh, F. and Goldfarb, D., 2003. Second-order cone programming. *Mathematical programming*, 95(1), pp.3-51.
- [16] Cai, Z. and Toh, K.C., 2006. Solving second order cone programming via a reduced augmented system approach. *SIAM Journal on Optimization*, 17(3), pp.711-737.
- [17] Lobo, M.S., Vandenberghe, L., Boyd, S. and Lebret, H., 1998. Applications of second-order cone programming. *Linear algebra and its applications*, 284(1-3), pp.193-228.
- [18] Sheen, H. and Yamashita, M., 2020. Exploiting aggregate sparsity in second-order cone relaxations for quadratic constrained quadratic programming problems. *Optimization Methods and Software*, pp.1-19.

6) Functional connectivity: Even at rest, but particularly during ERP or willed tasks, functional connectivity is an important metric of brain-to-brain communication, which would be expected to be affected by regions of 'silence'. I think they could provide some additional validation when applied to both real and simulated EEG data.

Response to 6) Thank you so much for this comment. We agree that there might be changes in the functional connectivity of the brain because of the region of silence, and that these changes could provide useful additional validation, and might also be interesting in its own right. This can be used for applications such as prediction of diaschisis (remote effects of a resection). This is beyond the scope of the current study but a worthwhile direction to pursue. In our current work, we have extracted the ground truth of the regions of silence based on the structural MRI scans (see Fig. 2), which is not affected by the change in the functional connectivity, and it is a reliable ground truth for assessing the performance of the silence localization.

To better address this comment, we have included this as a future direction of our work in Section III, subtitle "Limitations and future directions":

Lines 637-642:

tracking the evolution. (vi) Finally, there might be changes in
 the functional connectivity of the brain because of the region
 of silence. Understanding/estimating these can be important
 in applications such as prediction of diaschisis (remote effects
 of a resection) or other, wide scale changes in signal propaga-
 tion between regions of cortex.

7) The main goal of the paper, according to the authors, is the localization of the region of silence of patients with lobectomies. Can the authors comment on the time, frequency, time-frequency characteristics of the EEG that define this region in the real datasets of the patients? can you please compare them to the simulated EEG?

Response to 7) We thank you for this important comment on the time-frequency characteristics of the EEG signals when there is a region of silence in the brain. To address this comment, we

have explored the effect of power spectral density (PSD) profile of the neural sources on the performance of our SilenceMap method (hence incorporating the time and frequency characteristics). In the simulated data in our paper, we simulated the normal brain activities (i.e., outside the regions of silence) based on an *i.i.d.* assumption over (discrete) time and identical distribution assumption over space with an exponential decay profile for the covariance matrix. This results in a flat PSD for non-silent sources in the brain (Fig. 10), and, consequently a flat PSD profile for the simulated EEG signal at each electrode on the scalp (instead of the $1/f$ behavior observed in practice). In this revision, we extracted a general shape of PSD for the normal brain activities based on a real recorded electrocorticography (ECoG) dataset used in [19] and available through the open-source library in [20] (see Supplementary Materials, Section D for more details, which included at the end of this response as well). This general shape of PSD (Fig. 8, red curve) results from averaging over the PSDs of the recordings from 62 ECoG electrodes placed around the frontotemporal region of an epileptic patient in [20]. Based on this average PSD, we designed an FIR filter (Fig. 9, see Supplementary Materials, Section D for more details). This filter is then applied on the simulated non-silent source activities (generated initially with a flat PSD, see Fig. 10), which results in a simulated signal which has a PSD (Fig. 11) similar to the PSD extracted from the real recorded ECoG signals in the brain, called Real PSD hereinafter. We still assume the signals are identically distributed over the cortical space, but with spatial and temporal dependency profiles extracted from the real recordings of the brain. Following the steps explained in Section IV G of our paper, we obtain the electrical activities of scalp EEG electrodes based on the simulated signals in the brain. As shown in Fig. 12, the general shape of PSD for the simulated EEG signals based on the Real PSD is close to the PSD of a real recorded EEG signal from a patient with a region of silence in the brain (see Fig. 13, showing the PSD of Rest recordings for patient OT in this study, bandpass filtered in the [1,100] Hz interval). To explore the effect of using the Real PSD, instead of a flat PSD, on the performance of SilenceMap, we repeated the simulation experiment in Section II, “Simulation” subsection of our paper based on the simulated Real PSD signals. The results are included in Table I of our paper, which is included here as well:

TABLE I. Simulation experiment results ($SNR_{avg} = 9dB, k = 50$)

Algorithms	ΔCOM (mm)	JI	Δk	CR
modified MUSIC	60 ± 3.5	0.09 ± 0.012	2.84 ± 0.067	-
modified MNE	82 ± 2.2	0.01 ± 0.002	6.08 ± 0.036	-
modified sLORETA	54 ± 2.2	0.04 ± 0.002	9.62 ± 0.124	-
SilenceMap (flat PSD)	12 ± 0.7	0.50 ± 0.017	0.31 ± 0.023	0.98
SilenceMap (Real PSD)	13 ± 0.5	0.52 ± 0.015	0.28 ± 0.022	0.99

Based on these results, SilenceMap shows almost the same performance for the flat and Real PSDs. Why do the results not change substantially? To understand this, we looked at the effect of changing the (temporal) PSD on the *spatial* correlation for the simulated brain sources. We observe that it is indeed expected that changing this PSD does not affect the spatial correlations (as we next discuss), and hence the localization, which only depends on spatial correlations (with sufficient data) is unaffected.

We assume an exponential decay profile for the source covariance matrix (see equation (9) in Section IV A of our paper), which is consistent with our assumption in the flat PSD simulations. In the Real PSD simulations, the cross correlation coefficients of the brain source activities are being changed through the filtering step. This change can be explored by looking at the filtering process in the time domain. Let's assume $S_i(\mathbf{m})$ is the simulated signal, using the flat PSD, at the i^{th} source and time point m , and $h(\mathbf{m})$ is an FIR filter:

$$S'_i(\mathbf{m}) = (h * S_i)(\mathbf{m}) = \sum_{n=0}^{N-1} h(n) S_i(\mathbf{m} - n) \quad (1)$$

where $S'_i(\mathbf{m})$ is the filtered source signal with the Real PSD, and N is the length of the FIR filter. The cross correlation function of $S'_i(\mathbf{m})$ at lag l is as follows:

$$\begin{aligned} R_{s',i,j}(l) &= E[S'_i(\mathbf{m}) \overline{S'_j(\mathbf{m} + l)}] = E\left[\sum_{n=0}^{N-1} h(n) S_i(\mathbf{m} - n) \cdot \overline{\sum_{q=0}^{N-1} h(q) S_j(\mathbf{m} + l - q)} \right] \\ &= \sum_{n=0}^{N-1} \sum_{q=0}^{N-1} h(n) \overline{h(q)} E[S_i(\mathbf{m} - n) \overline{S_j(\mathbf{m} + l - q)}] \\ &= \sum_{n=0}^{N-1} \sum_{q=0}^{N-1} h(n) \overline{h(q)} R_{s,i,j}(n + l - q) \end{aligned} \quad (2)$$

Where $R_{s,i,j}(n + l - q)$ is the cross correlation of simulated source i and j with flat PSD, which implies independence over time points:

$$R_{s,i,j}(l) = C_{s,i,j} \delta(l) \quad (3)$$

Where $\delta(l)$ is the unit sample function with value 1 at $l = 0$ and value 0 elsewhere. Based on the equality in (3) we can rewrite the equation (2) as follows:

$$R_{s',i,j}(l) = \sum_{n=0}^{N-1} h(n) \overline{h(l+n)} R_{s,i,j}(0) = \rho_h(l) R_{s,i,j}(0) \quad (4)$$

where, $\rho_h(l)$ is the autocorrelation of the FIR filter h with lag l . Based on the equation (4), after filtering the flat PSD signals $S(\mathbf{m})$, the resulting signals $S'(\mathbf{m})$ have a covariance matrix at zero lag, which is only a scaled version of C_S by a constant $\rho_h(0)$. In addition, some non-zero lag correlations show up in $S'(\mathbf{m})$ which means that unlike the flat PSD signals, the simulated signals using the Real PSD are not independent over time. In our SilenceMap method we don't assume temporal independence for the source signals, which explains why there is no silence

localization performance reduction in the Real PSD experiments. However, due to the temporal dependence, a larger number of time points are required for the variance/covariance estimations in our algorithm to achieve the same performance as the flat PSD results. In Table I above, we have used 100000 time points in both the flat and the Real PSD simulations.

Fig. 8-13 have been added to the paper in Supplementary Materials, Section D:

FIG. 8. Power spectral density (PSD) of normal brain activities, averaged over 62 electrodes, based on a real recorded electrocorticography (ECoG) dataset used in [33] and available through the open-source library in [34]. The blue and the orange lines are the ECoG average PSD, with and without the power line noise components respectively (see Supplementary Materials, Section D for more details).

FIG. 9. Magnitude of a linear phase finite impulse response (FIR) filter, with the magnitude of equal to the square root of the average power spectral density (PSD) shown in Fig. 8.

FIG. 10. Power spectral density (PSD) of simulated non-silent source activities with independent time points (flat PSD simulations).

FIG. 11. “Real PSD” brain signals simulations: Power spectral density (PSD) of simulated non-silent source activities based on a general shape of PSD extracted from the normal brain activities based on a real recorded electrocorticography (ECoG) dataset.

FIG. 12. Power spectral density (PSD) of simulated scalp electroencephalography (EEG) signals based on the “Real PSD” simulated signals in the brain (see Fig. 11).

FIG. 13. Power spectral density (PSD) of differential scalp signals in a pediatric patient with resection (OT, see Table II), during the Rest recordings. The signals are bandpassed in the frequency interval of $[1, 100]Hz$.

To address this comment, we have now included the results of Real PSD simulations in Table I in the paper, and the above discussion on the effect of Real PSD on silence localization performance in Section II under the “Simulations” subtitle:

Lines 226-237:

for the EEG to record their electrical activity [13]. The non-
silent sources are assumed to have an identical distribution
and correlation across space, and identical distribution over
time. To explore the effect of different assumptions for the
time-frequency characteristics of neural sources on the silence
localization task, we considered two scenarios in the simula-
tions: (i) a flat power spectral density (PSD) profile for the
activities non-silent sources, and (ii) a “Real PSD” profile,
which is extracted from an open source electrocorticography
(ECoG) dataset used in [33] and available through the open-
source library in [34]. The detailed steps for the simulation
are available in the Methods (see Section IV G). For the Si-

Lines 255-265:

the modified source localization algorithms. In addition, Si-
lenceMap showed almost the same performance for the flat
and Real PSDs. This observation can be explained by look-
ing at the effect of changing the (temporal) PSD on the *spa-*
*tial* correlation for the simulated brain sources (the detailed
discussion on this is available in Section IV G). The simu-
lated dataset is based on the identical distribution assumption
of brain sources. However, this assumption does not appear
to hold in the real dataset, where SilenceMap *with* baseline
performs significantly better compared to SilenceMap with-
out baseline (see Fig. 3). The list of all parameters and their

In addition, the details of the Real PSD simulations are included in the Method Section VI G under the subtitle “Real PSD simulations”:

Lines 1307-1342:

symmetric brain model. (v) Finally, the calculated forward
matrix \mathbf{A} is used to simulate the scalp EEG signals following
the equation (7).
**Real PSD simulations:** To explore the effect of PSD pro-
file of the non-silent neural sources on the performance of Si-
lenceMap, we simulate source activities with “Real PSD” fol-
lowing these steps: (i) First, we extract a general shape of PSD
for the normal brain activities based on a real recorded elec-
trocorticography (ECoG) dataset used in [33] and available
through the open-source library in [34]. This general shape
of PSD results from averaging over the PSDs of the ECoG
recordings of an epileptic patient in [34] (see Fig. 8 and more
details on the reprocessing and the average PSD extraction
in Supplementary Materials, Section D). (ii) In the next step,

we design a linear phase finite impulse response (FIR) filter
 (Fig. 9 in Supplementary Materials, Section D), with the mag-
 nitude equal to the square root of the noiseless average PSD
 shown in Fig. 8 in Supplementary Materials, Section D. (iii)
 Following the steps (i) and (ii) in the previous section (“Flat
 PSD simulations”), we simulate non-silent source activities
 with a flat PSD (instead of the 1/f behavior observed in prac-
 tice, see Fig. 10 in Supplementary Materials, Section D), and
 then apply the designed filter on them. This results in sim-
 ulated signals which have PSDs similar to the PSD extracted
 from the real recorded ECoG signals in the brain, called “Real
 PSD” hereinafter (see Fig. 11 in Supplementary Materials,
 Section D). We assume the signals have identical distribution
 over the cortical space, but with spatial and temporal depen-
 dency profiles extracted from the real recordings of the brain.
 Following the steps (iii) to (v) in the previous section (“Flat
 PSD simulations”), we obtain the electrical signals of scalp
 EEG electrodes based on the Real PSD simulated signals in
 the brain. The general shape of PSD for the simulated EEG
 signals based on the Real PSD (Fig. 12 in Supplementary Ma-
 terials, Section D) is close to the PSD of a real recorded EEG
 signal from a patient with a region of silence in the brain (see
 Fig. 13 in Supplementary Materials, Section D, showing the
 PSD of Rest recordings for patient OT in this study, bandpass
 filtered in the $[1,100]$ Hz interval).

The mathematical details of the effect of different PSDs on the spatial pattern of correlation for non-silent sources are now included in the Method Section VI G under the subtitle “The effect of different PSD profiles on the silence localization performance”:

Lines 1343-1399:

**The effect of different PSD profiles on the silence local-**
 **ization performance:** Based on these results presented in Ta-
 ble I, SilenceMap shows almost the same performance for the
 flat and Real PSDs. Why do the results not change substan-
 tially? To understand this, we looked at the effect of chang-
 ing the (temporal) PSD on the *spatial* correlation for the sim-
 ulated brain sources. We observe that it is indeed expected
 that changing this PSD does not affect the spatial correlations
 (as we next discuss), and hence the localization, which only
 depends on spatial correlations (with sufficient data) is unaf-
 fected.

We assume an exponential decay profile for the source covariance matrix (see \mathbf{C}_s defined in (9)), which is consistent with our assumption in the flat PSD simulations. In the Real PSD simulations, the cross correlation coefficients of the neural source activities are being changed through the filtering step (see step (iii) in “Real PSD simulations”). This change can be explored by looking at the filtering process in the time domain. Let’s assume $S_i(t)$ is the simulated signal, using the flat PSD, at the i^{th} source and time point t , and $h(t)$ is an FIR filter:

$$S'_i(t) = (h * S_i)(t) = \sum_{q=1}^{N_h} h(q)S_i(t - q), \quad (49)$$

where $S'_i(t)$ is the filtered signal with the Real PSD at the i^{th} source, and N_h is the length of the FIR filter h . The cross correlation function of $S'_i(t)$ at lag l is as follows:

$$\begin{aligned} R_{S'_i S'_j}(l) &= E[S'_i(t) \overline{S'_j(t+l)}] \\ &= E \left[\sum_{q=1}^{N_h} h(q)S_i(t - q) \overline{\sum_{r=1}^{N_h} h(r)S_j(t + l - r)} \right] \\ &= \sum_{q=1}^{N_h} \sum_{r=1}^{N_h} h(q) \overline{h(r)} E \left[S_i(t - q) \overline{S_j(t + l - r)} \right] \\ &= \sum_{q=1}^{N_h} \sum_{r=1}^{N_h} h(q) \overline{h(r)} R_{S_i S_j}(q + l - r), \end{aligned} \quad (50)$$

where $R_{S_i S_j}(q + l - r)$ is the cross correlation of simulated signals at the i^{th} and j^{th} sources with flat PSDs, which implies independence over time points:

$$R_{S_i S_j}(l) = c_{s_{ij}} \delta(l), \quad (51)$$

where $\delta(l)$ is the unit sample function with value 1 at $l = 0$ and value 0 elsewhere, and $c_{s_{ij}}$ is the zero-lag cross correlation of brain activities at the i^{th} and j^{th} sources defined in (9). Based on the equality in (51), we can rewrite the equation (50) as follows:

$$R_{S'_i S'_j}(l) = \sum_{q=1}^{N_h} h(q) \overline{h(l+q)} R_{S_i S_j}(0) = \rho_h(l) R_{S_i S_j}(0), \quad (52)$$

where, $\rho_h(l)$ is the autocorrelation of the FIR filter $h(t)$ with
 lag l . Based on the equation (52), after filtering the flat PSD
 signals $\mathbf{S} \in \mathbb{R}^{p \times T}$ with the covariance matrix of $\mathbf{C}_s \in \mathbb{R}^{p \times p}$,
 the resulting signals $\mathbf{S}' \in \mathbb{R}^{p \times T}$ have a covariance matrix at
 zero lag, which is only a scaled version of \mathbf{C}_s by a constant
 $\rho_h(0)$. In addition, some non-zero lag ($l \neq 0$) correlations
 show up in \mathbf{S}' , which means that unlike the flat PSD signals,
 the simulated signals using the Real PSD are not independent
 over time. In our SilenceMap method, we do not assume
 temporal independence for the brain source signals, which ex-
 plains why there is no silence localization performance reduc-
 tion in the Real PSD experiments. However, due to the tempo-
 ral correlations in the Real PSD simulations, a larger number
 of time points are required for the variance and covariance es-
 timations in our algorithm to achieve the same performance
 as the flat PSD results. In Table I, we have used $T = 100000$
 time points for all of the simulations.

And the details of the data analysis for the Real PSD simulations, along with the related figures are included in the Supplementary Materials, Section D with the subtitle “Data analysis and figures for “Real PSD” simulations”:

Lines 1892-1931:

**D. Data analysis and figures for “Real PSD” simulations**

As explained in the step (i) of the “Real PSD” simulations in Sec-
 tion IV G, we extract a general shape of PSD for the normal brain
 activities based on an open source real recorded electrocorticography
 (ECoG) dataset used in [33] and available in the open-source library
 in [34]. This general shape of PSD (Fig. 8) results from averaging
 over the PSDs of the recordings from 62 ECoG electrodes placed
 around the frontotemporal region of an epileptic patient (patient “zt”
 in [34]).

*Ethics statement:* All patients participated in a purely voluntary
 manner, after providing informed written consent, under experimen-
 tal protocols approved by the Institutional Review Board of the Uni-
 versity of Washington (#12193). All patient data was anonymized ac-
 cording to IRB protocol, in accordance with HIPAA mandate. These
 data originally appeared in the manuscript “Rapid online language
 0907 mapping with electrocorticography” published in Journal of Neuro-

1908 surgery: Pediatrics in 2011 [33].

*Data analysis:* A Butterworth IIR filter is used to bandpass filter
the ECoG signals in the frequency interval of $[0.1, 200] Hz$. In addition,
we used the ZapLine method proposed in [95] to remove the
power line noise components at 60, 120, and 180 Hz (see Fig. 8).
ZapLine removes the power line components with minimal degradation
of other frequency components of the multi-channel ECoG
data [95]. The PSDs of the filtered and noise removed signals are
then averaged across the 62 electrodes to extract the general shape
of PSD in Fig. 8 (red curve). Following the steps of the “Real PSD
Simulations” in Section IV G, we design an FIR filter based on this
average PSD (see Fig. 9), apply this filter on the simulated non-silent
source activities (generated initially with a flat PSD, see Fig. 10) to
obtain the “Real PSD” simulated signals for the non-silent sources in
the brain (Fig. 11 shows the PSD of these signals). Finally, the scalp
signals are simulated based on these “Real PSD” signals, which have
PSDs (see Fig. 12) similar to the PSD of a real recorded EEG signal
from a patient with a region of silence in the brain (see Fig. 13, showing
the PSD of Rest recordings for patient OT in this study, bandpass
filtered in the $[1, 100] Hz$ interval). Please note that, in all of the presented
PSD plots in this section, the magnitude of the real recorded
signals are presented in μV , and the amplitude of the simulated signals
are presented in an arbitrary unit $a.u.$, where $\sigma_s = 1 a.u.$ in (9)
for simulations.

[19] Miller, K.J., Abel, T.J., Hebb, A.O. and Ojemann, J.G., 2011. Rapid online language mapping with electrocorticography. *Journal of Neurosurgery: Pediatrics*, 7(5), pp.482-490.

[20] Miller, K.J., 2019. A library of human electrocorticographic data and analyses. *Nature Human Behaviour*, 3(11), pp.1225-1235.

8) The model assumes symmetry across hemispheres. There are many studies that have shown cortical asymmetry and how it is affected by age, disease, etc. The authors should discuss this.

Response to 8) Thank you for this important comment. We have discussed the hemispheric symmetry assumption in SilenceMap with baseline in two parts in the paper:

i) In Section II, subtitle “Validity of hemispheric symmetry assumption in SilenceMap with baseline”, where we explored the approximate hemispheric symmetry assumption of the brain source activities in the healthy parts of the brain. We quantified the hemispheric symmetry of scalp average power in a healthy control participant (DH), in comparison to the three patients who have resected brain regions (UD, SN, and OT). Mean absolute difference of scalp average power (MAD) is reported for each participant (see equation (4) in our paper). The control participant shows significantly smaller MAD compared to the three patients with cortical regions of silence (see Fig. 4 in the paper). This result supports the fact that using the hemispheric baseline is helpful in localization of regions of silence, which are located in either the left or right hemispheres. We also explained the potential sources of hemispheric asymmetry in the neural

activities. As explained in this section, one possible way to improve the performance of the SilenceMap algorithm is to take into account the non-identical distribution of sources in brain (and perhaps use a more realistic model for the source covariance matrix (C_S) and normalize the source contribution measure accordingly.

ii) In Section III, subtitle “Introduced error in the silence localization by using symmetric brain models”, we have discussed in detail the potential source of error in our SilenceMap with baseline algorithm:

ease [48, 49]. In this paper, we used symmetric brain mod-
els of the patients with lobectomy, since the pre-surgery MRI
scans of these patients were not available (which may not
even have been symmetrical in the first instance). Fig. 5
shows the symmetric brain models of UD, SN, and OT, along
with their original models which have resected regions. To
quantify the introduced error in silence localization by using
symmetric models, instead of the original model, we calcu-
lated the average distance of sources/nodes of the intact part
of the hemisphere with a missing section to the correspond-
ing sources/nodes of the other hemisphere (the structurally
preserved hemisphere) mirrored across the longitudinal fis-
sure (see Fig. 5). Following the 3D shape matching approach
in [50], for a specific source/node in the brain hemisphere with
the region of silence, the corresponding source in the mirrored,
hemisphere is defined as the node with the minimum distance,
to that specific source. Based on our calculations, the defined
average distance between the symmetric brain model and the
original brain model is $2.41 \pm 0.055\text{mm}$, $2.50 \pm 0.043\text{mm}$, and
$2.03 \pm 0.044\text{mm}$, for UD, SN, and OT, respectively. We ex-
cluded the resected parts of the brain in calculating the average
distance between the symmetric brain model and the original
brain model in UD, SN, and OT. To make sure this average
distance is not affected by this exclusion of the resected re-
gions, we also calculated this hemispheric distance in three
healthy controls (intact brains) using an open source MRI
database (OASIS-1² [51] OAS1_0004_MR1 (male, 28yr),
OAS1_0005_MR1 (male, 18yr), and OAS1_0034_MR1 (male,

51yr) (see Fig. 5). The average distance between the
symmetric brain model and the original brain model was
$2.33 \pm 0.012\text{mm}$, $2.78 \pm 0.016\text{mm}$, and $2.35 \pm 0.012\text{mm}$, for
OAS1_0004_MR1, OAS1_0005_MR1, and OAS1_0034_MR1,
respectively. In fMRI studies, an acceptable motion and voxel
displacement, especially in scans of children and adolescents,
is typically up to 3mm [52]. Since the average distance of
the symmetric and the original brain models is less than 3mm,
using the symmetric brain model seems to be a reasonable
choice for silence localization.

To better address this comment and emphasize these potential sources of error, we have added the following sentences in Sections II and III:

Line 446-451:

brain sources have non-identical brain activities. Functional
studies of the brain have shown that the sources in the brain do
not have perfectly symmetric activity [42, 43], and this asym-
metry is affected by factors such as age [44], and (ii) the struc-
ture of the brain and the head (scalp, skull, CSF, and brain) is
not perfectly symmetric (see Section III for more discussion),

Line 461-464:

contribution measure accordingly. Another approach to ad-
dress this issue is to use an asymmetric baseline for silence
localization if possible, in which the baseline comes from the
recording of the brain without any region of silence.

Line 514-518:

**Introduced error in silence localization by using sym-**
**metric brain models.** Morphological studies of the human
brain have shown cortical asymmetry, and how it is affected
by different factors such as age, sex, and neurological dis-
ease [48, 49]. In this paper, we used symmetric brain mod-

9) Overall, given the points above, I am not convinced that the proposed method substantially outperforms existing modified source locations methods. This needs to be revisited.

Response to 9) Thank you for this feedback. In the revised version, we have now tried to provide further explanations on why existing modified source localization methods substantially fail (42mm larger average COM distance, 41% less average overlap (JI), and 253% more size error, compared to the SilenceMap algorithm.) In the Section I (Introduction), we included this explanation:

ing [12, 13]. The localization of a region of *silence*, how-
ever, poses additional challenges. The most significant further
challenge is in how the background brain activity is treated:
while it is usually grouped with noise in source localization¹,
it is of direct interest in silence localization where the goal is
to distinguish normal brain activity from abnormal silences.
Thus, in source localization paradigms applied to neurosci-
entific data [14–16], as also for the Event Related potential
(ERP) paradigm [17, 18], scalp EEG signals are aggregated
over event-related trials to average out background brain ac-
tivity and noise, permitting the extraction of the signal activity
that is consistent across trials. Consequently, as we demon-
strate in our experimental results below, classical source lo-
calization techniques, e.g., multiple signal classification (MU-
SIC) [19, 20], minimum norm estimation (MNE) [16, 21–23],
and standardized low resolution brain electromagnetic tomog-
raphy (sLORETA) [24], even after appropriate modifications,
fail to localize silences in the brain (Section IV E details our
modifications on these algorithms).

In addition, in the results section, the performance of the modified state-of-the-art source localization methods are compared with our SilenceMap technique, through rigorous simulations (over 100 different regions of silence), as well as experimental validations. In Section II, subtitle “Comparison of SilenceMap with source localization algorithms.”, we have provided the results of this comparison for the simulated dataset:

lenceMap algorithm (see Section IV). Based on the simula-
tion results in Table I, among the modified source localization
algorithms, sLORETA shows the minimum average COM dis-
tance of 54mm, and MUSIC shows the maximum average
overlap of 9% ($JI = 0.09$), and the minimum average size er-
ror of 284% ($\Delta k = 2.84$). This performance is still poor for the
silence localization task, while SilenceMap shows good per-
formance based on the simulation results in Table I ($\Delta COM =$
12mm, $JI = 0.50$, $\Delta k = 0.31$). Based on these results, source
localization algorithms, even after proper modifications, per-
form poorly in localizing the regions of silence in the brain.

Finally, we have included the detailed description of the modified source localization algorithms, namely, MNE, MUSIC, and sLORETA, along with the step-by-step explanation and justification of each modification, in Section IV (Methods), subsection E: “Modification of source localization algorithms for comparison with the SilenceMap algorithm”.

To better address this comment, we have separated and highlighted the discussion on the fundamental differences of silence localization vs. source localization as a subsection in the introduction (see Section I, subtitle “**Source vs. silence localization**”), and added the following reference to Section IV E in the introduction section:

Lines 76-83:

that is consistent across trials. Consequently, as we demon-
strate in our experimental results below, classical source lo-
calization techniques, e.g., multiple signal classification (MU-
SIC) [19, 20], minimum norm estimation (MNE) [16, 21–23],
and standardized low resolution brain electromagnetic tomog-
raphy (sLORETA) [24], even after appropriate modifications,
fail to localize silences in the brain (Section IV E details our
modifications on these algorithms).

10) Could the authors comment on the relatively small amount of EEG data used? Did they compute localization accuracy as a function of amount of EEG data (say 3 min vs tens of minutes)?

Response to 10) We are happy to respond to the request for comparison EEG data over time. In order to find the minimum required amount of EEG data for the SilenceMap to achieve the best possible localization performance (ΔCOM , ΔK , and JI), we did a grid-search for the length of the EEG signals in the interval of [20,40,80,120,160] seconds, and for each temporal length we quantified the localization performance of SilenceMap. Based on the results, for the participant UD, using only 80s of data showed almost the same localization performance as 160s ($\Delta COM = 17\text{mm}$, $JI = 0.382$, $\Delta K = 0.30$), while 40s of data and less showed significant reduction in the localization performance. For participant SN, the minimum possible amount of data, without compromising the localization performance, is only 40s ($\Delta COM = 9\text{mm}$, $JI = 0.440$, $\Delta K = 0.20$), while for participant OT, any amount of data less than 160s showed reduction in the performance of silence localization. This observation might be due to the noisy EEG recording of OT, as mentioned in Section II, but nevertheless 160s is still a relatively short amount of signal acquisition time. These sets of results suggest that in the case of having a good EEG recording (without significant noise and artifacts), SilenceMap with baseline is able to localize the regions of silence, with only a small amount of data.

The aforementioned description is already available in Section III, subtitle “**SilenceMap can localize the regions of silence with relatively little EEG data**”:

Lines 481-496:

**SilenceMap can localize the regions of silence with rel-**
**atively little EEG data.** As we showed in Section II, Si-
lenceMap successfully localized the regions of silence based
on only 160s of EEG data. Although this is already quite
small, how does SilenceMap perform if we reduce this times-
pan. To understand this, we did a search for the timespan for
[20, 40, 80, 120, 160]s, quantifying the performance for each
timespan. For UD, 80s of data showed almost the same per-
formance as 160s ($\Delta COM = 17mm$, $JI = 0.382$, $\Delta k = 0.30$),
while 40s showed significant reduction. For SN, the minimum
possible amount of data, without compromising the localiza-
tion performance, is only 40s ($\Delta COM = 9mm$, $JI = 0.440$,
$\Delta k = 0.20$), while for OT, this is 160 sec, potentially due to the
noisy EEG recording of OT, as discussed in Section II. Nev-
ertheless, the 160s upper limit is still a relatively short amount
of signal acquisition time.

To highlight this subsection, we added a reference to this section in the introduction, where we have reported the 160s value:

Line 119-126:

of our proposed SilenceMap algorithm. In simulations and
real data analysis, SilenceMap outperformed existing algo-
rithms in localization accuracy for localizing silences in 3 par-
ticipants with surgical resections for management of epileptic
seizures or evacuation of cerebral hematoma. It does so while
using only 160s of EEG signals using 128 electrodes (see Sec-
tion III for more details on finding the minimum amount of
EEG data for localizing silences using SilenceMap).

11) Abstract: The abstract appears to claim more than it is offered. The authors talk about 'a rapid and cost-effective tool to detect and characterize abnormal neural function...' however, the patient group was limited to a small group of subjects with lobectomy only, and it does not yet characterizes abnormal neural function.

Response to 11) As the reviewer correctly notes, we have only tested the SilenceMap's performance for detection and localization of neural silences, which is a subset of neural abnormalities. To address this issue and make the claim in our paper more focused on detection of neural silences, we have changed the first sentence in the abstract as follows:

Abstract

3

4 **A rapid and cost-effective noninvasive tool to detect and characterize neural silences can be of significant**
5 **benefit for the diagnosis and treatment of many disorders.** We propose a novel algorithm, SilenceMap,

In addition, in the introduction section (Section I), we made our claim more clear:

Line 24-27:

24 the sidelines all pose significant hurdles. In this paper, using
scalp electroencephalography (EEG) signals with relatively
little data, we provide theoretical and empirical support for a
novel method for the noninvasive detection of neural silences.

In addition, we tried to motivate our work by the growing utilization of EEG for diagnosis and monitoring of neurological disease such as stroke [19], and concussion [20]:

Lines 36-38:

There has been growing utilization of EEG for diagnosis
and monitoring of neurological disorders such as stroke [6],
and concussion [7]. Common imaging methods for detecting

[19] Erani, F., Zolotova, N., Vanderschelden, B., Khoshab, N., Sarian, H., Nazarzai, L., Wu, J., Chakravarthy, B., Hoonpongsimanont, W., Yu, W. and Shahbaba, B., 2020. Electroencephalography Might Improve Diagnosis of Acute Stroke and Large Vessel Occlusion. *Stroke*, pp.STROKEAHA-120.

[20] Fickling, S.D., Smith, A.M., Pawlowski, G., Ghosh Hajra, S., Liu, C.C., Farrell, K., Jorgensen, J., Song, X., Stuart, M.J. and D'Arcy, R.C., 2019. Brain vital signs detect concussion-related neurophysiological impairments in ice hockey. *Brain*, 142(2), pp.255-262.

12) please provide the data (as in Figure 1) for the other 2 patients in the appendix/supplementary material section.

Response to 12) Thank you for this great suggestion. We included the overview figures for patient SN and OT in the Supplementary Materials (Fig. 16 and 17 respectively), similar to the Fig. 1 for UD on page 2 of our manuscript:

FIG. 16. SilenceMap with baseline algorithm overview based on patient SN's Rest dataset: a) The EEG recording protocol and the locations of scalp electrodes. One of 10 reference electrodes (shown in red) is chosen along the longitudinal fissure for rereferencing against. b) Average power of scalp potentials for different choices of reference electrodes. c) Symmetric brain model of a patient (SN) with left temporal hematoma. d) Steps of the SilenceMap algorithm in a low-resolution source grid. A measure of the contribution of brain sources in the recorded scalp signals ($\hat{\beta}$) is calculated relative to a hemispheric baseline. In the brain colormap, yellow indicates no contribution. A contiguous region of silence is localized based on a convex spectral clustering (CSpeC) framework in the low-resolution grid. e) Steps of the SilenceMap algorithm in a high-resolution source grid. The source covariance matrix (C_S) is estimated through an iterative method, and the region of silence is localized using the CSpeC framework. f) Choosing the best reference electrode to reference against (C_z in this example), which results in minimum scalp power mismatch (ΔPow). The localized region of silence for this patient (SN) has 2mm COM distance (ΔCOM) from the original region, with more than 57% overlap ($JI = 0.570$), and it is 25% larger ($\Delta k = 0.25$).

FIG. 17. SilenceMap with baseline algorithm overview based on patient OT's Rest dataset: a) The EEG recording protocol and the locations of scalp electrodes. One of 10 reference electrodes (shown in red) is chosen along the longitudinal fissure for rereferencing against. b) Average power of scalp potentials for different choices of reference electrodes. c) Symmetric brain model of a patient (OT) with left temporal resection. d) Steps of the SilenceMap algorithm in a low-resolution source grid. A measure of the contribution of brain sources in the recorded scalp signals (β) is calculated relative to a hemispheric baseline. In the brain colormap, yellow indicates no contribution. A contiguous region of silence is localized based on a convex spectral clustering (CSpeC) framework in the low-resolution grid. e) Steps of the SilenceMap algorithm in a high-resolution source grid. The source covariance matrix (C_S) is estimated through an iterative method, and the region of silence is localized using the CSpeC framework. f) Choosing the best reference electrode to reference against (CPz in this example), which results in minimum scalp power mismatch (ΔPow). The localized region of silence for this patient (OT) has 11mm COM distance (ΔCOM) from the original region, with more than 47% overlap ($JI = 0.477$), and it is 36% smaller ($\Delta k = 0.36$).

We included a reference to these figures in Section II (Results) as follows:

Lines 163-166:

163 more details). All steps of SilenceMap, along with the inter-
 164 mediate results for patient UD, are summarized in Fig. 1. We
 165 have included similar overview figures for patients SN and OT
 166 in Supplementary Materials (Fig. 16, and Fig. 17).

13) Please include a table with list of assumptions and potential effect on the neural silence estimate, and possible way to mitigate the assumption.

Response to 13) Thank you for this great suggestion. We had discussed these assumptions in detail in Section IV A. In response to this comment, we have now created a Table of simplification assumptions we have adopted in designing the SilenceMap, the effect of each assumption on the neural silence estimate, and a brief explanation of possible ways to relax these assumptions (see Table III in Section IV A).

TABLE III. List of simplification assumptions and their effect on for silence localization.

Assumption number in Section IV A	Assumption	Effect	Possible ways to relax these assumptions
(ii)	Spatio-temporal independence of additive noise $\tilde{\mathbf{E}}$	It affects the noise variance estimation (see Supplementary Materials, Section B)	Using more realistic assumptions on the general shape of noise PSD (non-flat PSD), and the spatial correlation profile (non-diagonal \mathbf{C}_2), noise variance estimation can be improved.
(iii)	Spatial exponential decay profile for the source covariance matrix \mathbf{C}_s , with identical variances (σ_s^2) for all non-silent sources	It affects the source covariance estimation in SilenceMap (see equation (34))	Using more realistic and data-driven assumptions on the spatial correlation profile of brain sources, as well as estimation of non-identical source variances based on baseline recordings of silences.
(vi)	Contiguity of silent sources as a single region of silence	It affects the design of the CSpEC framework proposed in SilenceMap (see equations (36) and (18))	With the assumption of multiple regions of silence, with different sizes, using methods such as the extension of CSpEC method for multiple clusters in a graph can be used in SilenceMap [62].
(vii)	Silence lies in only one hemisphere	Based on this assumption, we use the hemispheric baseline for silence localization.	This assumption can be relaxed if we have a baseline recording for the regions of silence (e.g., recording of the brain without any silence).
(vii)	Hemispheric symmetry of scalp potentials for regions far from silence	Based on this assumption, we use hemispheric baseline and select a subset of scalp electrodes to estimate the source covariance matrix (see equations (32) and (33)).	This assumption can be relaxed if we have a baseline recording for the regions of silence (e.g., recording of the brain without any silence), and use a non-identical distribution model for the non-silent source activities (see assumption (iii) and its relaxation).

Lines 748-750:

748 The simplification assumptions in this section are summarized
749 in Table III, where we discuss the effect of each assumption,
750 along with possible ways to relax them.

Reviewer #2 and #3 (Remarks to the Author):

The authors have described a novel and interesting algorithm for purposes of localizing regions of brain damage, or regions cortical silence. The paper is very thorough and technically sound. The algorithms and equations are straight-forward and intriguing to read about. I applaud the authors for developing this technique.

Thank you for the positive evaluation of our manuscript.

However, there are a few main issues that needs to be addressed before the manuscript is suitable for publication:

1) The papers claim that the current algorithm is superior to others, and make comparisons to support this idea. The descriptions of these other algorithms is lacking and a discussion specifically on why they don't function as well as the current should be included.

Response to 1) The authors thank the reviewer for this comment. We have fully addressed this comment as the response to Reviewer #1's comment (see Response 9 on page 28 of this response letter).

2) Being able to detect a lack of a signal, rather than the presence unique signal, poses a unique challenge in terms of background noise. The researchers need to overcome background to a great degree to be ensure that a negative signal is being recorded. There was not much detail about this in the manuscript.

Response to 2) This is a very important comment. As we have discussed in Section I (Introduction), under the subtitle "Source vs. silence localization", the fundamental difference between source localization and silence localization lies in the way that the background brain activity is treated:

Lines 65-76:

ing [12, 13]. The localization of a region of *silence*, how-
ever, poses additional challenges. The most significant further
challenge is in how the background brain activity is treated:
while it is usually grouped with noise in source localization¹,
it is of direct interest in silence localization where the goal is
to distinguish normal brain activity from abnormal silences.
Thus, in source localization paradigms applied to neurosci-
entific data [14–16], as also for the Event Related potential
(ERP) paradigm [17, 18], scalp EEG signals are aggregated
over event-related trials to average out background brain ac-
tivity and noise, permitting the extraction of the signal activity
that is consistent across trials. Consequently, as we demon-

Lines 84-88:

In order to not average out the background activity, we es-
timate the contribution of each source to the recorded EEG
across all electrodes. This contribution is measured in an av-
erage power sense, instead of the mean, thereby avoiding can-
celing out the contributions of the background brain activity.

In essence, our proposed method can use a large number of data points to estimate the variances and correlations required for localizing the regions of silence, e.g., estimation of variance of μ_{qt} , the measure of contribution of brain sources to the recorded differential signals from the scalp (see equation (49) in our paper). As an example, 160 seconds of data with the sampling frequency of 512 Hz provides our algorithm with around 81,920 data points to be used, which achieves a very good estimation of the variances and correlations. This helps us significantly improve the signal-to-noise ratio (SNR) over source localization techniques, where they have only a few tens of event-related trials to average over and improve the SNR.

To better address this important comment, we have added the following sentences in Section I:

Lines 93-101:

Our silence localization algorithm, that we refer to as “Si-
lenceMap,” estimates these contributions, and then uses tools
that quantify our assumptions on the region of silence (conti-
guity, small size of the region of silence, and being located in
only one hemisphere) to localize it. **Because of this, another**
**difference arises: silence localization can use a larger number**
**of time points (than typical source localization). E.g., 160 sec-**
**onds of data with the sampling frequency of 512 Hz provides**
**SilenceMap with around 81,920 data points to be used, boost-**
**ing the signal-to-noise ratio (SNR) over source localization**
**techniques, which typically rely on only a few tens of event-**
**related trials to average over and extract the source activity**
**that is consistent across trials.**

3) The brain physiology background component was unclear as well. It appears that the success of the algorithm is dependent on finding a cortical baseline for comparison. This can be done if you know where the damage is but may be challenging otherwise. How do the researchers expect to overcome a case where the damage is diffuse or unknown so that a reliable baseline cannot be established?

Response to 3) Thanks for the comment. In the paper, we provided the explanations on the background brain activity component and how we model this term in our SilenceMap algorithm, as well as in the simulations:

The background brain activity in source localization:

Lines 66-76:

ever, poses additional challenges. The most significant further
challenge is in how the background brain activity is treated:
while it is usually grouped with noise in source localization¹,
it is of direct interest in silence localization where the goal is
to distinguish normal brain activity from abnormal silences.
Thus, in source localization paradigms applied to neurosci-
entific data [14–16], as also for the Event Related potential
(ERP) paradigm [17, 18], scalp EEG signals are aggregated
over event-related trials to average out background brain ac-
tivity and noise, permitting the extraction of the signal activity
that is consistent across trials. Consequently, as we demon-

The background brain activity in silence localization:

Line 84-101:

In order to not average out the background activity, we es-
timate the contribution of each source to the recorded EEG
across all electrodes. This contribution is measured in an av-
erage power sense, instead of the mean, thereby avoiding cancel-
ing out the contributions of the background brain activity.
Our silence localization algorithm, that we refer to as “Si-
lenceMap,” estimates these contributions, and then uses tools
that quantify our assumptions on the region of silence (conti-
guity, small size of the region of silence, and being located in
only one hemisphere) to localize it. Because of this, another
difference arises: silence localization can use a larger number
of time points (than typical source localization). E.g., 160 sec-
onds of data with the sampling frequency of 512 Hz provides
SilenceMap with around 81,920 data points to be used, boost-
ing the signal-to-noise ratio (SNR) over source localization
techniques, which typically rely on only a few tens of event-
related trials to average over and extract the source activity
that is consistent across trials.

To better address this comment and highlight the model we use for background brain activities in our SilenceMap algorithm, we have added the following sentence in Section IV A:

Lines 700-703:

between the sources increases. We assume a spatial exponen-
tial decay profile for the source covariance matrix \mathbf{C}_s , with
identical variances (σ_s^2) for all non-silent sources, whose sig-
nals model the background brain activities:

$$\begin{aligned} c_{sij} &= \sigma_s^2 e^{-\gamma \|\mathbf{f}_i - \mathbf{f}_j\|_2^2}, & \text{for all } i, j \notin \mathcal{S}, \\ c_{sij} &= 0 & \text{for all } i, j \in \mathcal{S}. \end{aligned} \quad (9)$$

Also, following the reviewer's comment, we realized that there is a sentence in Section IV A, which may have led to the misunderstanding that in order to use the baseline for silence localization, we need to know where the region of silence is located. This is not true as we don't make any a priori assumption about the location of the region of silence. To address this issue and prevent this misunderstanding for the readers, we have modified this part as follows:

Lines 680-689:

For this objective, we consider two different scenarios: (1)
there are no baseline recordings for the region of silence, i.e.,
no scalp EEG recording is available where there is no region
of silence, (2) with baseline recording, i.e., we consider the
recording of the hemisphere of the brain, left or right, which
does not have any region of silence, as the baseline for the si-
lence localization task. Note that the location of the baseline
hemisphere (left or right) is not assumed to be known a pri-
ori. Rather, locating the region of silence is the goal of this
approach.

Lines 722-726:

connecting path. (vii) For simplicity, we assume that silence
lies in only one hemisphere (as is the case for the 3 individ-
uals examined in the Results). However, the location of this
hemisphere is not assumed to be known (see Section IV C for
the details of SilenceMap algorithm with baseline).

4-part 1) Although the algorithm is interesting and the authors provide proof of concept on real brain recordings. The extent of damage in the patients used here is so great that any standard neuropsychological exam could easily pick this up and localize lesion with good accuracy. Therefore, no machine would be needed.

Response to 4-part 1) The reviewer is correct in noting that neuropsychological testing might, in some cases, suffice for determining where the lesion is. Of course, the sensitivity of these measures is somewhat coarse and localization is often at the level of the lobe with some indication of whether the lesion is in the left or right hemisphere (e.g. if performance is poor on

the Wisconsin Card Sorting Test or on Verbal/Visual fluency generation tasks, we might conclude that the lesion is in left or right frontal cortex or if performance is poor on list recall, we might conclude that the lesion is in temporal cortex and perhaps anterior and medial). Note that these measures do not permit characterization of the site or size of the lesion with any precision, and this contrasts directly with the localization achieved by SilenceMap algorithm. But perhaps even more relevant here is that, in the 3 cases presented here of children/adolescents with resection, there is rather minimal, if any, effect of the resection on behavioral performance, indicative of substantial plasticity in the children’s brain. We do not have detailed behavioral data for SN, but we do know that he is at school at the grade level commensurate with his age. As shown in Table V, for the other two patients, performance on neuropsychological tests post-surgically is high with only an occasional measure that is relatively low (e.g. 34th percentile for WISC in UD but post-surgical IQ is 118 (full scale), 123 (verbal), 108 (performance)). We present scores of various neuropsychological measures for the patients (OT and UD) and show that, dramatically, notwithstanding the large resection, performance is very good [21, 22] . The use of a silence localization algorithm would therefore, provide an important source of lesion site and size in postsurgical cases.

Table V was added to the paper in Supplementary Materials, Section H:

TABLE V. Pediatric patients’ neuropsychological evaluation test performance pre- and post-surgery [103]

Patient	Hemisphere	Detailed IQ measures	Vision or visual motor integration	Memory learning	Executive function	Academic skills/performance
OT	left	Pre-surgery: WASI: 122 (full scale), 125 (verbal), 114 (performance) Post-surgery: WASI: 127 (full scale)	Grooved pegboard: average (dominant hand)	CVLT-C: high average WRAML-2: high average	DKEFS: superior	WJ III ACH: above age and grade expectancy
UD	right	Pre-surgery: WASI: 116 (full scale) 135 (verbal), 97 (performance) Post-surgery: WASI: 118 (full scale), 123 (verbal), 108 (performance)	Grooved pegboard: 50th percentile (dominant hand)	not done	Working memory (from WISC-V): Post-surgery: 34th percentile	WJ III ACH: Reading: 63rd percentile Letter-Word: 67th percentile Passage: 56th percentile Calculation: 91st percentile

CVLT-C: California Verbal Learning Test–Children’s Version
D-KEFS: The Delis–Kaplan Executive Function System
Grooved Pegboard: Grooved Pegboard for Manipulation and Dexterity Testing
WASI: Wechsler Abbreviated Scale of Intelligence
WISC-V: Wechsler Intelligence Scale for Children–Fifth Edition
WJ III ACH: The Woodcock-Johnson III Tests of Achievement
WRAML-2: Wide Range Assessment of Memory and Learning–Second Edition

We have added the neuropsychological exam results and the above discussion in the Supplementary Materials, Section H, with the subtitle “Discussion on the performance of standard neuropsychological tests in silence localization”

Lines 1992-2020:

1992
1993

H. Discussion on the performance of standard neuropsychological tests in silence localization

1
1
1

Neuropsychological testing might suffice for determining where
the lesion is in many cases. Of course, the sensitivity of these mea-
sures is somewhat coarse and localization is often at the level of the
lobe with some indication of whether the lesion is in the left or right
hemisphere (e.g. if performance is poor on the Wisconsin Card Sort-
ing Test or on Verbal/Visual fluency generation tasks, we might con-
clude that the lesion is in left or right frontal cortex or if performance
is poor on list recall, we might conclude that the lesion is in temporal
cortex and perhaps anterior and medial). Note that these measures do
not permit characterization of the site or size of the lesion with any
precision, and this contrasts directly with the localization achieved
by SilenceMap algorithm. But perhaps even more relevant here is
that, in the 3 cases presented here of children/adolescents with resec-
tion (see Table II in Section II), there is rather minimal, if any, effect
of the resection on behavioral performance, indicative of substantial
plasticity in the children’s brain. We do not have detailed behavioral
data for SN, but we do know that he is at school at the grade level
commensurate with his age. As shown in Table V, for the other two
patients, performance on neuropsychological tests post-surgically is
high with only an occasional measure that is relatively low (e.g. 34th
percentile for WISC in UD but post-surgical IQ is 118 (full scale),
(verbal), 108 (performance). We present scores of various neu-
2016ropsychological measures for the patients (OT and UD) and show
that, dramatically, notwithstanding the large resection, performance
is very good [40, 103]. The use of a silence localization algorithm
would therefore, provide an important source of lesion site and size
in postsurgical cases.

And a brief version of this discussion in Section II, under the “Participants” subtitle:

Lines 358-366:

$k = 55$ out of $p = 1744$ total nodes. Despite the relatively large
sizes of the resected regions in these three pediatric patients,
there is rather minimal, if any, observable effect of the resec-
tion on performance, indicative of substantial plasticity in the
children’s brain. This suggests that we cannot characterize the
site or size of the resected areas with any precision using the
available neuropsychological exams (see Supplementary Ma-
terials, Section H for the detailed discussion on this along with
the neuropsychological test results for these patients).

[21] Liu, T.T., Nestor, A., Vida, M.D., Pyles, J.A., Patterson, C., Yang, Y., Yang, F.N., Freud, E. and Behrmann, M., 2018. Successful reorganization of category-selective visual cortex following occipito-temporal lobectomy in childhood. *Cell reports*, 24(5), pp.1113-1122.

[22] Liu, T.T., Freud, E., Patterson, C. and Behrmann, M., 2019. Perceptual function and category-selective neural organization in children with resections of visual cortex. *Journal of Neuroscience*, 39(32), pp.6299-6314.

4-part 2) Thus, there is a question of sensitivity here, if the researchers hope to use this in a medical setting as suggested in the introduction, they need to show it can be reliably used in smaller lesions. At this stage they might not be able to answer this question, but they should at least think forward on how to adjust the algorithm to increase localization sensitivity.

Response to 4-part 2) We thank the reviewers for this comment. Due to the COVID-19 situations, we have not been able to recruit more patients to test the performance of our SilenceMap algorithm. We acknowledge that further improvements and modifications may be needed to use SilenceMap for smaller and deeper regions of silence, e.g., improvements for estimation of the size of the region, where prior knowledge on the shape of the region (e.g., tumors) may be helpful. In addition, having a baseline recording (recording without any silence in the brain) may be helpful for localization of smaller and deeper regions.

To better address this comment, we have included this discussion in Section III (Discussion), under the “Limitations and future directions” subtitle:

Lines 616-620:

silence, SilenceMap needs to be improved. (iii) We plan to
extend our work to examine silence localization in individuals
with etiologies other than resection. We believe that further
improvements and modifications might be needed to use Si-
lenceMap for smaller and deeper regions of silence. (iv) We

REVIEWERS' COMMENTS:

Reviewer #1 (Remarks to the Author):

Thank you for addressing in detail the concerns of the reviewers. Although the number of subjects tested is small (due to the pandemic), which precludes sensitivity analysis, e.g., by analyzing performance of the proposed algorithm in smaller and deeper areas of 'silence', the proposed methodology and potential pitfalls are clear and the potential impact is high.

Reviewer #2 (Remarks to the Author):

We have carefully reviewed all of the responses and updates to the manuscript and are very pleased with changes. The manuscript has been greatly improved and is much more suitable for publication.

Neural silences can be localized rapidly using noninvasive scalp EEG

Alireza Chamanzar, Marlene Behrmann, Pulkit Grover

Response letter cover:

We thank the reviewers and the editor of our paper for giving us the exciting opportunity to publish our work in *Nature Communications Biology*. We are pleased that they have found the revised manuscript suitable for publication. In this final version of manuscript, we believe we have fully addressed the remaining editorial requests and suggestions regarding the style and format. Our detailed and itemized responses to these requests are available in the Editorial Requests Table, which is included in the final submission. We are grateful for these suggestions, which has, in our view, resulted in further substantial improvement in the readability and formatting our paper. As a major change, we moved the following sections from Discussion to Results since they include the results of additional analyses in the paper: “SilenceMap can localize the regions of silence with relatively little EEG data”, “Introduced error in silence localization by using symmetric brain models”, “Effect of error in the structural segmentation of MRI”, and “Effects of brain-to-skull conductivity ratio.” In addition, other minor edits and modifications have been made to meet the formatting and word count requirements.

We have also included the supplementary information as separate LaTeX and pdf documents in the final submission and we make very clear the means by which the data and algorithm can be accessed. Each figure of the main manuscript is submitted as a separate high-resolution pdf file and is not included in the main text following the style requirements.

Following is our point-by-point responses to the reviewers’ comments:

Color codes:

- Reviewers’ comments/suggestions/feedbacks
- Authors’ responses

REVIEWERS' COMMENTS:

Reviewer #1 (Remarks to the Author):

Thank you for addressing in detail the concerns of the reviewers. Although the number of subjects tested is small (due to the pandemic), which precludes sensitivity analysis, e.g., by analyzing performance of the proposed algorithm in smaller and deeper areas of 'silence', the proposed methodology and potential pitfalls are clear and the potential impact is high.

The authors thank the reviewer for the positive feedback and we are pleased that the reviewer is satisfied with the modifications in the revised version of our manuscript. We also appreciate that the reviewer recognizes the challenges we have faced in recruiting subjects during pandemic.

Reviewer #2 (Remarks to the Author):

We have carefully reviewed all of the responses and updates to the manuscript and are very pleased with changes. The manuscript has been greatly improved and is much more suitable for publication.

The authors would like to thank the reviewer for the positive feedback and we are pleased that the reviewer is satisfied with the modifications in the revised version of our manuscript.